# Enhancing the landing guidance of a reusable launch vehicle by improving genetic algorithm-based deep reinforcement learning using Hybrid Deterministic-Stochastic algorithm

**Larasmoyo Nugroho** [1,2¶]**, Rika Andiarti**[2‡]**, Rini Akmeliawati**[3‡]**, Sastra Kusuma Wijaya**[1‡]***

**1** Physics Dept., Universitas Indonesia, Depok, Indonesia, **2** Rocket Technology Center, National Research and Innovation Agency, Bogor, Indonesia, **3** School of Mechanical Eng., University of Adelaide, Adelaide, Australia

‡ RA, RA and SKW also contributed equally to this work.
¶ This author is the main contributor (larasmoyo.nugroho@brin.go.id).
* skwijaya@sci.ui.ac.id

**Data Availability Statement:** The data underlying the results presented in the study are available from https://github.com/arex18/rocket-lander.

## Abstract

The PbGA-DDPG algorithm, which uses a potential-based GA-optimized reward shaping function, is a versatiledeep reinforcement learning/DRLagent that can control a vehicle in a complex environment without prior knowledge. However, when compared to an established deterministic controller, it consistently falls short in terms of landing distance accuracy. To address this issue, the HYDESTOC Hybrid Deterministic-Stochastic (a combination of DDPG/deep deterministic policy gradient and PID/proportional-integral-derivative) algorithm was introduced to improve terminal distance accuracy while keeping propellant consumption low. Results from extensive cross-validated Monte Carlo simulations show that a miss distance of less than 0.02 meters, landing speed of less than 0.4 m/s, settling time of 20 seconds or fewer, and a constant crash-free performance is achievable using this method.

## 1 Introduction

Reinforcement Learning (RL) is a type of machine learning agent that aims to control an unknown object in an unknown environment by maximizing cumulative reward number.

### 1.1 State-of the-art

In 2014, when the Deep Neural Network reached its height, Tesla shows his technological prowess by deploying first highway lane keeping Autopilot Hardware 1.0 [1]. As deep neural networkapplications bloom fast and their underlaying tool(Tensorflow) matured, in just two years later, Autopilot Hardware 2.0 employedconvolutional neural network, recursive neural network and DRL whether partially or wholly mixture of them, to becomesfully autonomous

https://github.com/larasmoyo/GA_RSF_DDPG_
Landing_Rocket.

**Funding:** The research and scholarship is funded
by Direktorat Riset and Pengembangan,
Universitas Indonesia/NKB-483/UN2.RST/
HKP.05.00/2022/ - Dr. Sastra Kusuma Wijaya
Lembaga Ilmu Pengetahuan Indonesia/SK Kepala
LIPI No. 59/H/2020 - Mr. Larasmoyo Nugroho
Kementerian Riset dan Teknologi /Badan Riset dan
Inovasi Nasional/SK KaORPA No.15/III/HK/2022 -
Mr. Larasmoyo Nugroho. The funders had no role
in study design, data collection and analysis,
decision to publish, or preparation of the
manuscript.

**Competing interests:** The authors have declared
that no competing interests exist.

point-to-point driving system called Full Self-Driving [2]. Tesla's AI approach is considered
suitable for long-distance car cruise control, nevertheless, in its final phase of ridingjourney, it
struggles to park and falls short compared to more conventional control methods. Confirming-
gits development status, Tesla's Autopilot trailed behind its competitors in terms of fulfilling
its parking assistance function [3].This is where this research tried to propose, a switch
between the AI control, which is primarily a stochastic approach, and the PID control, which
represents a deterministic approach.

DRL is frequently used to solve a wide range of issues in robot manipulation [4], autono-
mous UAV/unmanned air vehicle [5], self-driving car [6], and other AI models. In a recent
series of virtual military fighter aircraft encounters, DRL performed better than humans [7, 8].
Rocket landing is one of the applications of reinforcement learning, where a highly accurate
guidance algorithm is required to attain pinpoint accuracy, despite the presence of numerous
challenging obstacles [9].

The backpropagation process and policy gradient discovery are the core aspects that define
deep reinforcement learning (DRL) [10]. Basic RL algorithms evaluated the outcomes of their
learning process based on predetermined values known as Q-values. Q-learning is the early
implementation of this RL [11, 12]. In [13] combined the idea of a Q-learning table with a convo-
lutional neural network (CNN), creating the framework for the Deep Q-Network (DQN), a well-
known DRL algorithm. DQN achieved remarkable outcomes when landing the lunar module in
the OpenAI Gym environment [14]. The drawback, DQN was only concerned with discrete state
spaces and action spaces. Different RL problems require continuous form for both states and
action spaces. Although DQN may have finished continuous tasks by converting continuous
areas to discrete ones, this will increase the unpredictability of the overall control mechanism.

The Deep Deterministic Policy Gradient (DDPG) method, as introduced by [15], was
developed by incorporating DNN (Deep Neural Network) techniques into the Deterministic
Policy Gradient (DPG) algorithm. This integration aimed to address certain challenges, as
mentioned by [16]. However, DDPG comes with its own set of significant concerns that
require careful consideration. Numerous aspects merit thorough examination and optimiza-
tion within the framework of DDPG, including the balance between exploration and exploita-
tion, reward shaping, hyperparameter selection, experience buffer management, and the
design of neural networks.

DDPG often encounters issues when it gets trapped in local optima, particularly when its
capacity for exploration is limited. To ensure that the DDPG algorithm can autonomously
explore and learn from its interactions within a complex environment, numerous parameters
need to be predefined. These parameters encompass factors like the size of the neural network,
learning rates, exploration strategies, reward shaping techniques, and the discount factor,
among others. It's important to note that these parameters are not adjusted automatically dur-
ing the training phase. Instead, developers must make informed choices based on their prior
expertise, as highlighted by [6].The training of an agent can be positively or negatively
impacted by the dimensions of the neural network, as noted by [17].

A large neural network will have many hidden layers or a high number of neuron nodes,
which could improve accuracy but also consume a lot of processing power and lead to overfit-
ting of predictions [18]. A model that has been overtrained or overfitted frequently performs
well during the training phase but poorly during the test set [19]. Many explorations are
required to obtain a respectable reward value, and in reinforcement learning it becomes prob-
lematic when the level of exploration is low [20]. Except when entering the goal state, the agent
typically receives a negative reward in delayed reward problems. The main role of reward
shaping functions is to accelerate learning. However, the agent's autonomy is diminished by
the extensive knowledge and work required for human-designed reward shaping.

## 1.2 Problem statement

Developing controller in the field of reinforcement learning, the objective is to train a controller agent to make decisions that maximize a reward signal. One approach to optimizing the reward function is to use an evolution algorithm to construct and maximize a reward shaping function. The evolution algorithm involves generating a population of candidate solutions and using selection, mutation, and crossover operators to produce new generations of candidate solutions. The reward shaping function can be optimized by finding the optimal coefficients via a genetic algorithm (GA) or by using a potential-based GA search. The GA procedure can be lengthy and requires careful chromosome mapping, but it has the advantage of being able to handle complex and dynamic environments.

Additionally, the hyperparameters of the neural network (NN) used in the reinforcement learning algorithm should also be optimized. This can be done by conducting a hyperparameter search using optimization algorithms such as evolutionary algorithms again, particle swarm optimization etc. Few aspects should be considered when choosing optimization method, whether improving the efficiency and accuracy of decision-making processes, reducing the risk of errors and failure, or enabling the development of more sophisticated optimization models.

The goal of this project is to develop a method for controlling a rocket using reinforcement learning. The method should involve optimizing the reward shaping function and the NN hyperparameters using a GA search to improve the performance of the agent in controlling the rocket. The resulting solution will be benchmarked against other control algorithms and the advantages of the GA-based individuals should be cross-validated.

The performance of landing guidance for a reusable launch vehicle will be improved further by combining the advantages of deterministic methods, due to their capacity to find optimal solutions, and stochastic methods, due to their capacity to handle uncertainty and navigate in complex/dynamic environments. Experiments on a hybrid deterministic-stochastic method for reinforcement learning will be conducted to offer insightful information about its potential.

## 1.3 Realization

The powered descent guidance (PDG) is an extremely difficult task that requires precise and smooth control to make a reliable and safe landing for both commercial and interplanetary rocket flights [21]. Deep Reinforcement Learning (DRL) has shown promise as an approach to solve PDG problem [9, 22], but the DDPG algorithm as a state-of-the-art DRL method needs to optimize its reward shaping function to face the challenges in managing exploitation vs. exploration dichotomy when searching solutions in under-explored environments [23]. A hybrid approach combining GA-based DDPGas stochastic method and PID as deterministic method has shown experimental success in defeating referenced DDPG and classicalPID controller. This research aims to further explore the potential of this hybrid approach for the development of a reliable PDG system.

## 1.4 Contribution statement

The following is a contribution made by this article:

- Investigate the effect of reward shaping function to the relation between state and action via mapping phase.

- Optimize the AI-based DRL algorithm DDPG, by finding the best form of reward shaping function.

- Apply GA methodology to find the best form of reward shaping with fitness to be maximized.

- Apply cross-validation to the RL algorithm training results against various RSF individuals intesting phase.

- Improve landing performance by combining GA-based DDPGas the stochastic method and PID as the deterministic method.

The rest of this paper is organized as follows: Section 2 shows the background. Related work is described in Section 3. Section 4 delves deeper into the recommended strategy. The experimental results of the suggested approach are discussed in Section 5. The paper comes to a close with conclusion in Section 6.

## 2 Background theory

The Landing Rocket Simulator (OpenAI Gym)., the Deep Deterministic Policy Gradient algorithm (DDPG), the Genetic Algorithm (GA), and Reinforcement Learning are all discussed in this section.

### 2.1 Reinforcement learning

The general learning technique of reinforcement learning involves an agent interacting with the environment to learn how to act in a new environment without any prior knowledge by maximizing reward or minimizing punishment [24]. Policy is the process of connecting observations (states) to actions [25]. An agent learns to produce the best policy in discrete time for each step that is provided by the environment in the early conception of reinforcement learning. The Markov decision process is used to explicitly describe the RL problems (MDP). The agent outputs an action at each time step t, the environment receives the current state s, returns the next state s', and returns the reward r.

The stochastic policy (s) of the agent, which maps the current states to a probability distribution over potential actions, determines the agent's course of action: S → P(A).

The MDP is a 5-tuple (S, A,R,P,T), where S stands for the state space, A for the action space, and R for the response space. R is the reward function that describes the immediate reward $R(s,a)$ achieved at each state-action pair, $P(s'|s,a)$ is a transition function for the environment that predicts the following state $(s')$ given a current state-action pair $(s,a)$, and T(s), which denotes the Boolean condition state where 1 is at termination state and 0 if otherwise.

The cumulative reward received by an agent, starting at time step t along the trajectory of interactions with an environment, is defined by Eq (1) as follows:

$$R = \sum_{t}^{T} \gamma^t r(s, a) \tag{1}$$

where $r(s,a)$ is the reward function w.r.t states $s$ and actions $a$, with a discounting factor $\gamma\epsilon$ [0,1].

When the agent maximizes the anticipated accumulated reward E[R], the optimal strategy (policy) is said to have been attained. Learning a policy that maximizes the expected return in the form of the global cost function is the objective of reinforcement learning:

$$J = E[R] \tag{2}$$

The Q value function or Q-function in accordance with stochastic policy π can be used to describe the expected return (next discounted reward) after the agent takes action in the

following state s.

$$Q_\pi(s, a) = E[R|s, a] \tag{3}$$

Introducing Bellman equation R = r(s,a) + γE (Qπ (s',a')) and the Q-function will have a recursive relationship:

$$Q_\pi(s, a) = E[r(s, a) + \gamma E(Q_\pi(s, a))] \tag{4}$$

If the policy is deterministic $\mu$ then there is a function $\mu: S \to A$ which map states into action and the Q-function becomes:

$$Q_\mu(s, a) = E[r(s, a) + \gamma E(Q_\mu(s, a))] \tag{5}$$

The greedy policy $\mu(s)$ = argmax Q is used in the widely used off-policy algorithm Q-learning [26]. The DDPG method employs deep learning to deterministically estimate the policy function $\mu(s)$ in addition to approximating an action-value function Q.

## 2.2 Deep Deterministic Policy Gradient algorithm (DDPG)

The Deep Q-Network (DQN) algorithm's discreteness of data problem was solved by the actor-critical, off-policy, model-free reinforcement learning algorithm known as DDPG [27]. DDPG is a continuous action space algorithm.

In the following, the fundamental components of the DDPG technique will be discussed. The actor-critic framework serves as the basis for the DDPG algorithm [27]. The actor and the critic are two segments that are present, according to this. The actor maintains a policy. The policy outputs an action in response to the input state. To assess the actor's suitability, the critic approximates the action-value function.

An actor-critic architecture employs two independent networks, one called the actor network $\theta^\mu$ with weight $\theta$ to approximate the output policy $\mu$ and the other called the critic network $\phi^Q$ with weight $\phi$ to estimate the Q-value. Both networks (actor and critic) have their own target networks $\theta^{\mu'}$ and $\phi^{Q'}$ (with weights θ' and φ', for actor and critic respectively) for greater learning robustness.

In an actor-critic architecture, there are two distinct networks: the actor network, denoted as θμ, which approximates the output policy μ using weight $\theta$, and the critic network, denoted as $\phi^Q$, which estimates the Q-value. Each of these networks (actor and critic) is accompanied by its own target networks, $\theta^{\mu'}$ and $\phi^{Q'}$ (with weights θ' and φ', respectively), to enhance learning stability.

The target network weights are updated using a soft updating algorithm, while the main network weights are updated using a stochastic gradient descent [28]. A diagram of the DDPG algorithm is shown in Fig 1.This diagram is sufficient to illustrate the various challenging issues that an RL-based DDPG controller agent must deal with in a continuous space problem. Once the actor has received their training, they must follow forwarding policies with increasing reward values until the ideal one is attained.

Actor networks (main and target) produce policy values as outputs using states (observed current and observed next) as inputs. A stochastic policy $\mu(a \,|\, s)$, which provides a probability distribution over actions, is what policy gradient methods traditionally produce. For the DDPG actor, we take into account a parameterized deterministic policy $a = \mu\theta(s)$ with parameter θ. The weight of the neural network is represented by the policy parameter $\theta$ following the deterministic policy gradient theorem: $\frac{\delta}{\delta\theta}J(\theta) = \nabla_\theta J(\theta)$. By employing a gradient-based policy, the actor cost function J() is maximized, updating the actor network's weights. It being a maximization problem, we take the gradient descent/ascent with the partial derivative of the

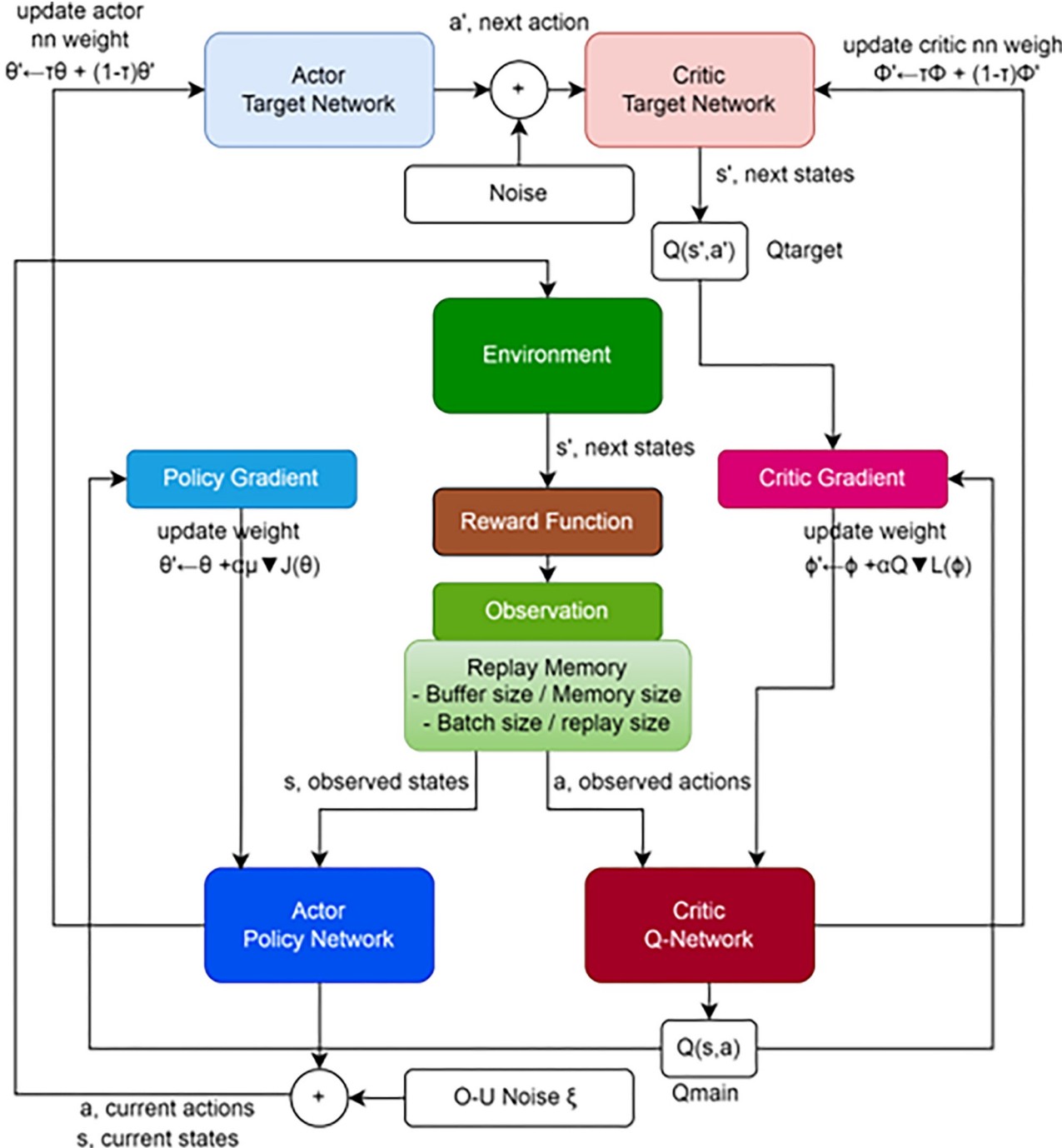

**Fig 1. The entity relationship of DDPG algorithm.**

objective with respect to the policy parameter in order to optimize the policy J. And represents the actor main network's learning rate.

$$
\theta_\mu^{\text{new}} \leftarrow \theta_\mu^{\text{old}} + \alpha_\mu \frac{\delta}{\delta\theta} J(\theta)
$$

$$
\theta_\mu^{\text{new}} \leftarrow \theta_\mu^{\text{old}} + \alpha_\mu \nabla_\theta J(\theta)
$$

(6)

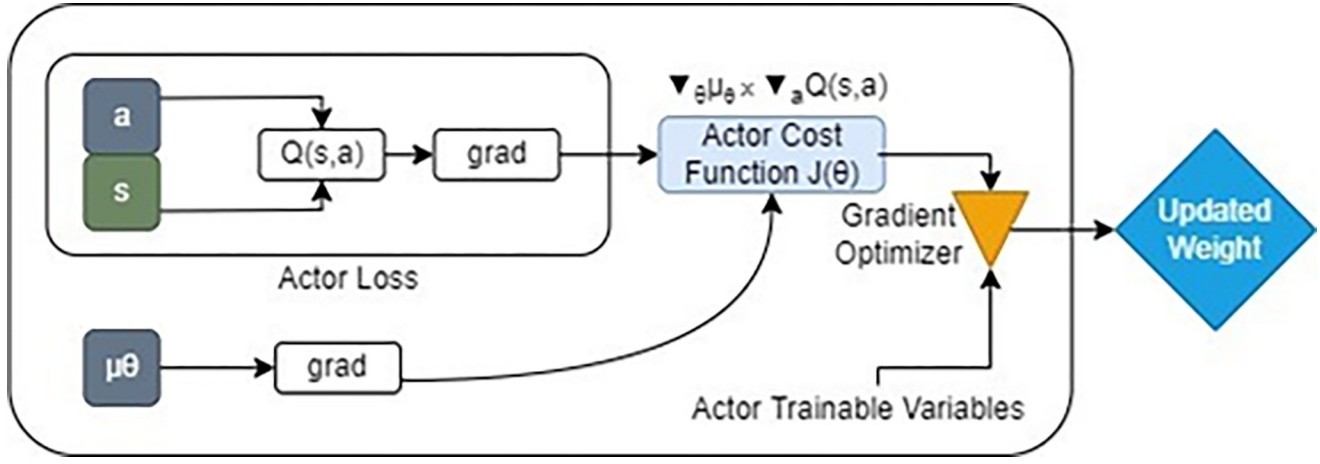

**Fig 2. Internal process of a policy gradient block.**

By analyzing the policy gradient as depicted in Fig 2, the actor is updated. The expected return from the start distribution J with respect to the actor parameters and the estimated Q-values by the critic is calculated using the chain rule of calculus.

$$\nabla_\theta J(\theta) = \frac{1}{N} \sum_i [\nabla_\theta \pi(s|\theta^\mu) \nabla_a Q(s,a)]$$

$$\nabla_\theta J(\theta) = E[\nabla_\theta \mu(s|\theta^\mu) \nabla_a Q(s,a|\theta^Q)]$$

(7)

The state-action value function $Q(s,a)$, the mini-batch size N, the policy neural network parameters and, the stochastic and deterministic policies, and J, the actor cost function that needs to be maximized, are all included in the formula. The policy parameters are updated by this gradient policy in a gradient ascent setting. The actor-critic architecture results from this equation, where $\nabla_\theta \mu(s)$only depends on the parameterized actor and $\nabla_a Q(s,a)$ acts as a kind of critic by advising the actor to change its policy in the direction of actions that will yield greater rewards.

When the critic simply outputs the Q-value, the question is how to extract the gradient of the Q-value with respect to the action $\nabla_a Q_\phi(s,a)$. Fortunately, automatic differentiator libraries like TensorFlow, Theano, PyTorch, and others can automatically generate this gradient while computing the weights of deep neural networks. The finite difference method (Euler) is the most practical to be used to approximate this gradient when other methods are not available. The Q-value must be calculated in $(a+da)$, where $da$ is a very small change in the executed action, and the gradient must be estimated using

$$\nabla_a Q_\phi(s,a) \approx \frac{Q_\phi(s, a+da) - Q_\phi(s,a)}{da}$$

(8)

The optimal policy is found by maximizing the global reward returns or the true Q-value of all actions.

$$Q^\pi(s,a) = E_\pi[R(s,a)]$$

(9)

The discrete-based Q-learning algorithm cannot be directly applied to continuous action spaces because in continuous spaces, finding the greedy policy necessitates optimizing $a(t)$ at

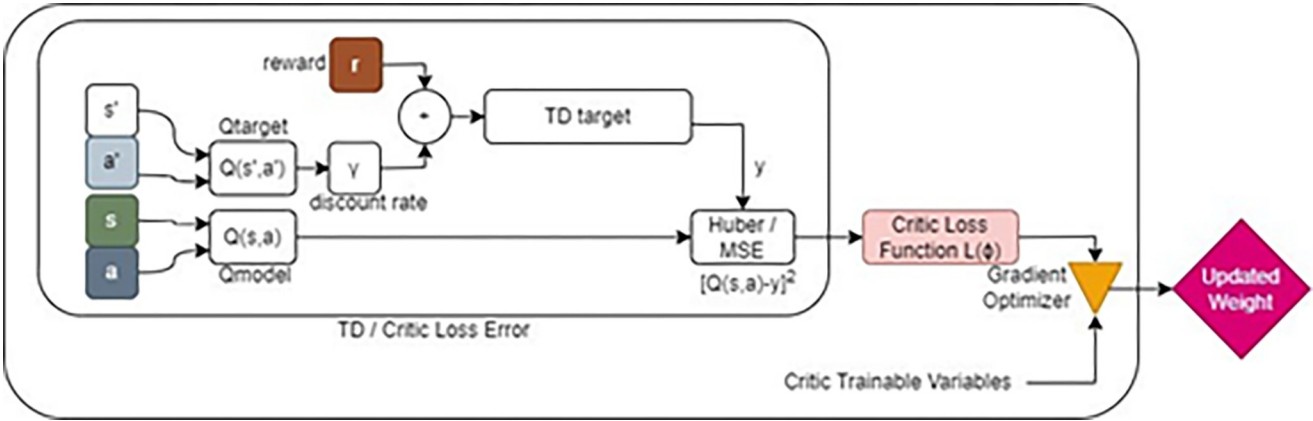

**Fig 3. Internal process of a critic gradient block.**

each timestep $t$. Dynamic programming's fundamental premise is that all state-action combinations have a true Q-value, which is then determined via policy evaluation by choosing the best action which yields the highest Q-value:

$$a^*_t \leftarrow \mathrm{argmax} Q(s, a) \tag{10}$$

In the continuous problem, the gradient of the objective function J(θ) is the same as the gradient of the Q-value estimation. Given an unbiased estimate of the value $Q^\mu(s,a)$ of any action a in s, changing the policy $\mu^\theta(s)$ in the direction of $\nabla \theta Q^\mu(s,a)$ results in an action with a higher Q-value, and hence a higher associated return:

$$\nabla_\theta J(\theta) = E_{s \sim \rho^\mu}[\nabla_\theta Q^\mu(s, a)|_{a=\mu_\theta(s)}] \tag{11}$$

The actor network is then updated by using the action gradient estimated with the critic network (Fig 3).

$$\theta_\mu^{new} \leftarrow \theta_\mu^{old} - \alpha_\mu \nabla_a Q(s_t, \mu(s_t|\theta_\mu)|\theta_Q) \nabla_a \mu(s_t|\theta_\mu) \tag{12}$$

Where $\alpha_\mu$ is the learning rate of the actor network.

By improving the policy incrementally, the stochastic gradient maximize the Q-value of the critic network $Q(s,a)$. The critic-value function $Q(s,a|\theta_Q)$ is learned using the Bellman equation as in DQN [28].

$$Q_\mu(s, a) = \mathrm{E}[r(s, a) + \gamma Q_\mu(s, a)] \tag{13}$$

The state s, the next state s' and the return Rt are obtained by executing action At = π(S | θμ) through the policy μ.

The researchers [29] established in their study that the policy's performance is represented by this deterministic policy gradient. As seen in Fig 3, the critic network determines the next state s' of the actor network by assessing the state-action *(s,a)* pair value performance in order to maximize Q-value [30]. The output $Q(s,a|Q)$ of the critic network is used to compute the definition of the critic loss function L.,

$$L(\phi_Q) = \frac{1}{N} \sum_i (Q(s, a) - y_i)^2 \tag{14}$$

where N is minibatch size sample from the replay buffer, $i$-index refer the $i$-th sample, and $y_i$ is

the temporal differencetarget.

$$y_i = r + \gamma Q(s', a') \tag{15}$$

which is computed from the sum of the immediate award r and the *output Q(s',a' | $\phi_Q'$)* of the critic target network times the discount rate γ.

By using gradient descent to minimize the critic loss function for the critic network L(Q), which is represented as follows, the weights of the deep critic network can be updated [31].

$$\phi_Q^{new} \leftarrow \phi_Q^{old} - \alpha_Q \nabla_\phi L(\phi_Q) \tag{16}$$

where $\alpha_Q$ is the learning rate of the critic network.

The actual (or subsequent) action will be decided by these policy values after they have been coupled with [32] noise. The last challenge is exploration. The policy's determinism means that it might quickly produce the same results while dispensing with potentially more advantageous options. There are some places that are naturally noisy, which promotes exploration, but this cannot be taken for granted. In DDPG, the environment is explored using a deterministic action with additive noise as the solution.

$$a = \mu_\theta(s) + \xi \tag{17}$$

Any kind of additive noise could be used, but the most practical method is the Ornstein-Uhlenbeck process [32], which generates temporally correlated noise with zero mean. Physics models the velocity of Brownian particles with friction using Ornstein-Uhlenbeck processes. It uses a stochastic differential equation (SDE) to update the variable $x_t$

$$dx_t = \theta(\mu - x_t)dt + \sigma dWt$$
$$dW_t = N(0, \ dt) \tag{18}$$

with μ is the mean of the process (usually 0), θ is the friction (how fast it varies with noise) and σ controls the amount of noise [33]. Fig 4 shows two independent runs of an Ornstein-Uhlenbeck process, successive values of the noise variable $x_t$vary randomly but coherently over time.

Experience replay can also be used with DDPG for reducing bias introduced from temporally correlated transitions data and more learning efficiency. To solve computational resource consumption difficulties, the DDPG approach eventually borrows the replay buffer functionality from DQN [34]. Off-policy algorithms, such as DDPG, can benefit from training in random mini-batches across unrelated environmental interactions while preserving the concept of i.i.d. replay buffers (independent identically distributed). Stability in continuous domains is improved by using deterministic techniques. Training is also more stable due to the "soft" updating rule that updates target networks.

## 2.3 Hybrid Deterministic-Stochastic Controller (HYDESTOC)

DDPG as a state-of-the-art RL algorithm exploits well deterministic policy gradient to search and converge into solution quickly. However, the solutions can be prone to bias results due to the inherent perturbations created by OU noise as a way to explore the environment efficiently. This shows that DDPG is basically a stochastic approach to construct controller, and internal perturbations are the natural risk of it.

On the other hand, deterministic approach to construct controller such as the classic PID, can deliver high performance/accurate rocket landing guidance, but the cost of energy needed is very high, therefore this approach is hard to be optimized and easy to converge into suboptimal solutions.

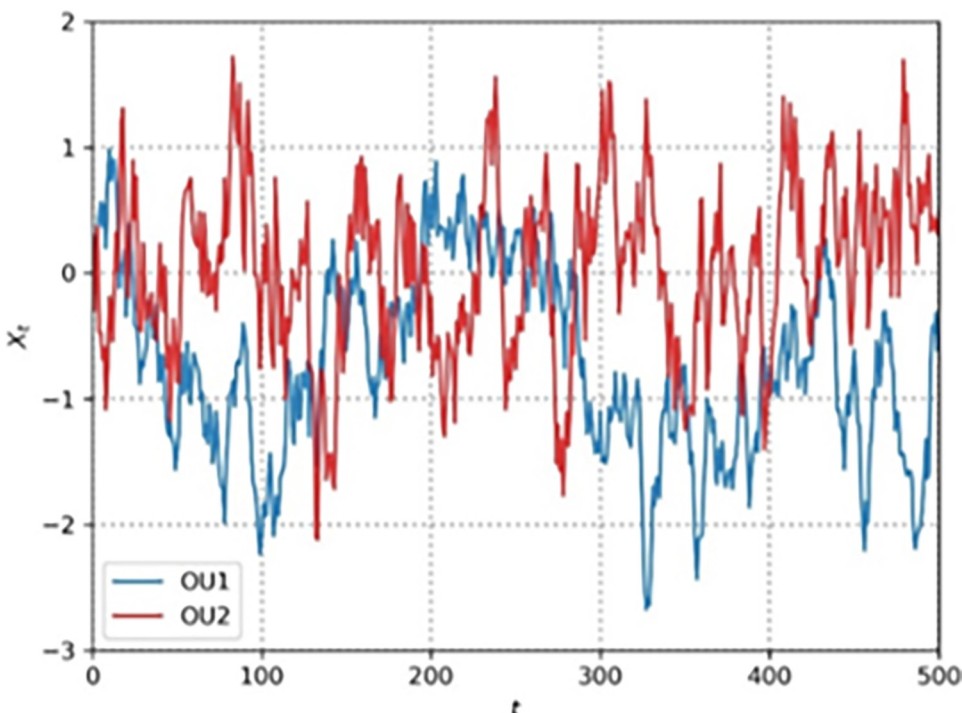

**Fig 4. Two typical OU-noise process.** Noise variable in normal scale versus time in steps.

The proposed Hybrid Deterministic-Stochastic Controller (HYDESTOC) approach addresses these limitations by combining the advantages of both deterministic and stochastic control approach. This HYDESTOC allows to switch between GA-based DDPG as stochastic controller and PID as deterministic one. It can regulate the flight time using the stochastic part or the deterministic part. It employs a hybrid objective function that is switched based on altitude state space.

Let considera controlled landing process governed by a deterministic time-invariant differential equation

$$\dot{x} = Ax + Bu$$
$$y = Cx$$

(19)

where x is the state vector, A is the state matrix, B is the input matrix, and u is the control input. And the deterministic control input is a PID controller in continuous form:

$$u(t) = Kp.e(t) + Ki \int_0^t e(t)dt + Kd.de(t)/dt$$

(20)

where $u(t)$ is the control input, $e(t)$ is the error signal (the difference between the desired behavior and the current system state), $Kp$, $Ki$, $and$ $Kd$ are the proportional, integral, and derivative gains. The PID controller generates a control input that can effectively regulate the system's behavior and achieves the desired response.

The optimal cost for deterministic case is

$$J = E[x(t)] + \int_0^t f(x(t), u(t))dt \tag{21}$$

The cost function includes a function of the states, inputs, and time. But in our case, the PID controller is not optimizing the input signals, therefore the second term is omitted,

$$J = \min E[x(t)] \tag{22}$$

where the minimization function deals the expected states only, in this deterministic case is the miss distance state.

For the stochastical part, the state space equation is

$$\begin{aligned} \dot{x} &= Ax + Bu + w \\ y &= Cx + Du + v \end{aligned} \tag{23}$$

where x is the state vector, u is the control input, y is the output, w and v are zero-mean random processes representing the disturbance and measurement noise. A, B, C, and D are matrices that represent the system's dynamics, inputs, outputs, and direct feedthrough, respectively.

In RL, the control input, or action, is determined by a policy $\pi$, which maps the current state of the system to an action. The control input at time t, u(t), is determined by the action which is resulted from computation of current state of the system, s(t), and the current policy $\pi(t)$ according to the following equation:

$$u(t) = a(t) = \pi(s(t)) \tag{24}$$

The policy represented the output of the neural network. It is updated as the agent interacts with the environment and learns from its experiences. The RL controller finds an optimal policy that maximizes a reward signal. The cost function for a stochastic process is expressed as the maximization of the expected value of a function that represent the cost of the mapping process from state to action space, given the current policy. This can be expressed as:

$$J = -\max E[R(s, a, t)|\pi] \tag{25}$$

where:

E[.] is the expected value operator.

R(x, a, t) is the reward function that accumulate the reward gained across time t.

This stochastic cost equation reflects the trade-off that can be achieved by maximizing the rewards or minimizing the fuel cost in our stochastic case. Reward function is basically a gradient descent that by maximizing it can lead to the optimal policy, or minimizing the fuel cost, as shown by the negative sign, as it is vice versa. The quality of a policy is determined by this trade-off that yield accuracy, cost efficiency, and robustness.

The proposed HYDESTOC will switch the action inputs according to the altitude threshold.

$$u(h, t) = \begin{cases} \pi(s(t)), & if \ h_{switch} \leq h_{initial} \\ Kp.e(t) + Ki \int_0^t e(t)dt + Kd.\dfrac{de(t)}{dt}, & if \ 0 \leq h_{switch} \end{cases} \tag{26}$$

This switch is designed to help the decision-making process to search and find the right altitude that produce the optimal action policy. (Sethi, 1988)

The HYDESTOC approach can be applied to a variety of control tasks, such as robotic manipulation, that is compatible to traditional deterministic and the modern stochastic methods.

## 3 Related works

In this section, the recent research efforts in the domain of DRL for PDG as well as optimization of DRL reward function shaping are presented.

### 3.1 DRL for Powered Descent Guidance

Studies that have approached Powered Descent Guidance (PDG) as a continuous control problem through the use of Deep Reinforcement Learning (DRL) algorithms are explained in this section.

Due to the exclusive study of landing guidance using optimal control [35] is not researched in this paper. Paper [36] compares RL to optimal control, but the density of sample is not determined reliably and without gradient search. [37] uses PID and observer that can be used in this research. [38] uses RL and optimal control as combined controller, is an interesting research.

First stage rocket landing is used as a model case by [39–42] and majority employed AI as controller. OpenAI Gym physics simulator is considered as a stable yet robust platform to train any discrete or continuous vehicle model.

The interesting trend is the usage of DDPG or PPO (proximal policy optimization) as the RL controllers, where both used target networks. To address the issue of greediness associated with the maximum value of $Q(s)$, a target network can be employed or alternatively, continuous data can be directly utilized for actions as a solution.

The surveyed features of DRL usages for solving PDG problems are summarized in Table 1, while our perception of the surveyed drawbacks is shown in last column. Our research is decided to concentrate on a modeled reusable launch vehicle (RLV)and try to solve the guidance of landing phase [40] using GA-based DRL validated by introduction of wind disturbance and compared against established algorithms in an effort to plug the drawbacks.

### 3.2 Shaping of reward functions

The reward function plays a crucial role in shaping the behavior of a Deep Reinforcement Learning (DRL) algorithm. Two ways to shape the reward function include changing the weight and adding/removing terms after the training conditions. A planetary spacecraft's trajectory correction can be improved through the use of a well-designed reward function in DDPG as demonstrated by [43]. There are two methods to construct the reward function, including using a potential function as the shaping reward and using a model-based heuristic function. According to [44], reward shaping can improve the reward process and training speed while preserving the action policy's optimality. The ability to maintain a positive gradient of reward is crucial to achieving this. The study by [45] used GA to examine how the parameters of the reward function's weights could be shaped.

The reward function must take into account a number of factors in Powered Descent Guidance (PDG) problems, including distance to go, rocket landing velocity, angular attitude-keeping, crash avoidance, and wind disturbance compensation. The reward function is often chosen to be linear for changes in mass and quadratic for changes in the relative position and velocity of the lander. According to recent research by [46], a valid reward function is crucial to the implementation of PPO as the policy will learn to explicitly maximize this function. In the case of 6-DOF docking operations, the reward function is made up of several terms that collectively consider state tracking errors, control effort minimization, collision avoidance, and successful docks, all fairly weighted thanks to carefully chosen coefficients.

**Table 1. Previous usages of DRL in solving PDG problems.**

| No. | Authors | Algorithm | Case | Features | Disadvantages |
|---|---|---|---|---|---|
| 1 | [35] | Survey—CVX<br>• SCP<br>• MPC<br>• lossless cvx<br>• SNOPT, IPOPT | PDG problem:<br>• rocket landing<br>• apollo docking<br>• asteroid landing<br>• space telescope<br>• 2nd stage goes to orbit<br>• ascent<br>• Soyuz capsule landing<br>• orbit transfer | • these are optimal control problem cases | • not any A.I based algorithm is used |
| 2 | [36] | GPOPS trajectory as input<br>DNN +<br>Bayesian regulator as trainer<br>MDP<br>• RL in value and reward<br>+ policy<br>optimization | Mars landing | • problem in MDP<br>with policy optimization<br>• Dynamically retarget position<br>• RL can adapt to<br>noise and system de-lays | • no reliable heuristic for determining sample density s.t. trial and error<br>• not using gradient search so that large number of simula-tions needed |
| 3 | [37] | • PID + Observer<br>• DisturbanceObserver<br>• Learning Observer | Mars landing | • wind disturbance<br>• wind compensation<br>• CL stability<br>• zero error approach | • LO + DO must be coupled |
| 4 | [38] | RL + Pseudospectral | Mars Entry + PDG Mid L/<br>D = starship<br>style | • combination of hd-<br>pseudospectral (propellant)<br>+ RL (handover) + CVX<br>(nominal + disturbed) | |
| 5 | [39] | DDPG | Lunar Landing | • action noise vs parameter noise vs both noises | • too few parameters variation |
| 6 | [40] | DDPG<br>MPC<br>PID<br>Q-Learning | Rocket Landing -<br>Monte Carlo test | • OpenAI Gym usage as<br>physics simulator | • still 2-dimensions |
| 7 | [41] | GEKKO Optimiza-<br>tion | Rocket Landing | • 6-Dof Model<br>• 3-D Visualization<br>• Pybullet physics simulator | • OCP controller still not optimized against wind |
| 8 | [9] | PPO | Rocket Landing | • 4 weighted terms of reward<br>function | • no specific physics / environment simulator |
| 9 | [14] | Q-Leaming DQN,<br>DDQN PID | Lunar landing | • DQN exploit experience replay<br>QL = PID yields better results | • no external disturbance<br>• no trajectory comparison from multi algorithm |

### 3.3 Optimization of reward shaping function

To maximize rewards in deep reinforcement learning, several optimizations must be conducted, including reward function shaping and hyperparameter tuning [47]. The Bayesian Optimization (BO) method [48], which uses a black-box objective function, model surrogate function, and acquisition function, is widely used for hyperparameter tuning. However, BO requires a lot of computational power. Alternative methods include Multiagent Evolutionary Reinforcement Learning (MERL), which combines gradient-based and gradient-free optimization, and Evolutionary Reinforcement Learning (ERL) [49], which uses an Evolutionary Genetic Algorithm [50] to address challenges like ineffective exploration and unstable convergence.

## 4 Proposed framework

The main contribution of this study is presented in this chapter, where the GA searches the space of reward shaping function (RSF) elements using Potential based GA (PbGA) method which maximizes achievements of DDPG agents in terms of proposed fitness value based or Potential based Functions (PbF) in the OpenAI Gym environment.

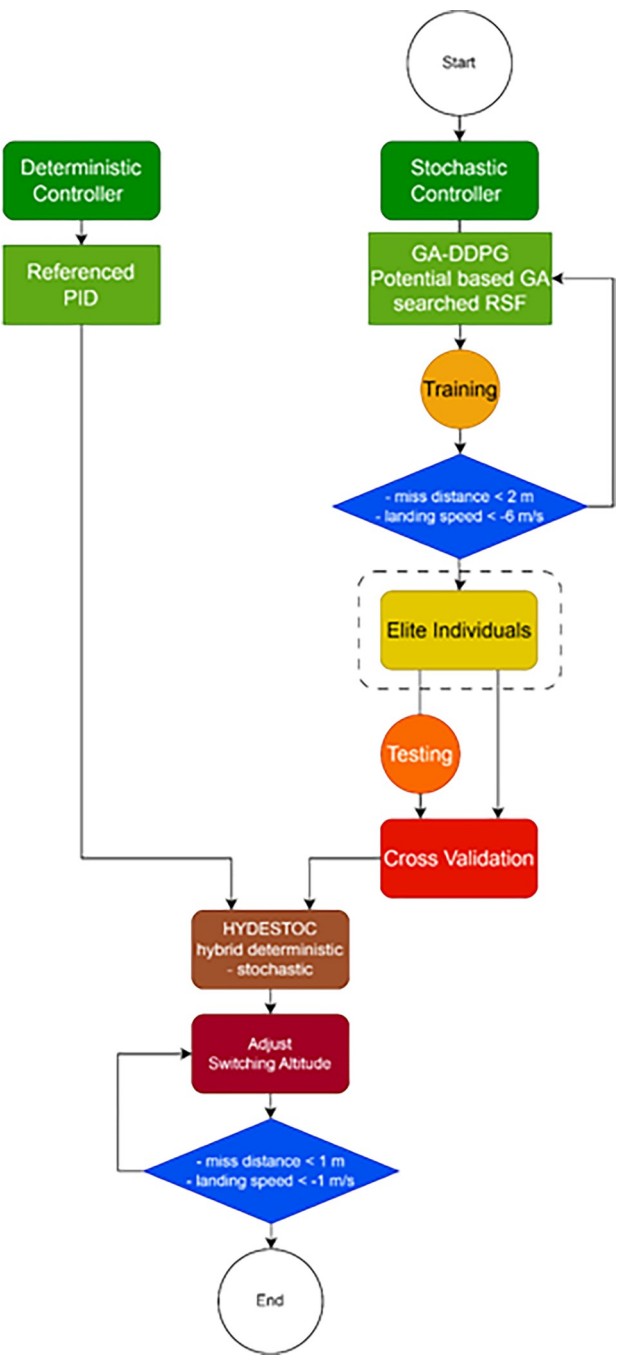

**Fig 5.**

Fig 5. Flowchart illustrating the pipeline used by the research methodology. The pipeline is organized into two main processes, stochastic and deterministic. The stochastic one is a lengthy process, and comprises the main proposed method. The deterministic controller is referenced in the benchmarking test. Finally, the HYDESTOC combines both approaches and find optimal yet robust hybrid controller.

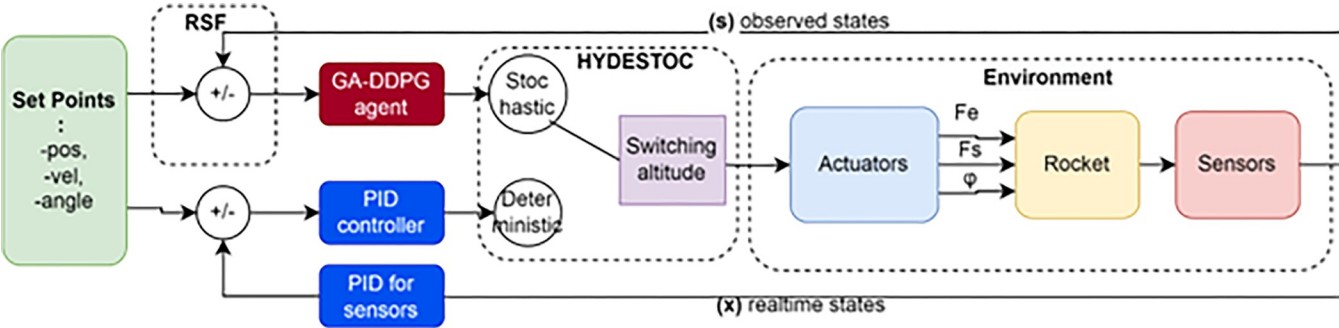

**Fig 6. A proposed GA-DDPG augmented with PID framework named as HYDESTOC.**

In Fig 6, the proposed GA search method is shown applied to the RSFblock which lies outside of the GA-DDPG Agent. Every aspect that comes from outer side of the agent are considered as the environment.

## 4.1 DDPG agent neural network structure

In the OpenAI Gym rocket-lander environment, the input to the actor network is a 6-dimensional state vector ($dx$, $dy$, $vx$, $vy$, $\theta$, $\theta dot$) plus optional terminal condition states of sensor inputs (left & right landing gears). The output is a 3-dimensional action vector, consisting of three continuous actions: main engine throttle, side engine throttle, and thrust vector nozzle angle. The DDPG architecture uses two types of neural networks: critic networks and actor networks. The critic network serves as the gate for incoming data like states, actions, and rewards, and returns Q-values as its main network output, which is used to update the weights of the neural networks. The ADAM algorithm updates the weights of the networks, with a decaying weight for stability during offline RL experiments. The actor target network outputs actions, with a separate target network for stability during the learning process. The weights of the target network are updated to match the main network's weights every predetermined number of steps.

Before entering the critic network, the three-dimensional action vector "a" left the actor network. For each of the throttling functions Fe and Fs, two single units with tanh activation functions are distributed, where 0 for Fe denotes no acceleration to the main engine and 1 denotes full throttle. For Fs or side thrusters, 1 denotes full right thrust, 1 denotes full left thrust, and 0 denotes no acceleration.

A maximum right turn of 1 degrees and a maximum left turn of 1 degrees are indicated by the single unit that makes up the TVC swivel angle's Tanh activation function.

With the number of nodes shown in Fig 7 below, both actor and critic networks have two hidden layers.

In the meantime, these network's hyperparameters are set using two configurations—optimized version and reference version—elaborated in Table 2. ALR stands for Actor Learning Rate, CLR stands for Critic Learning Rate, and $\gamma$ a discount factor.

At each discrete time stage, a batch of samples is used to update the core network.

The batch size is calculated using the fifth value of each individual's position vector (as specified by the individual's arrangement). The replay memory is accessed via these batch of samples.

## 4.2 Potential-based reward shaping functions

The term "shaping" in reinforcement learning originates from psychology where it was introduced by B.F. Skinner [51]. Shaping is a training method where animals are given rewards for

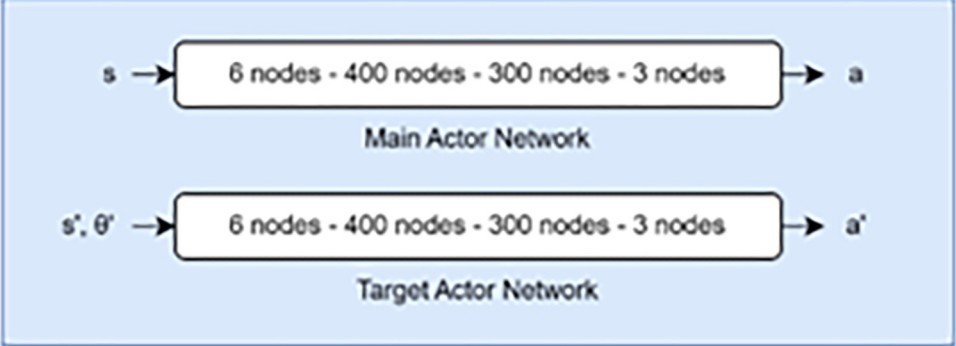

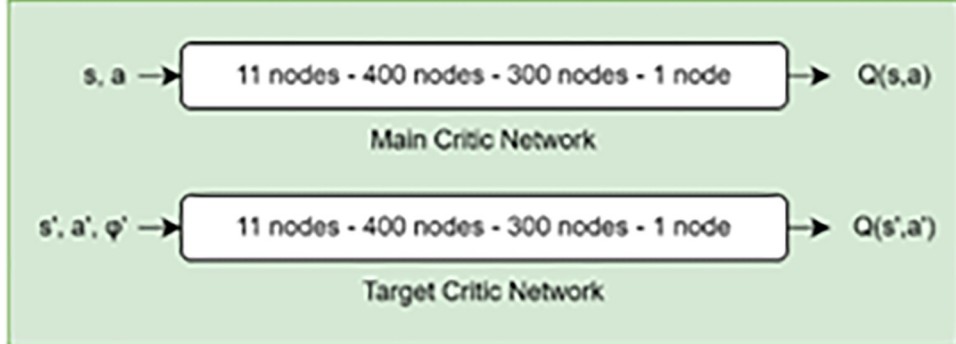

**Fig 7. Input-output relations of the DDPG neural network architecture.**

completing subgoals, which improves their behavior towards the main objective [52]. Reinforcement learning algorithms can be enhanced through shaping rewards, where an additional reward signal directs the agent towards states with the highest rewards. Shaping rewards, also known as progress estimators, were used successfully in one of the earliest real robot reinforcement learning experiments by Mataric to teach multirobot foraging [53, 54]. The shaping rewards were used to decompose the overall goal into subgoals.

The reward shaping function (RSF) in Fig 8 is composed of a reward function and the shaping operation, explained in a detailed diagram in Fig 9. The environment is broken down into components: actuators, plants (rocket vehicle model), sensors, and states. The plant is non-linear as the fuel level is constantly decreasing. Shaping rewards are used to direct the agent towards states with the greatest rewards and can be used to decompose the overall goal into subgoals. However, shaping rewards must be done carefully as the agent may become trapped in a suboptimal behavior. The Potential-based Genetic Algorithm (PbGA) will search for the best set of reward shaping parameters to achieve the highest level of fitness, which adheres to the designer's desired conditions.

**Table 2. Two versions of actor and critic network hyperparameter settings.**

| Hyperparameters | Optimized Version | Reference Version |
|---|---|---|
| ALR | 0.001 | 0.0001 |
| CLR | 0.01 | 0.001 |
| Batch size | 100 | 100 |
| Buffer size | 1,000,000 | 1,000,000 |
| $\gamma$ | 0.99 | 0.99 |

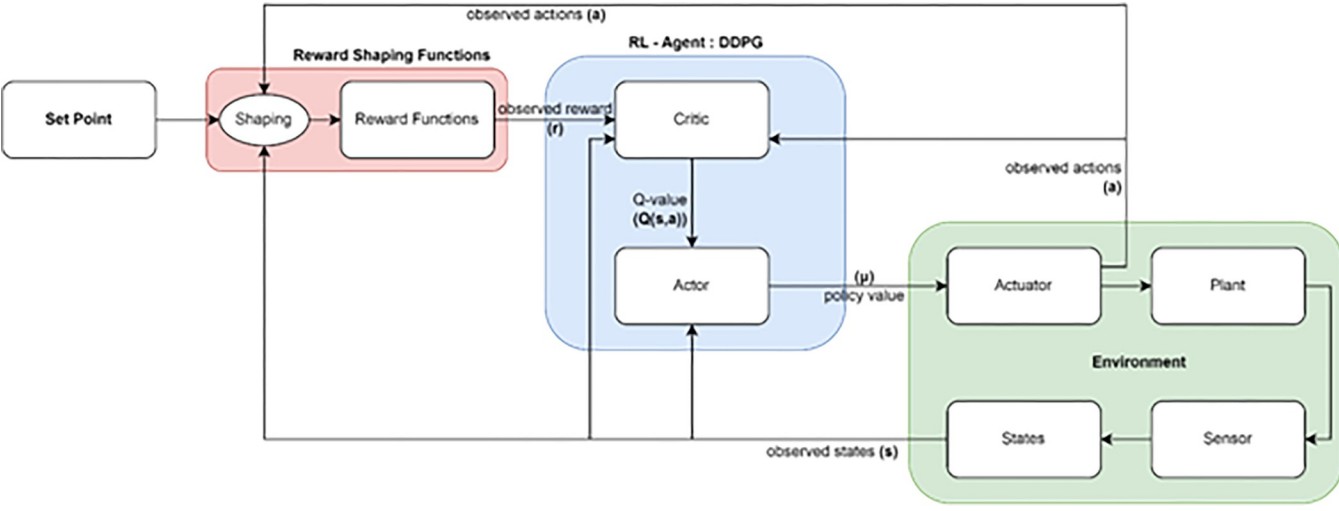

**Fig 8. A more detailed interaction between DDPG controller agent–GA-searched reward shaping function–Environment.**

This paper introduces the Potential-based Fitness (PbF) algorithm to quantify the combination of the best reward shaping parameters. The environment block in Fig 9 contains the flight vehicle, the flight simulator, and the Reward Shaping Function (RSF). The Reward Shaping

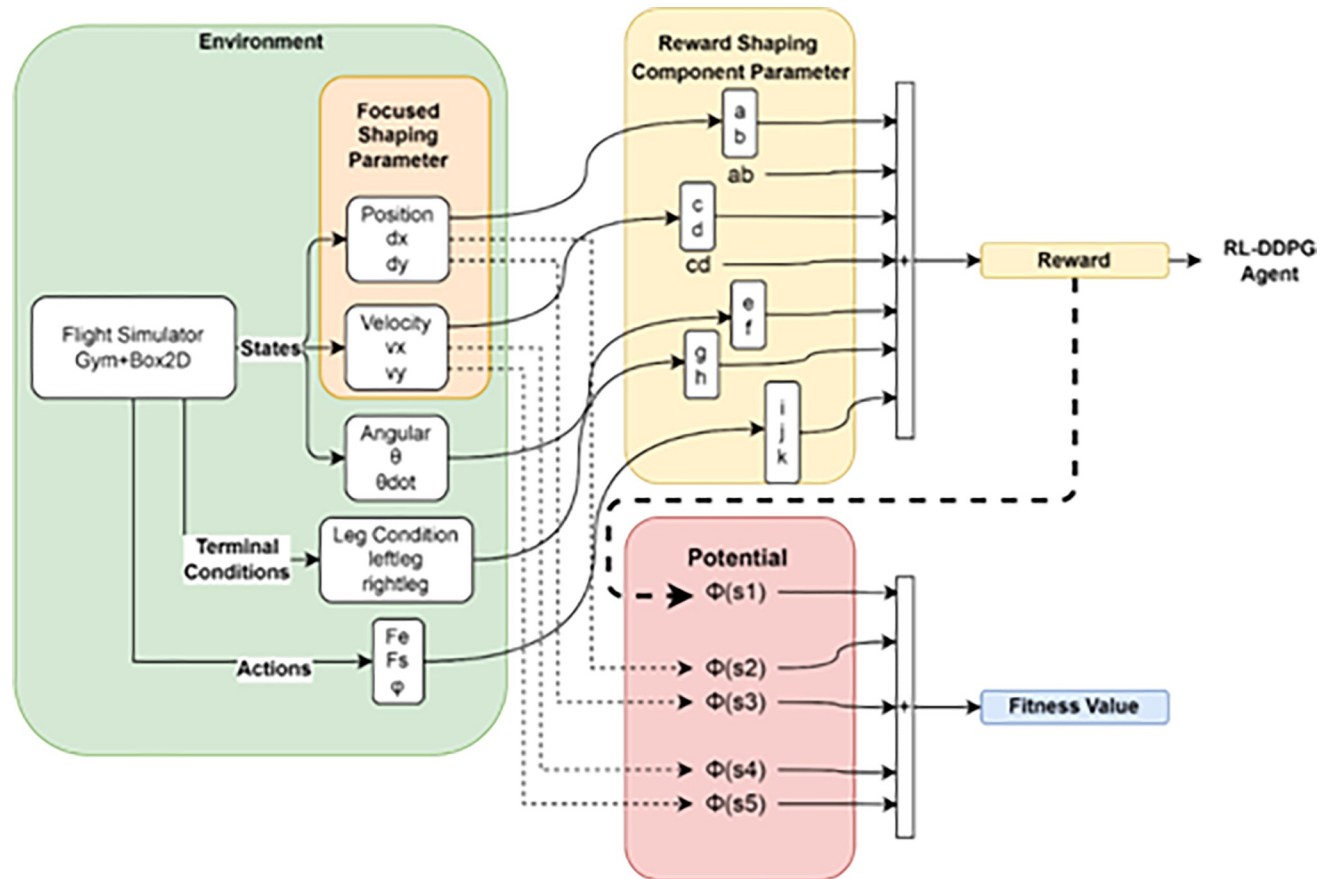

**Fig 9. Process of fitness value establishment from environment and reward shaping component parameters.**

Component Parameters in the yellow block, which must be optimized, are the components of the RSF and become the chromosomes of GA individuals. The output of the yellow block is used to calculate the fitness value in the red PbF block, which is used to assess the individuals produced in the orange and green box. The fitness value is calculated as the sum (blue box) of a few defined potential functions over the vehicle's terminal states, where terminal states meeting desired conditions are assigned a value of 1 for the potential functions and states that don't are assigned a value of 0.

Consequently, each individual's fitness is described as follows:

$$Fitness_{individual} = \begin{cases} \sum_{i=x,y,\dot{x},\dot{y},\theta} fit_i, & fit_y = 1 \\ 0, & fit_y = 0 \end{cases} \tag{27}$$

where $i$ is the component of reward shaping parameters. The desired terminal conditions are described below.

Each test's outcomes are assigned a fitness score. According to how well the rocket landed, the state is given points. Fitness state and total fitness are the two types of fitness that are used. The following conditions apply to fitness state.

$$\phi_{reward} = \begin{cases} 1, & reward > 0 \\ 0, & \text{otherwise} \end{cases}$$

$$\phi_x = \begin{cases} 1, & -3 \le x \le 3 \\ 0.5, & -5 < x < -3 \text{ or } 3 < x < 5 \\ 0, & \text{otherwise} \end{cases}$$

$$\phi_y = \begin{cases} 1, & y < 1 \\ 0, & \text{otherwise} \end{cases}$$

$$\phi\dot{x} = \begin{cases} 1, & -5 \le \dot{x} \le 5 \\ 0, & \text{otherwise} \end{cases}$$

$$\phi\dot{y} = \begin{cases} 1, & -7 \le \dot{y} \le 0 \\ 0, & \text{otherwise} \end{cases}$$

$$\phi_\theta = \begin{cases} 1, & -5 \le \theta \le 5 \\ 0, & \text{otherwise} \end{cases} \tag{28}$$

The perfect reward shaping function will be found once the complete fitness value of 5 is reached, which will happen if all potential functions $\Phi(s')$ for each individual (s') are equal to 1. Each state transition's reward function is calculated by adding the reward shaping parameters (s $\rightarrow$ s'). To produce accurate reward feedback, the agent must make a certain number of decisions, which is measured by computing time. When the desired terminal conditions are satisfied or the vehicle leaves the computing range, computation time is stopped.

Average fitness for multiple tests (multiple testing) is calculated by averaging the fitness of *N* tests.

$$Fitness_{generation} = \frac{\Sigma Fitness_{individual}}{N} \tag{29}$$

Each individual's performance is defined using their average fitness value, allowing for performance comparisons between them.

## 4.3 Genetic algorithm

The ability of a rocket booster to land on the ground is largely dependent on the amount of fuel already used during the majority of its mission to generate thrust. However, determining the precise magnitude, direction, and timing of the rocket engine's force at a specific point along its trajectory can be a time-consuming task to be optimized as an efficient trajectory. Numerous studies have shown that the genetic algorithm (GA) can effectively solve high-thrust impulsive spacecraft trajectory problems and low-thrust impulsive spacecraft trajectory problems [55–57].

This study sought to evaluate the effectiveness of the GA in predicting thrust profiles (Fe, Fs) and thrust vector angles (φ) in a landing trajectory scenario.

**4.3.1 Natural selection process.** As depicted in Table 3, each chromosome is made up of DDPG's reward shaping function component parameters that must be optimized in the order explained in Table 3.

A GA chromosome is a long chain of DNA (Deoxyribonucleic Acid)which encodes the coefficient of each shaping element in the form of a single place decimal number.

Table 3 shows that each chromosome in the genetic algorithm (GA) consists of component parameters of the Deep Deterministic Policy Gradient's (DDPG) reward shaping function that need to be optimized as outlined in Algorithm 1. The chromosome of GA represents a string of DNA which encodes the coefficient of each shaping element as a one decimal place number.

**Table 3. Initial values of the reward shaping function components as individual's chromosome.**

| Individual | *a* | *b* | *ab* | *c* | *d* | *cd* | *e* | *f* | *g* | *h* | *i* | *j* | *k* |
|---|---|---|---|---|---|---|---|---|---|---|---|---|---|
| | | | | | Reward Shaping Components (Chromosome) | | | | | | | | |
| 1 | 1.0 | 0.0 | 0.0 | 0.0 | 0.0 | 0.0 | 0.0 | 0.0 | 0.0 | 0.0 | 0.0 | 0.0 | 0.0 |

The six-state shaping elements (DNA) are arranged in order:

*a* = *x*-position error

*b* = *y*-position error

*ab* = *xy* position's general coefficient

*c* = *x*-velocity error

*d* = *y*-velocity error

*cd* = *xy* velocity's general coefficient

*e* = Angular position and

*f* = Angular velocity

Followed by two terminal trigger states DNA:

*g* = left leg contact

*h* = right leg contact

Added with three action shaping elements DNA:

*i* = main engine power

*j* = side engine power

*k* = thrust vector control

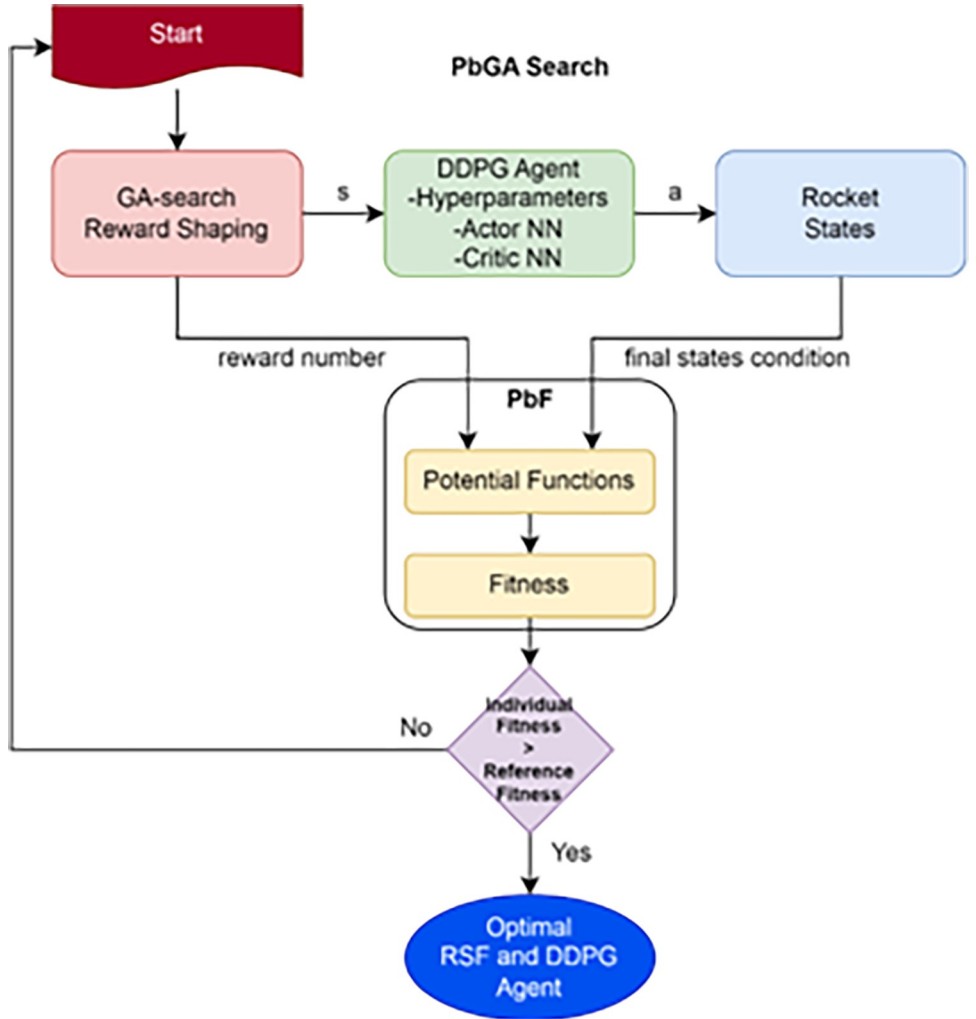

**Fig 10. Algorithm of the PbGA search for the optimal PbF.**

13 reward parameters altogether are taken into consideration for this GA study. Even though only 6 shaping elements (states highlighted in yellow above) and 3 next shaping elements (actions) are focused on and drilled by Potential-based GA methodology (Algorithm 1) to find the best individual of reward shaping function that performs better than the reference one, this is based on the mapping phase's results.

Each set of chromosomes is treated as an individual of the GA population, and the training process for each individual is based on the reward function shaping parameters that were found in the state error vector for that individual.

Each individual's fitness function is determined by adding up all of the potential functions they could have acquired after completing the training phase.

The optimization achieved by the current hyperparameter setting's DDPG agent in combination with the reward shaping function found through GA research is shown in Fig 10. This concludes the first contribution of the paper.

```
Algorithm 1. Potential-based GA (PbGA)-search for the fittest
individual
```

```
BEGIN
MAP effect of each component of reward shaping to flight performance;
INITIALISE population with hybrid random/mapped candidate solutions;
EVALUATE potential ofeach candidate;
    REPEAT  UNTIL  (TERMINATION  CONDITION  is  satisfied)
    DO
        1 SELECT parents;
        2 CROSSOVER pairs of parents;
        3 MUTATE the resulting offspring;
        4 EVALUATE new candidates;
        5 SELECT individuals for the next generation;
    END
    REPLICATE reference individual;
    COMPARE ELITE individuals TO reference individual;
    CONDUCT MONTECARLO to get the best individual
END
```

The training procedure for each individual, as shown in Fig 11, is described as follows:

1. The actor produces the continuous values of the main throttle, side engine throttle, and TVC swiveling angle as the action (a) in response to the agent's request for the environment's current state (s), where a = (s) + N is the action with noise added for the Ornstein-Uhlenbeck method of exploring the action space (Uhlenbeck & Ornstein, 1930).

2. OpenAI Gym returns the reward and sends the agent the subsequent state (s') (r).

3. An experience replay buffer is used to store the sampled data (s, a, r, and s') for later use.

4. To train the networks, as shown in section (2.2), a minibatch of size N is selected at random from experience replay.

5. To improve each individual's fitness function, the step reward accumulates over the course of the training episodes. The best individual with the best fitness score is updated every generation if a better chromosome combination is discovered, as shown in section (4.3.1).

This is the paper's second contribution in this section: In order to find the optimal combination of RSF coefficients and the most suitable function's term formula that maximizes task performance, maximizes fitness value, while minimizing the number of population size and

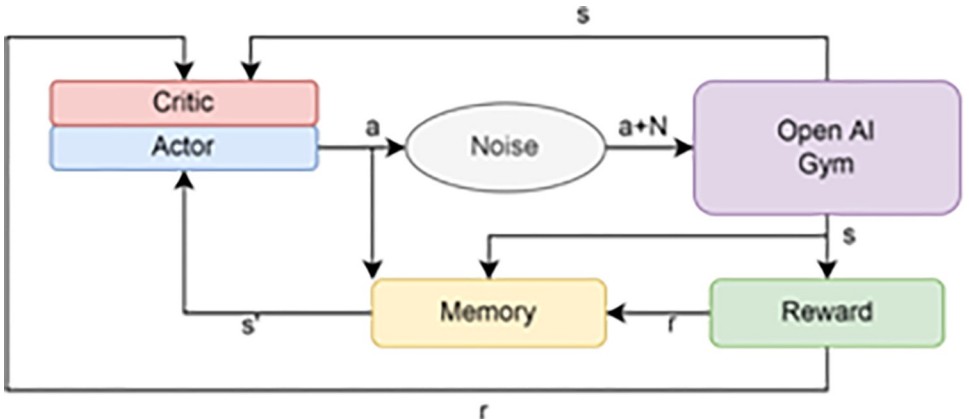

**Fig 11. Training process of a DDPG individual.**

**Table 4. Genetic algorithm meta-parameters.**

| GA Hyperparameters | Initial Amount | Final Amount |
| --- | --- | --- |
| Population size | 50 | 74 |
| Generation | 10 | 15 |
| Mapping Probability | 100% | 27% |
| Crossover Probability | 0% | 24% |
| Mutation Probability | 0% | 31% |
| Replication Probability | 0% | 1% |

generation, the genetic algorithm thoroughly searches the space of reward shaping function used in DDPG using PbGA. Setting up the GA with the aforementioned meta-parameters:

Table 4 displays the evolution of the GA meta-parameters from their initial value to their final value. The reward shaping function term's parameters, which have a range of 0 to 1, represent the DNA of each chromosome. According to the findings of the GA mapping phase tests, altering the reward shaping parameters' values had no discernible linear effect on how agents learned. Therefore, a simple hill climber may have trouble locating the ideal parameters. GA is used to optimize these parameter values because it was developed specifically for problems with such a lack of knowledge.

In addition to reward, fitness, landing position and velocity errors, and successful landing percentage, controllers were also assessed.

**4.3.2 Reward shaping components parameterization as chromosome DNA.** Reward function represents the change of reward shaping function with respect to time.

$$reward = (current\ reward\ shaping) - (previous\ reward\ shaping) \qquad (30)$$

The reward shaping function measures the distance between the current position of the rocket and the target position. The shaping components consider factors such as translational velocity, angular states, conditions of the touching leg, and power throttles.

$$
\begin{aligned}
rewardshaping \\
= ab\sqrt{ax^2 + by^2} + cd\sqrt{c\dot{x}^2 + d\dot{y}^2} + e\theta + f\dot{\theta} + g.leg_{left} + h.leg_{right} + i.MainEnginePower \\
+ j.SideEnginePower + k.TVC\_Angle \qquad (31)
\end{aligned}
$$

The impact of each action (main engine power, side engine power, and thrust vector angle) on the flight maneuver was analyzed to understand the weight magnitude of the shaping components in the reward shaping function. This analysis was done by mapping the controller's steps in successfully landing the rocket. The mapping provides an objective view of the effectiveness of the controller's actions during the landing phase, using the rocket's states as feedback throughout each test. For example, if a controller yields a rocket with an average pitch angle greater than 8 degrees, it suggests excessive use of gas thrusters, which could result in a landing on one leg. Fig 12 illustrates the difference between a non-contact landing leg (rocket is in the air) and a leg in contact (rocket's leg touching the landing pad).

Both states and actions, indefault, arepenalized by the reward function. Positive rewards were given for actions that gave "good progress," such as the correcting motion of rocket strafing maneuver produced by the side rocket force, because it is a crucial component of the RL mechanism.

**4.3.3 DNA characteristics of the chromosomes.** The smallest representation of the component parameter in this study is DNA, which carries a single genetic code from a whole chromosome or the entire reward shaping function. Given that each DNA molecule can be

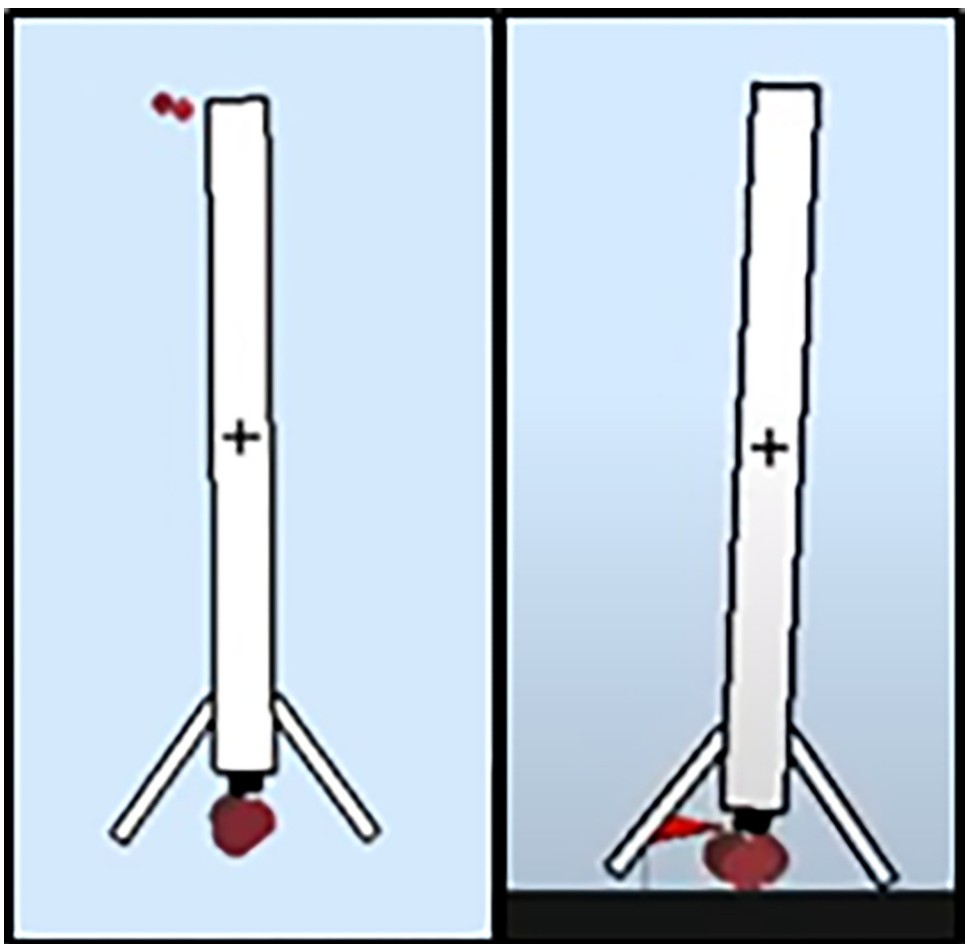

**Fig 12. Leg condition of the landing rocket.** Still in the air (left), one leg touches pad (right).

represented by a single place decimal number or by two digits for each parameter value, it requires 12 digits for the six parameters.

Because significant increases in success rate require large steps of parameter value changes, only one decimal place is chosen to represent each parameter value. We use a scanning method that mimics the gain search in Bode plot, which is based on logarithmic scales, to map and search the appropriate coefficient.

The value parameter for the x-position error is thought to be more representative than 0.9, 1.0, or 1.1, so a choice of three coefficients between 0.5, 1, and 2 (multiplication of 2) can be made (multiplication of 0.1).

A thorough 36-grid-size space search is not feasible due to the length of each fitness evaluation, so a PbGA search is done instead.

Prior to the main search, a mapping phase is carried out to determine how the flight performances are influenced by the shaping components. As a result, chromosome DNAs are organized step by step, and the primitive individuals are tested one at a time. Using pair of 1-point crossover or multiple 2-point crossover, few parents created from the mapping results are re-crossover-ed to generate new offsprings of the next generation. (Fig 13). Multiple parents's crossover is utilized here to increase the search speed, since each offspring generated will conduct a 5 times training processes that really demanding.

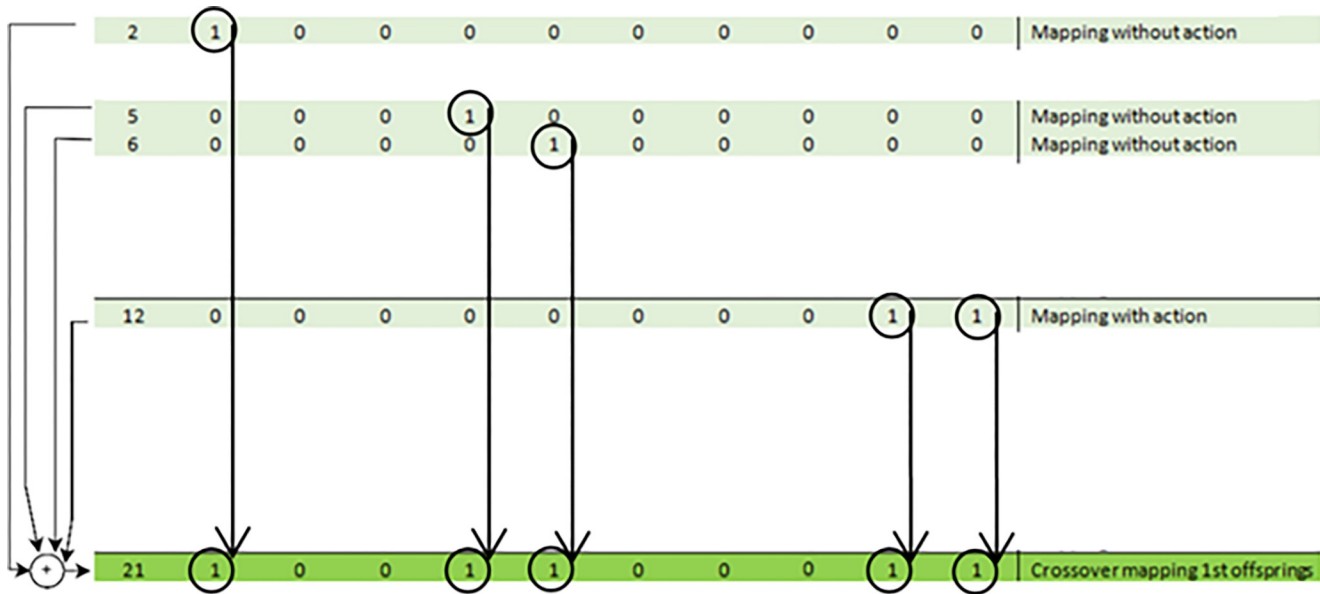

**Fig 13. Sample of crossover offspring generation (#21) from mapping generations (#2, #5, #6, #12) as parents.**

The generation of new offspring forms is then made possible by these chromosome strings, space-independent crossovers, and mutations of coefficient (Fig 14) or formula string operators. The potential of each reward shaping component added to the potential of the reward divided by the number of elements yields the fitness for each DNA chromosome (set of parameter values).

The minimization of shaping rewards is transformed into a maximization fitness problem by the proposed fitness evaluation, known as PbF or Potential based Fitness. A total space with a population of 70 is created in 15 generations from the integration and optimization of DDPG agent + GA-searched Reward Shapingfunction. The fittest candidate from the elitism generation is chosen using a ranking selection from a combination of two parameters. The remainder of the offspring are produced by shape mutation with a multiplier of mutation of 2, with uniform crossover producing a small portion (Fig 14) of the total DNA.

## 5 Experimental results

This section presents the experimental analysis, in which a few DDPG neural network models, including Ferrante's reference model, are subjected to the optimized reward shaping function results.

### 5.1 Reward shaping functions determining process

To determine the reward shaping function, the first step is mapping the relationship between states and actions using a reward shaping function, as shown in Fig 15 and described in

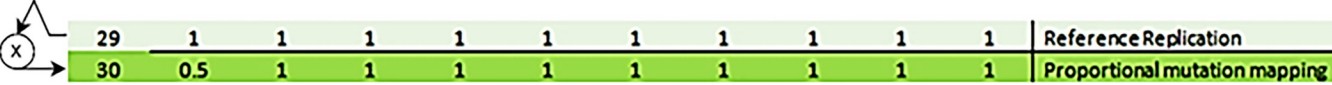

**Fig 14. Sample of mutation offspring generation (#30) from replication generation (#29) as parent.**

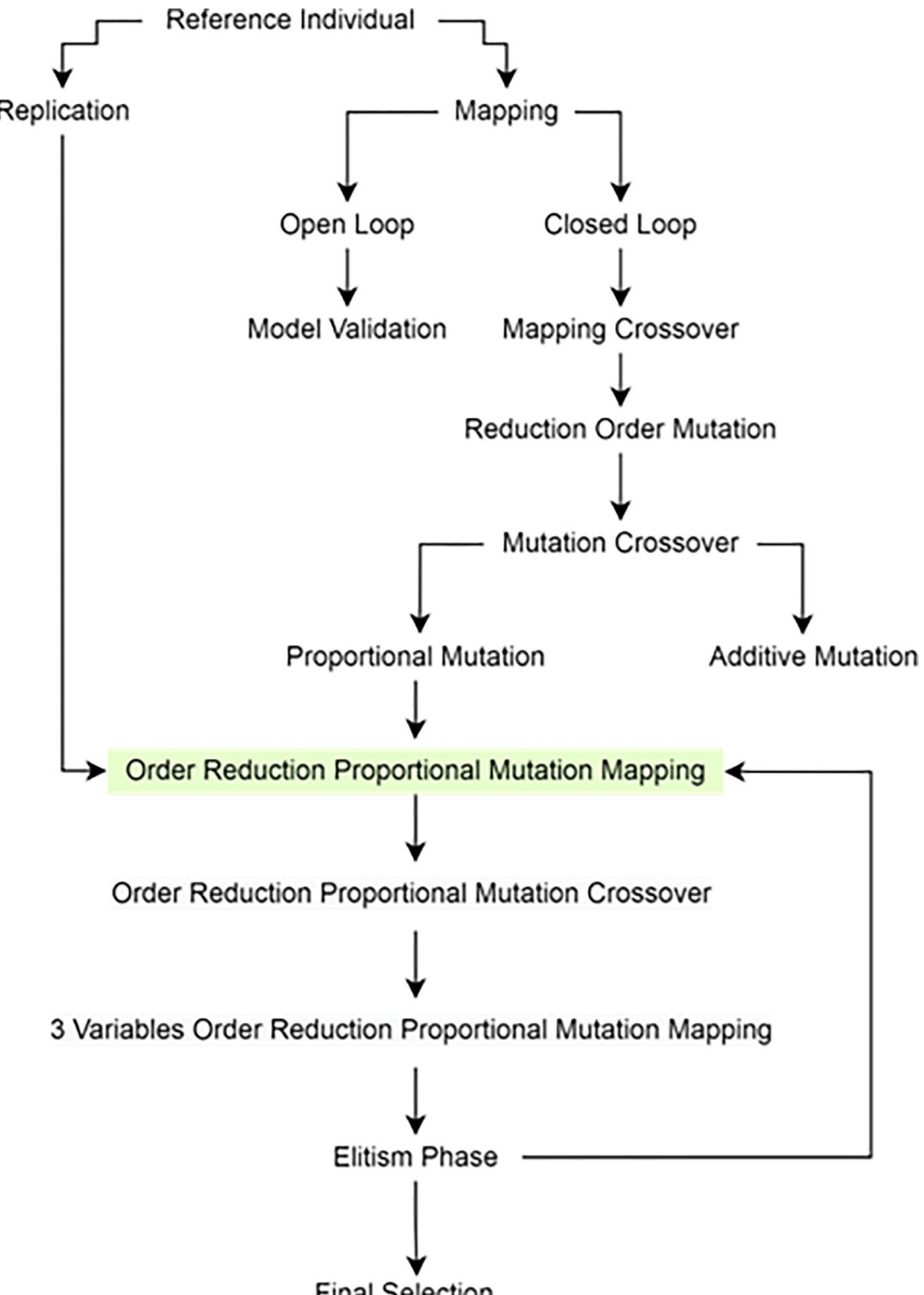

**Fig 15. PbGA search phases to find the best individuals.** Conducted in 15 Generations.

Section 4.3.1. The mapped correlations will be further analyzed to understand the impact of each chromosome element on flight performance."

GA search is a loop of generation producing new offsprings using the genetic operators, in our case the loop is conducted from the last phase before final selection that is elitism phase, to the phase where the replication generation is utilized, i.e., Order Reduction Proportional Mutation Mapping. In other word, the replication generation plays a significant role in our search. This loop from Fig 15 share the same activity with the loop in Fig 10. The loop from Fig 10 shows the reviewing process of each offspring, which is extensively used in the loop of Fig 15.

## 5.2 GA-based search simulations

### 5.2.1 Open-loop simulation: Model validation.
The rocket model is validated using open loop simulations with step-by-step inputs for each action. Three inputs—FE = 0.01 or 1% throttle, FS = 1 or 100% or maximum left side thruster, and FS = 1 for maximum right side thruster—are activated in three separate simulations. From the initial state condition until the maximum allowed states condition is reached, all thrusters are lit. For the zero-action condition, a fourth simulation is added.

Fig 16 depicts a simulation of a free fall in which linear falling velocity is maintained in accordance with Newton's second law (Hall, 2022).When input FE = 1% is applied to the main thruster, the rocket drastically reduces its falling speed, modifies its falling motion, and continues to delay the landing due to the upward force, but is then followed by a rightward drift (x(t) = 3.05m) as a result of a small deflection angle (0.00146 rad) that occurred at the beginning of the falling motion.

Input FS = 100% causes the final value of x(t) to be -0.94 m, whereas FS = = 100% causes x(t) to be 0.94 m, indicating that the rocket is pushed to the right when the left booster is on,

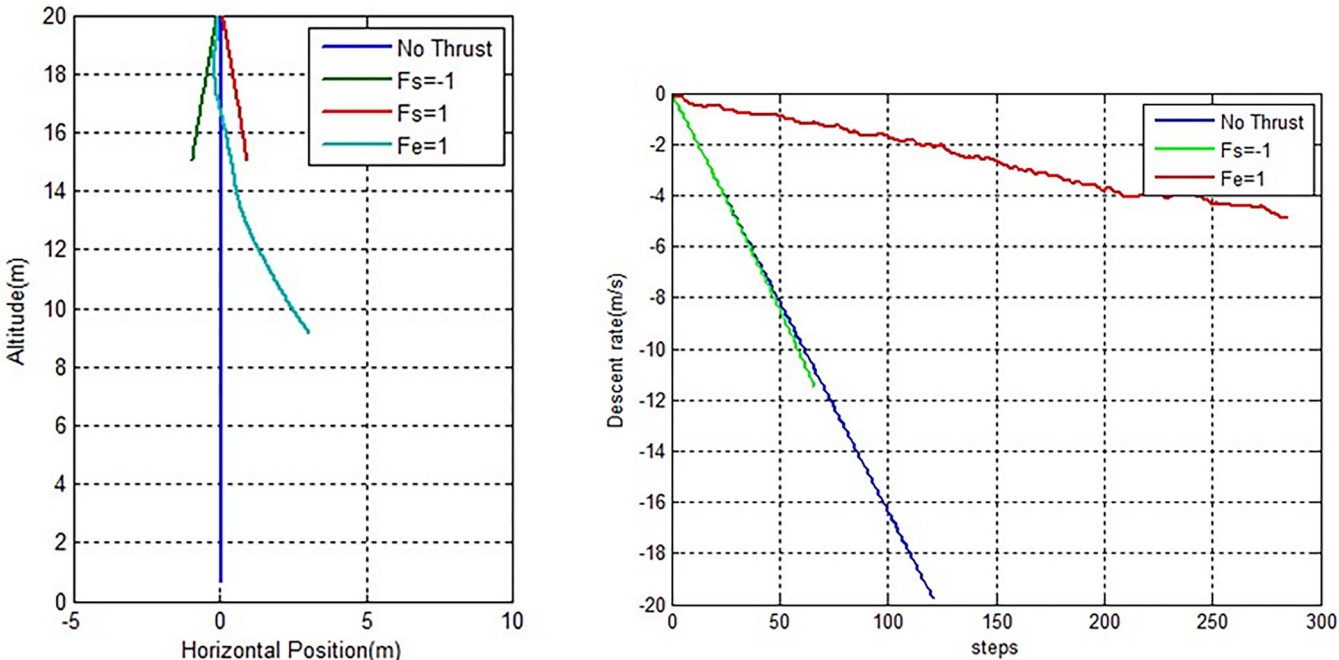

**Fig 16. Open loop simulation.** Trajectory(left) and descentrate (right).

**Table 5. Individual #1 - #11 chromosome mapping.** This table describes the initial numbers put in the components used by the RSF. Each RSF structure become a chromosome of an individual.

| Individual | a | b | c | d | e | f | g | h | i | j | k |
|---|---|---|---|---|---|---|---|---|---|---|---|
| 1 | 0.0 | 0.0 | 0.0 | 0.0 | 0.0 | 0.0 | 0.0 | 0.0 | 0.0 | 0.0 | 0.0 |
| 2 | 1.0 | 0.0 | 0.0 | 0.0 | 0.0 | 0.0 | 0.0 | 0.0 | 0.0 | 0.0 | 0.0 |
| 3 | 0.0 | 1.0 | 0.0 | 0.0 | 0.0 | 0.0 | 0.0 | 0.0 | 0.0 | 0.0 | 0.0 |
| 4 | 0.0 | 0.0 | 1.0 | 0.0 | 0.0 | 0.0 | 0.0 | 0.0 | 0.0 | 0.0 | 0.0 |
| 5 | 0.0 | 0.0 | 0.0 | 1.0 | 0.0 | 0.0 | 0.0 | 0.0 | 0.0 | 0.0 | 0.0 |
| 6 | 0.0 | 0.0 | 0.0 | 0.0 | 1.0 | 0.0 | 0.0 | 0.0 | 0.0 | 0.0 | 0.0 |
| 7 | 0.0 | 0.0 | 0.0 | 0.0 | 0.0 | 1.0 | 0.0 | 0.0 | 0.0 | 0.0 | 0.0 |
| 8 | 0.0 | 0.0 | 0.0 | 0.0 | 0.0 | 0.0 | 1.0 | 0.0 | 0.0 | 0.0 | 0.0 |
| 10 | 0.0 | 0.0 | 0.0 | 0.0 | 0.0 | 0.0 | 0.0 | 1.0 | 0.0 | 0.0 | 0.0 |
| 11 | 0.0 | 0.0 | 0.0 | 0.0 | 0.0 | 0.0 | 0.0 | 0.0 | 1.0 | 0.0 | 0.0 |

and to the left when the right booster is on. This implies that Newton's third law of motion is being followed by the rocket's motion (Hall, 2022). The final landing speed for the free fall, $vy$ $(t)$ = 19.7835 m/s, shows that the vehicle touches down too soon. A rocket landing control system is necessary, with terminal velocity being one of the most crucial criteria. High landing speed could seriously harm or even destroy the rocket.

**5.2.2 Closed-loop simulation: Mapping.** When using the Genetic Algorithm (GA) approach to find the reward function, the first step is the mapping phase. In this step, the influence of the reward function's elements on the state and action is mapped by activating the reward function's state components individually and turning the two action components on and off. At this point, the Euclidean weight is set to 1.

The values of FS for individual #2 in the mapping of the x-position state (individual #2) in Table 5 are quite large, similar to those of individual #7 in Table 6. This indicates that the rocket is trying to balance itself by stabilizing the x-position command and pitch angle command in order to land in the middle of the landing pad. The value of FE is quite low when evaluating the y-position state (individual #3), as shown in Fig 17, indicating that the rocket is trying to land on the pad as quickly as possible. The controllers of individual #3 were trying to land cautiously, but were unable to do so, causing the rocket to crash instead. According to Ferrante (2017), FS and play dominant roles in horizontal movement, but not in vertical movement. The mapping of the y-position state reveals an anomaly that still cannot be explained and requires further investigation, as it contradicts Ferrante's equation stating that FS and play have a dominant role in horizontal movement, but not in vertical movement.

When examining the x-horizontal velocity state (individual #4) as shown in Fig 18, it appears that the FS actions are in the ideal proportion, not too small or too large, able to

**Table 6. Percentage of throttle actions in the mapping phase.** This table shows how much the action commands are used by the corresponding mapped individuals in the landing scenario.

| Individual | State Mapped | $F_E$ | $F_S$ | $\Phi$ |
|---|---|---|---|---|
| 2 | $x$ | 31% | 94% | 91% |
| 3 | $y$ | 5% | 100% | 97% |
| 4 | $\dot{x}$ | 0% | 71% | 95% |
| 5 | $\dot{y}$ | 81% | 86% | 60% |
| 6 | $\theta$ | 88% | 63% | 77% |
| 7 | $\dot{\theta}$ | 21% | 96% | 90% |

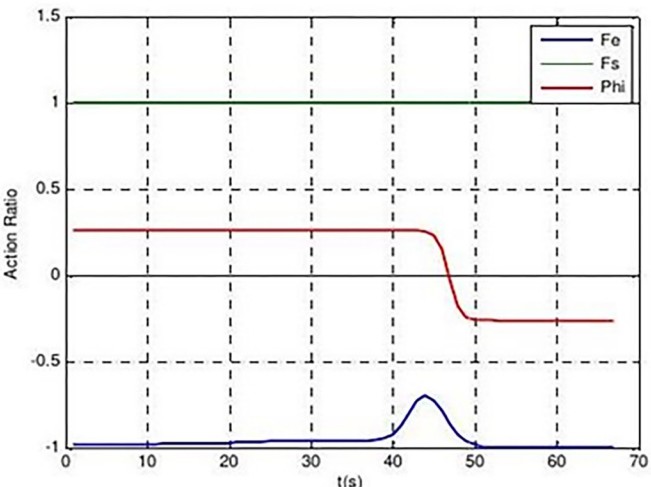
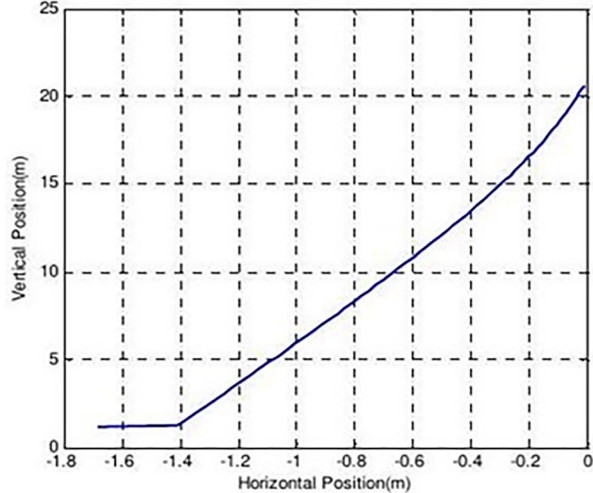

**Fig 17.** Individual #3's action ratio (left) and position states (right).

prevent the rocket from deviating too far (0.35m) while maintaining a reasonable horizontal velocity (0.43 m/s). However, this individual was unable to prevent the rocket from exploding due to the high rate of descent (23.77 m/s) caused by the lack of weight in the vertical part of the reward shaping function elements.

The percentage of firing up each action against the state is displayed in Table 6 above so that you can see how dependent each mapped state is on the controlling actions. A good candidate is found in Fig 19 when considering the vertical velocity state only (individual #5); all positions and velocities terminate in good condition, with the exception of one final angular position or leaning angle (Fig 19 right), which becomes 35 degrees (leaning too much). As a result, this individual has a fitness rating of 4, which is a decent score.

With the exception of individuals #5 and #17, mapping these initial, basic individuals yields fitness results ranging from zero to just 2.5. (Fig 20). The highest number of fitness 5 meeting the requirements for an elite generation is produced by individual #17.

**5.2.3 GA-based search process.**   Following the mapping phase, the crossover phase involves 'crossbreeding' individuals from the mapping phase to create the first generation of offspring. Evidently, no one has a good landing performance at this stage because the majority of fitness scores are below 5. The reward function is then replicated in the replication phase, during which all of its components are given values of 1. The individual numbered 29 will be a member of the elite generation as a result of this phase's landing with a fitness value of 5.

The mapping of mutations is the following stage. There are two different kinds of mutations: order reduction mutations, which change the structure of the reward function formula, and proportional mutations, which are based on coefficient multiplication. To determine how multiplication and division affect each element of the reward function and the landing results, the next phase entails performing proportional mutation mapping beforehand. Only individual #36, who had a fitness value of 5, performed a good landing among individuals #30 to #41. The following stage, known as proportional mutation crossover, entails combining chromosomes from the proportional mutation mapping phase in an effort to produce the best progeny. The only individual to emerge with fitness equal to or greater than 4.5 is individual #42; the crossover offspring are apparently insufficient to generate individuals with decent performance levels.

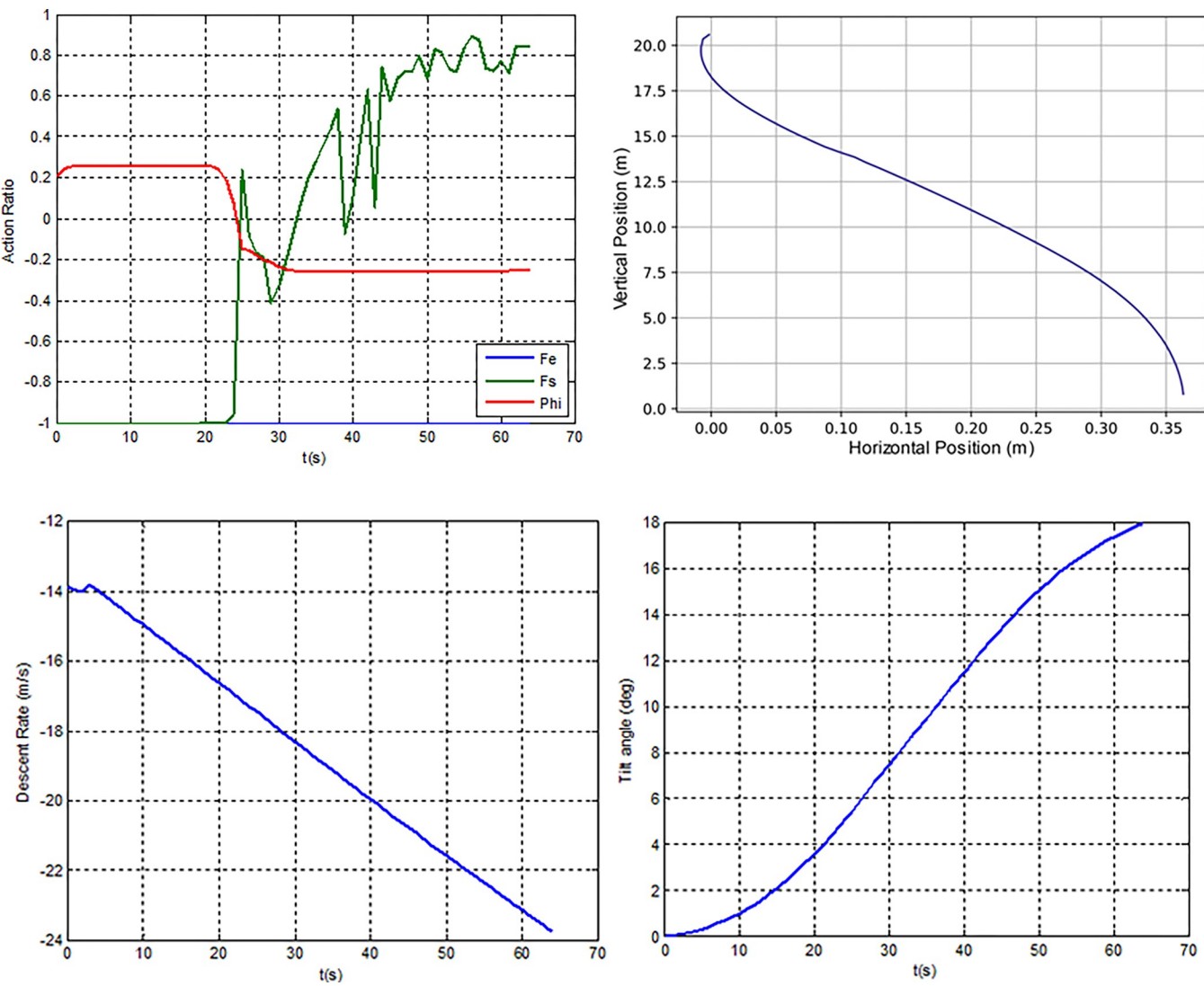

**Fig 18.** Individual #4's (a) action ratio, (b) position states, (c) descent rate and (d) tilting angle.

As part of the PbGA Successive Offspring Generation, in the phase of "Order-reduction Mutation with Two Variables", the quadratic form is transformed into an absolute form. At this point, individual #52 had the highest fitness score of 5, but there were also several other individuals with a score of 4, including individuals #50, 57, and 58.

The top performers from the mapping phase were then combined with the top performers from the order reduction mutation phase in a process called the crossover mapping-order reduction phase. However, currently none of the individuals are producing satisfactory results. In the additive mutation phase, the chromosomes of the above individuals are combined to create new ones. These chromosomes can have unlimited value in unlimited addition but are limited to a value of one in limited addition. There are currently not enough high-performing individuals.

The final phase, called Euclidean mutation, involves multiplying the position element's Euclidean component. Individual #71 had the best results at this stage with a fitness value of 5.

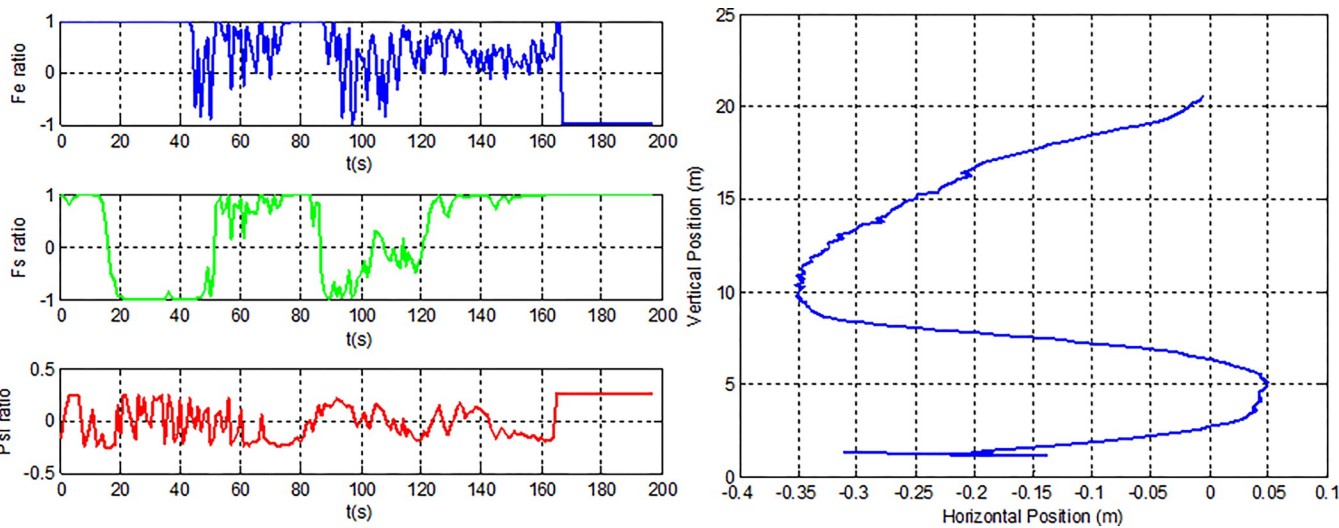

**Fig 19. Individual #5's action ratio, position states.**

A summary of the average fitness for each generation can be made based on the phase or generation. Table 7 shows that the replication, order reduction mutation, closed-loop mapping, and proportionate mutation mapping phases are the most important for producing individuals with successful landings. The crossover and addition mutation phases did not result in a successful landing.

## 5.3 Powered descent guidance comparison

Since it is based on individuals who have fitness values greater than 4, this last stage is also known as the "elitism generation." Each participant underwent a Monte Carlo test, which consisted of 10 repeated landing tests, to assess the consistency of landing performance. The fitness is then determined for each landing and averaged over the course of ten trials. The average fitness of the Monte Carlo test is shown in Table 7 below.

Most of the individuals in Table 8 had a lower average fitness score in the elitism phase than they did in previous tests, due to the inconsistent landing performance observed in early Monte Carlo tests (10 tests per individual, or a total of 100 tests). In these tests, the majority of the rockets did not successfully land.

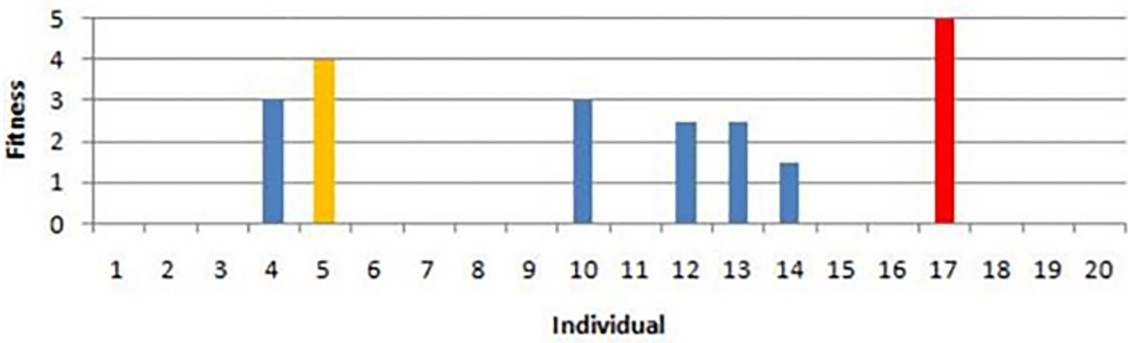

**Fig 20. Best Individuals from the mapping phase.**

**Table 7. Average fitness for each phase or generation under the PbGA successive offspring generation.**

| Individual | Generation | Generation's Average Fitness ($fit_{avg}$) |
|---|---|---|
| 1 − 11 | Open-loop Mapping | 0.55 |
| 12 − 20 | Closed-loop Mapping | 1.27 |
| 21 − 28 | Crossover mapping Parents | 0.375 |
| 29 | Replication | 5 |
| 30 − 41 | Proportional Mutation Mapping | 1.25 |
| 42 − 48 | Proportional Mutation Crossover | 0 |
| 49 − 56 | Order-reduction Mutation with Two Variables | 1.375 |
| 57 − 58 | Order-reduction Mutation with Three Variables | 4 |
| 59 − 61 | Order-reduction Crossover Mapping | 0.7 |
| 62 − 69 | Additive Mutation | 0 |
| 70 − 74 | Euclidean Mutation | 1 |

Individuals who land successfully less than 30% of the time are likely to have a fitness score between 3 and 4. To further test this, the top five performers from the elitism phase were selected. The average fitness of these five individuals is shown in Table 9, along with the fitness of the PID and DDPG reference individuals.

The review of the reward functions for the top five individuals shows that both proportional mutations and order reduction mutations play important roles in finding the reward function. However, individual 17 from the mapping generation had the best and most consistent landings. From a genetic perspective, this individual focused on improving angle control and propulsion efficiency, which suggests that angle control is a critical factor in a descent-powered landing problem. Many of the landing failures among the individuals were caused by the rocket's tilt angle becoming too large, resulting in an environment reset.

The graphs below (Fig 22) compare the trajectory, vertical speed, and tilt angle of the rocket during landing. The PID trajectory (in blue) tends to be linear at first before deviating towards the end of the landing phase. Despite this, the controller was still able to move the rocket back and land close to the target position. Table 10 shows the RSF components for the top five individuals.

The experiment compares the performance of three different controllers (PID, reference DDPG, and GA-based Individual 17 DDPG) in controlling the landing of a rocket. It shows that the reference DDPG controller produced a trajectory that deviated significantly from the starting point, resulting in a landing further from the target than the other two controllers.

**Table 8. Monte carlo test ranking from elitism phase.**

| Individual | $fit_{avg}$ | Ranking |
|---|---|---|
| 17 | 4.6 | 1 |
| 29 | 3.25 | 2 |
| 36 | 3.2 | 3 |
| 57 | 3.1 | 4 |
| 52 | 3.05 | 5 |
| 50 | 2.9 | 6 |
| 58 | 2.2 | 7 |
| 42 | 2.1 | 8 |
| 30 | 1.15 | 9 |
| 71 | 0.9 | 10 |

**Table 9. Comparison of elite generation to PID and reference individuals.**

| Individual | fit*avg* |
|---|---|
| PID | 4.95 |
| DDPG | 4.6 |
| 17 | 4.6 |
| 29 | 3.25 |
| 36 | 3.2 |
| 57 | 3.1 |
| 52 | 3.05 |

However, it also consumed less fuel and landed quicker than the PID controller. It was determined that the GA-based Individual 17 was the optimal controller, as it had the best balance of landing accuracy, time, and fuel efficiency. It is also proven that the design of RL controllers does not require prior knowledge of the system model, making them relatively easy to use for those without an in-depth understanding of the system.

The following Fig 21 and Table 11 display the final states data population and $fit_{avg}$ from each 100 tests (MC-100). This Monte Carlo test is also applied to PID, GA-DDPG and reference DDPG controllers, which don't focus in reward or fitness valueafterward, but purely on performance benchmarking in terms of state. As stated in MC-10, the MC-100 also underlines the effectivity of PID in miss distance, landing speed, landing time, and remaining fuel.

In Fig 22, from the start, the DDPG controller's trajectory deviated significantly. At a height of 5 meters, it tried to compensate in the opposite direction, but the landing ended up passing the midpoint. Individual 17 also deviated, though not as far as reference does, and landed closer to the landing point than the PID controller. In terms of landing speed, GA-DDPG, reference DDPG, and the PID controller were the most successful, in that order.

The tilt angle of the PID-guided rocket is relatively stable with a maximum tilt of 2.5 degrees, while the reference DDPG-controlled rocket is more tilted at 2 degrees and Individual 17 landed with a slope of 3 degrees, despite attempting to return to an upright position. All three rockets' inclinations were within the criteria for a safe landing angle as stated in Section 4.3.2.

In terms of fuel consumption, the PID-guided rockets leaving less than 3 tons of fuel remaining while the reference DDPG and GA-based Individual 17 control rockets were more fuel-efficient, leaving around 4 tons of fuel, which is closely related to the shorter time of the landing process. The PID-guided rockets took more than 300 seconds to land while the reference DDPG and GA-based Individual 17 took around 130 seconds.

The best values were taken from the characteristics of miss distance (horizontal distance from the target point on the landing pad), landing time, and fuel efficiency. Table 11 shows

**Table 10. RSF Components of the best five individuals.**

| Individual | a | b | c | d | e | f | g | h | i | j | k |
|---|---|---|---|---|---|---|---|---|---|---|---|
| 17 | 0.0 | 0.0 | 0.0 | 0.0 | 1.0 | 0.0 | 0.0 | 0.0 | 1.0 | 1.0 | 1.0 |
| 29 | 1.0 | 1.0 | 1.0 | 1.0 | 1.0 | 1.0 | 1.0 | 1.0 | 1.0 | 1.0 | 1.0 |
| 36 | 2.0 | 1.0 | 1.0 | 1.0 | 1.0 | 1.0 | 1.0 | 1.0 | 1.0 | 1.0 | 1.0 |
| 57 | 0.05* | 1.0 | 0.1* | 0.2* | 1.0 | 1.0 | 1.0 | 1.0 | 1.0 | 1.0 | 1.0 |
| 52 | 0.01* | 1.0 | 0.1* | 1 | 1.0 | 1.0 | 1.0 | 1.0 | 1.0 | 1.0 | 1.0 |

*: Absolute operations

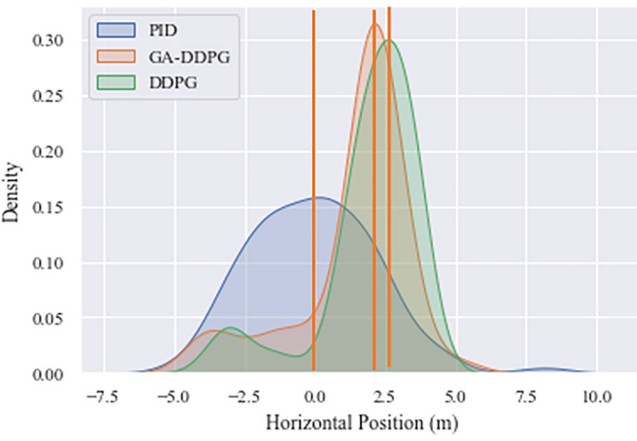 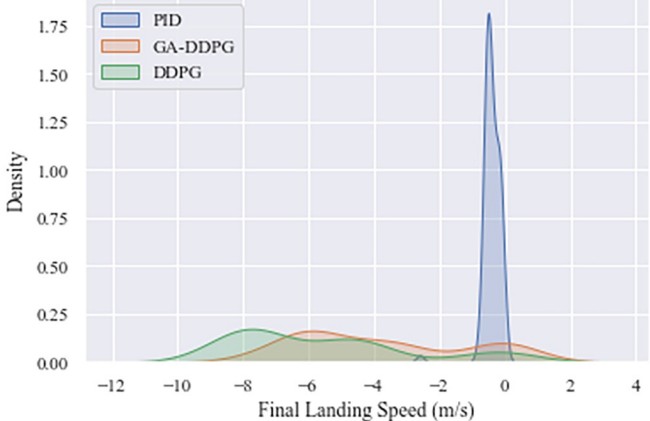

**Fig 21. Landing performance probability algorithm under MC-100 testings.**

that the PID controller excels in terms of miss distance, with the best landing less than 10 centimeters from the target point, followed by DDPG and GA-DDPG controllers. However, PID-controlled landings take longer and require more fuel than DDPG. It can be concluded that the GA-based Individual #17 is the optimal controller.

It is noteworthy that prior knowledge of the system model is not required to design RL controllers, making the controller design process relatively easy and accessible to those without in-depth understanding of the system. As stated in Section 4.2, the MIMO system in this case had to be simplified to a SISO system to facilitate the design of the PID controller, while the DDPG controller did not need to.

Reinforcement learning is not a magic solution to attain optimal control performance. The test results show that DDPG control performs well, but not as well as PID. However, the use of a reward function in reinforcement learning allows for greater control over the design process. In this case, the reward shaping function can be used to teach the agent to use fewer resources such as fuel while the PID controller is limited to attaining the set point of the desired state value.

To check the system's stability Lyapunov stability test is conducted using Lyapunov function in quadratic form. From the Fig 23 below, it is shown that in an unobstructed landing, all algorithm comply to the PD-condition for the function and ND-condition for its derivative function. The reward number also shows that all algorithm have positive trend and the PID produce higher number than the RL controllers, but has a longer flight time needed to land the vehicle.

A comparison on how the three main controllers work can be seen from the action progression of the rocket. As seen in Fig 24 below, the rocket using the PID controller (red) is powered by the FE from the first step until the height of 3 meters, where the controller then begins to turn on the FE with a magnitude that slowly diminishes. The rocket with the DDPG controller

**Table 11. MC Benchmarking of the fittest GA individual to PID and Reference DDPG.**

| Controller | Miss distance (m) | Landing Speed (m/s) | x Higher | Landing time(s) | % Faster | Remaining fuel (tons) | % Better |
|---|---|---|---|---|---|---|---|
| GA-DDPG | 1.22 | -2.5 | 2x | 128 | 60% | 4.09 | 30% |
| DDPG | 1.56 | -5.56 | 4.6 x | 129 | 58% | 4.13 | 39% |
| PID | 0.09 | -1.2 | 0% | 313 | 0% | 2.97 | 0% |

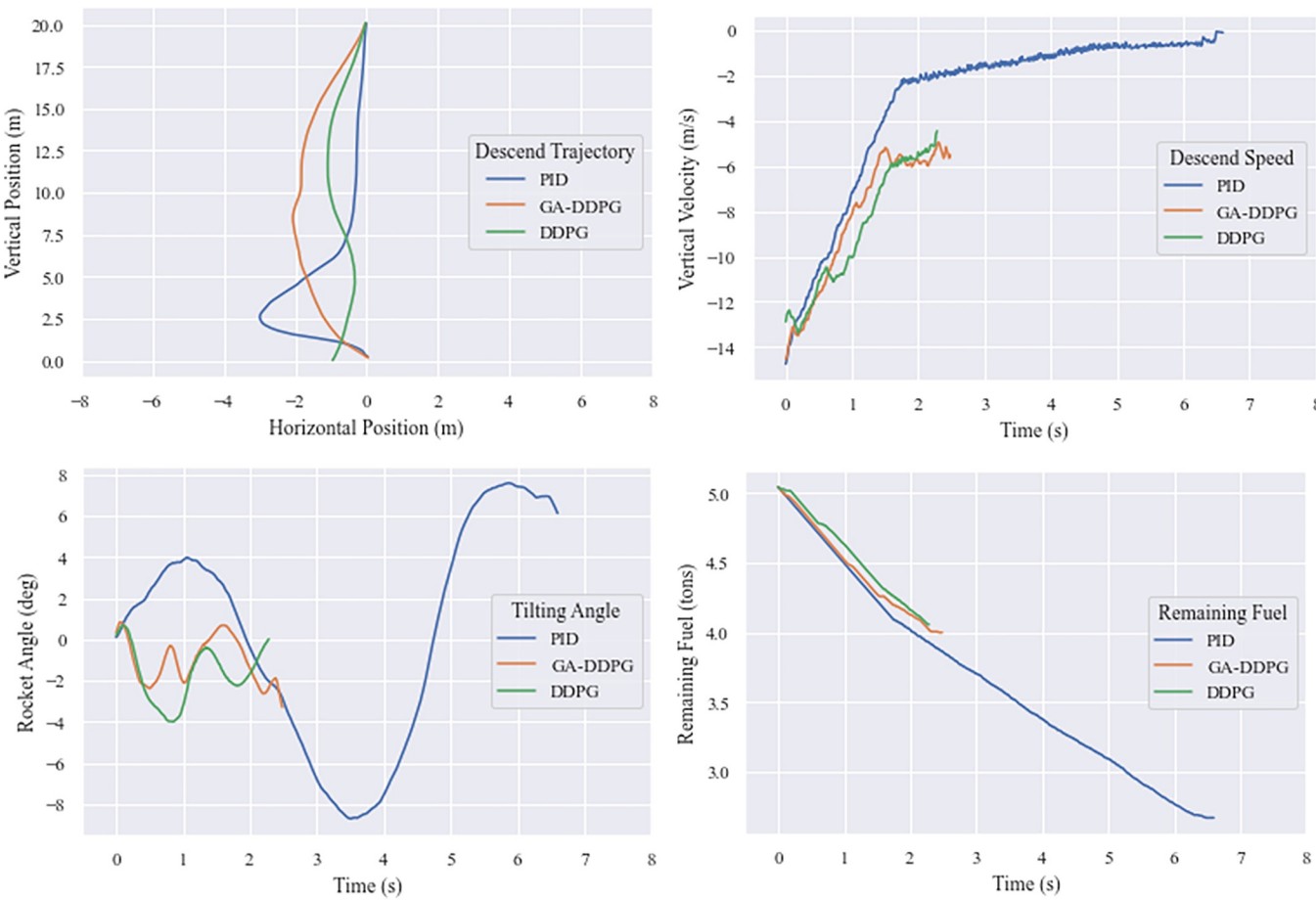

**Fig 22. Landing performance comparison of the fittest individual GA-DDPG against PID and reference DDPG model.** (a) Horizontal position against vertical position, (b) vertical velocity with respect to time, (c) rocket tilt with respect to time, and (d) fuel over time.

(green), on the other hand, begins with the FE off, allowing it to descend to a range of 18 meters before turning the FE on. The referenceDDPG (green) then keeps the FE on from 17 m to 5 m, whereas the Individual #17's action (blue) is more dynamic. This #17's dynamic behavior is most likely due to its reward function, which only accounts for FE and FS as actions, and theta or tilt angle as the input state, causing the controller to focus more on translating the theta angle into those two action aspects. It can also be seen that the PID controller does not hesitate to produce FE = 100% continuously, while the DDPG controller (reference or #17) does. This is likely attributed to the implementation of a reward function that penalizes the usage of boosters, thereby leading to the DDPG-controlled rocket firing less frequently.

### 5.4 Cross validation in RL

Cross-validation is a statistical method used to evaluate and validate learning algorithms in RL, which is a stochastic approach for constructing control algorithms [58]. It involves dividing the data into two segments, one used to train the model and the other used to test the model. The purpose of cross-validation is to detect overfitting, meaning the model fails to generalize patterns. In typical cross-validation, the training and validation sets rotate in successive rounds so that each data point can be validated against. Cross-validation in RL is different

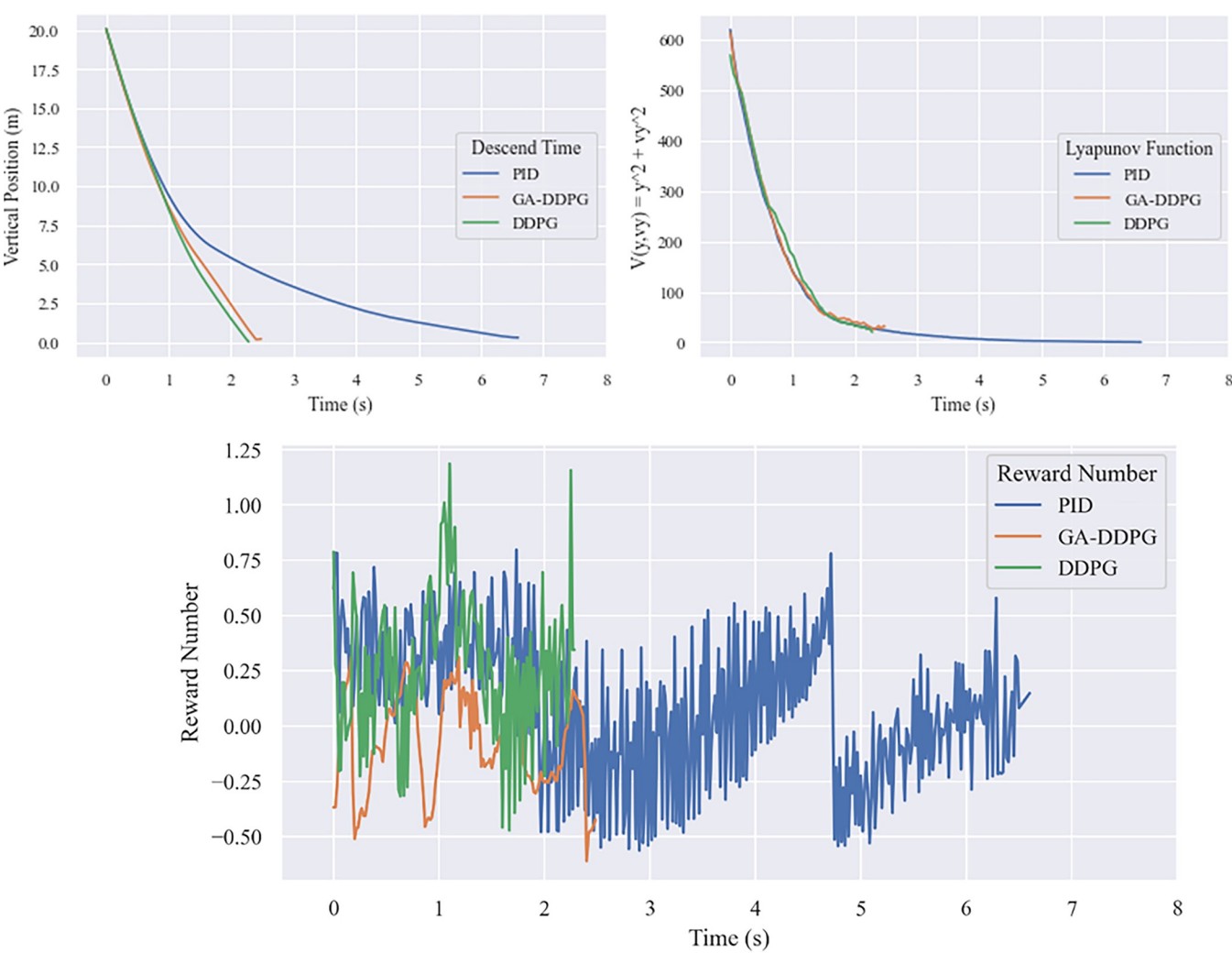

**Fig 23. Comparison of unobstructed landing.** (a) descending time, (b) Lyapunov function, (c) reward number.

from the Machine Learning version, where in ML, the k-fold method is used to tune hyper-parameters or evaluate model performance. The proposed method in Fig 25 is novel and combines the trained agents' body $(RSF + NN)_A$ with few GA-searched RSFs.

Proposed cross validation in RL is used to determine the best combination of a neural network trained model and a reward shaping function (RSF) through resampling based on the fitness values. In our proposed approach, the fitness resampling is conducted over 150 testing iterations, resulting in a matrix that compares trained NN models on the x-axis and tested RSFs on the y-axis, as shown in Table 12.

The results of the cross validation show that the combination of NN + RSF ind #17 has the highest fitness for trained agent, as indicated by the ranking on the x-axis, while the ranking on the y-axis indicates that RSF ind #29 is more accepted when used in extensive testing phase.

The x-axis reflects the performance of the NN agent as a controller, while the y-axis is useful for any selected NN to improve performancefurther via fine-tuning RSF. For example, in the testing phase, the GA-DDPG agent of ind #17 (red ellipse) still can be improved by using ind #29 (blue ellipse) as the RSF in testing phase.

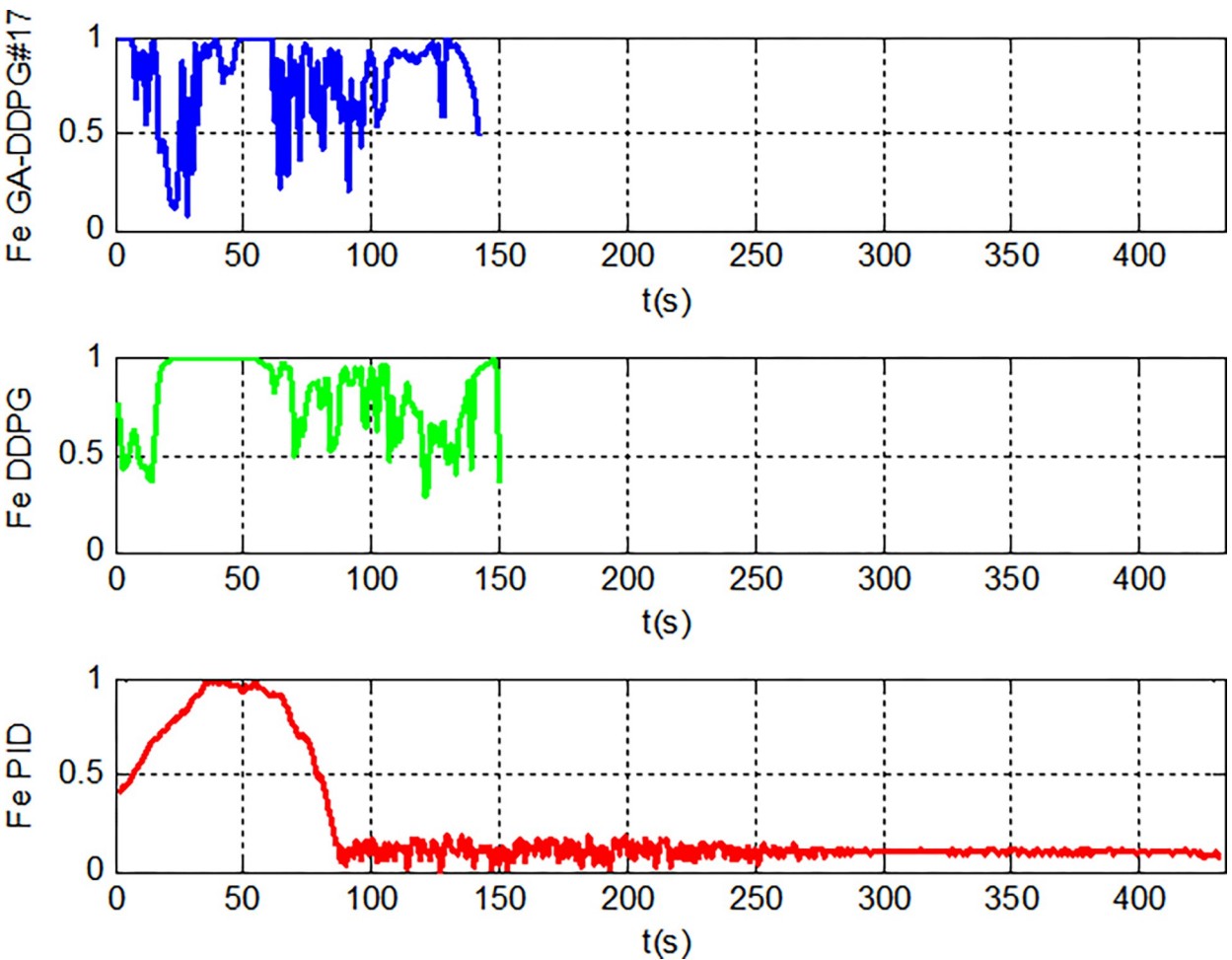

**Fig 24. Main engine thrust profile in algorithm comparison test.**

From the PbGA (potential-based genetic algorithm) search results of the reward function, the best landing trajectories are achieved when the reward shaping function are:

$$individual\ 17 = -[1]100\theta - [1]0.03FE - [1]0.003FS \tag{32}$$

as the best RSF for TRAINING phase which encourage the agent to stay upright and use the side thruster efficiently. While the following individual:

$$individual\ 29 = -[1]20\sqrt{[1]x^2 + [1]y^2} - [1]10\sqrt{[1]\dot{x}^2 + [1]\dot{y}^2} - [1]100\theta - [1]3\dot{\theta} + [1].leg_{left}$$
$$+ [1].leg_{right} - [1]0.03FE - [1]0.003FS - [1].TVC\_Angle \tag{33}$$

as the best RSF for TESTING phase which beats closely the previous individual by activating all aspects of sensors and actuators.

An exhaustive 1000 times Monte Carlo test was conducted to compare the best individual from the GA-searched finding (ind #17) against the reference DDPG, which ranks second in the cross-validation of NN vs. RSF in Table 12.

In term of positional miss distance (Fig 26A), GA-DDPG can beat the reference DDPG to 50% more accurate with/without wind disturbance, while in term of vertical landing speed (Fig 26B) the GA-DDPG beat DDPG to 30% more slower.

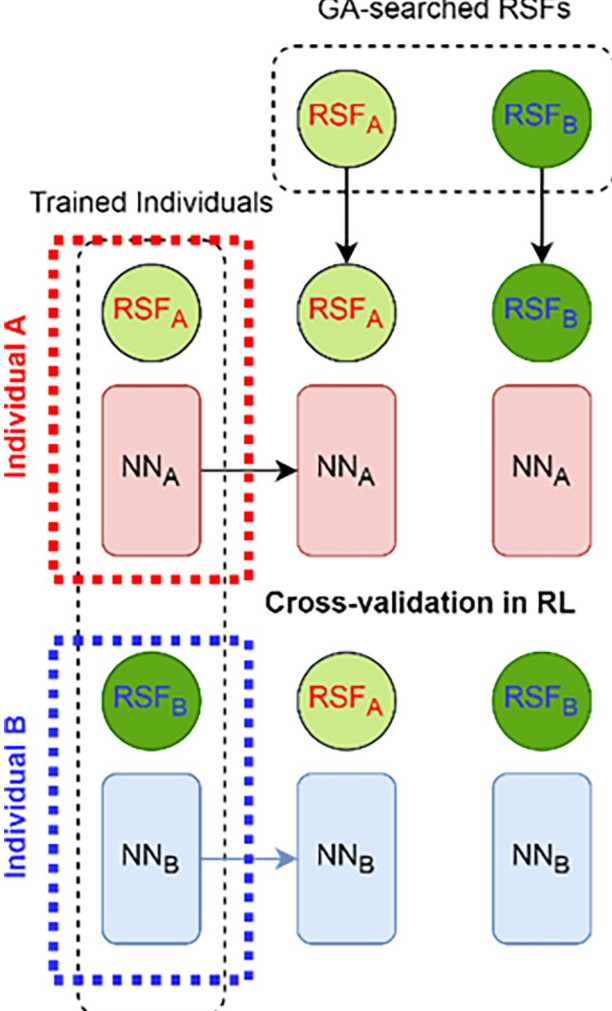

**Fig 25. Cross Validation mechanism for an RL case.**

Why GA-DDPG can have better result than reference DDPG? It seems that from the chromosome analysis of the individual #17, shows that lesser chromosome doesn't necessarily mean yield lower performance. By having slimmer DNA itself, the results from usage of ind

**Table 12. Heatmap of cross-validation of the 1000x monte carlotest.**

| | Individual | ind17 | DDPG | ind29 | ind36 | ind52 | ind57 | SUM | Rank |
|---|---|---|---|---|---|---|---|---|---|
| | | **NN Model–Training Results** | | | | | | | |
| RSF–Testing Inputs | ind17 | 4.8 | 4.65 | 1.75 | 2.7 | 3.1 | 3.9 | 16.1 | 5 |
| | ind29 | 5.75 | 5.5 | 4.05 | 2.6 | 3.8 | 4.35 | 20.3 | 1 |
| | ind36 | 5.05 | 5.15 | 1.45 | 3.8 | 5.05 | 4.55 | 20 | 2 |
| | ind52 | 5.6 | 5.3 | 2.8 | 4.7 | 3.95 | 3.35 | 20.1 | 3 |
| | ind57 | 4.7 | 5 | 2.7 | 1.2 | 4.65 | 3.7 | 17.25 | 4 |
| | SUM | 21.1 | 20.95 | 11 | 12.3 | 17.45 | 15.95 | | |
| | RANK | 1 | 2 | 6 | 5 | 3 | 4 | | |

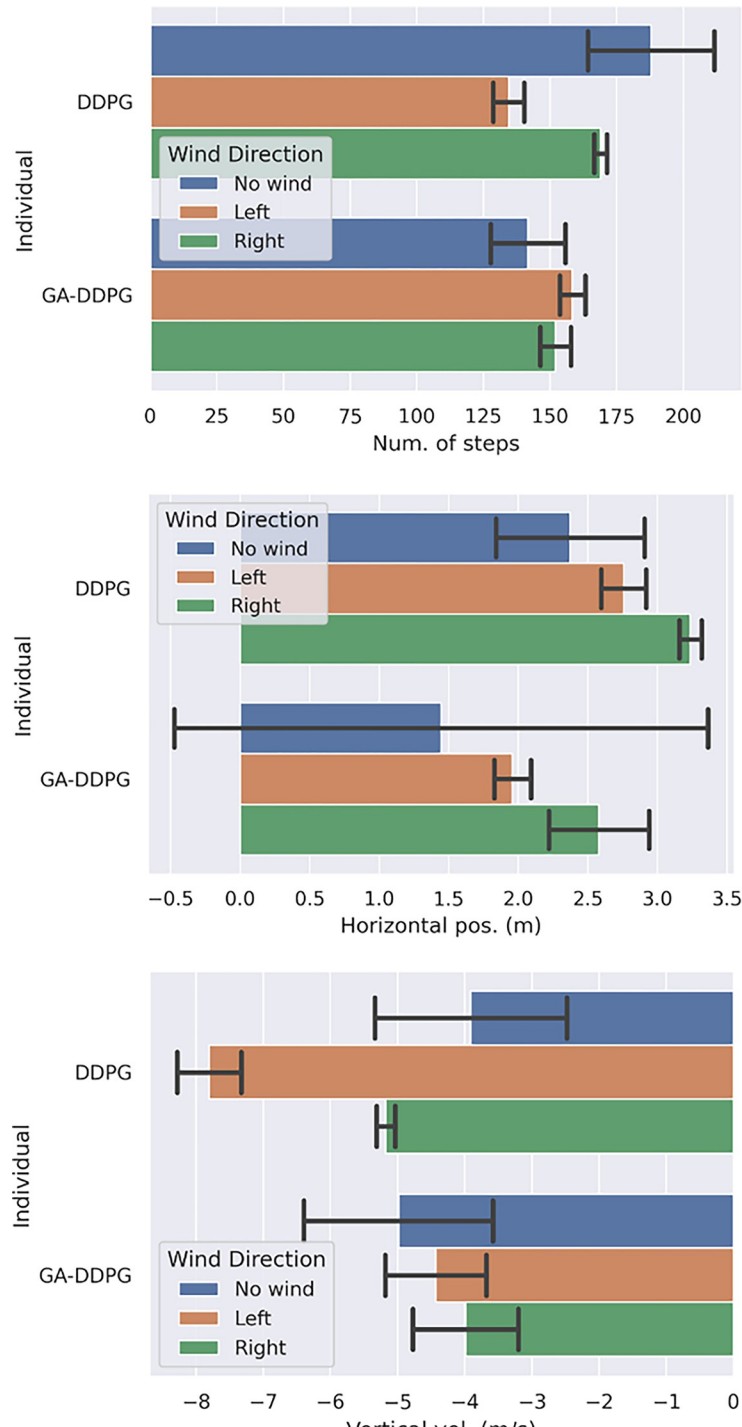

**Fig 26.** Mean values of 1000x Monte Carlo test results for the both models DDPG and GA-DDPG. (a) final horizontal position (in meters), and (b) final landing velocity (in meters per second) and (c) number of steps done. All graphs used band strings as the standard deviation.

#17 (which only has 3 shaping factors) as an RSF for training can provide an increase in producing better DDPG agent. A factor that playsmain role here is the initial main position which has zero horizontal deviation, and the relatively short initial altitude.

## 5.5 Hybrid deterministic stochastic controller switching test

The RL agent GA-DDPG Individual #17 is a proven controller in Monte Carlo test and cross validation test. It beats other controllers, whether PID or reference DDPG in all aspects, but the miss distance of the terminal landing position (Table 11 and Fig 21). The miss distance difference between PID and GA-DDPG is quite huge (0.09m vs 1.22m). This condition leads this research to overlook the possibility to combine the deterministic approach and the stochastic one to construct a new controller, called the HYDESTOC controller(Fig 6). It operates by switching action based on altitude. In Table 13, the switching altitudes are: 1m, 2m, 5m, 10m, 15m, with the initial descending altitude is 20m as depicted in Eq (26).One other aspect that needed to be alternated is the nozzle's thrust vector control (TVC), whether it is on or off, anticipating the shortness of the initial altitude. The basis of the HYDESTOC controller is the GA-DDPG agent individual #17, which is the fittest individual from the previous PbGA search. Below are the flight performance of the HYDESTOC controllersvs. PID controllers, where each row/scenario is conducted in Monte Carlo test for 100 times each.

Table 13 shows that combining deterministic approach and stochastic one is indeed provides a higher level of controller. As we can see, the PID is consistently providing good miss

**Table 13. Performance Table of the HYDESTOC controller vs. PID controller.**

| Controller | TVC | Switch Height | total_step | lastX | lastY | lastVx | lastVy | lastAngle | lastFuel | switchedAt | failedLand | total_reward | dX < 0.1m | dx < 0.02m | Fitness |
|---|---|---|---|---|---|---|---|---|---|---|---|---|---|---|---|
| | | | m | steps | m | m | m/s | m/s | deg | tons | steps | # | | % | % |
| PID | Off | - | 442.72 | -0.38 | 0.27 | -0.04 | -0.40 | -0.74 | 2.49 | 0 | 8 | 36 | 9 | 0 | 6 |
| PID | On | - | 450.41 | -0.22 | 0.23 | 0.10 | -0.48 | 0.32 | 2.45 | 0 | 5 | 41 | 5 | 1 | 5 |
| GA-DDPG + PID | Off | 15 | 227.00 | 0.05 | 0.27 | 0.59 | -0.53 | 3.50 | 3.59 | 26 | 0 | -14 | 6 | 2 | 8 |
| GA-DDPG + PID | On | 15 | 212.00 | 0.59 | 0.15 | 0.37 | -0.55 | 0.25 | 3.56 | 26 | 9 | -4 | 5 | 2 | 7 |
| GA-DDPG + PID | Off | 10 | 211.00 | 1.55 | 0.28 | -0.13 | -0.55 | -3.36 | 3.75 | 54 | 0 | -16 | 2 | 0 | 6 |
| GA-DDPG + PID | On | 10 | 199.00 | 0.26 | 0.14 | 0.27 | -1.26 | 0.04 | 3.72 | 51 | 7 | -5 | 0 | 0 | 5 |
| GA-DDPG + PID | Off | 8 | 227.00 | 1.78 | 0.26 | -0.19 | -0.73 | -3.07 | 3.79 | 64 | 7 | -15 | 3 | 1 | 6 |
| GA-DDPG + PID | On | 8 | 189.00 | 0.32 | 0.15 | 0.27 | -1.37 | 0.01 | 3.76 | 65 | 6 | -4.8 | 2 | 0 | 5 |
| GA-DDPG + PID | Off | 5 | 189.00 | 0.79 | 0.27 | 0.19 | -0.71 | 0.01 | 3.89 | 89 | 6 | -15 | 1 | 0 | 5 |
| GA-DDPG + PID | On | 5 | 186.00 | 0.77 | 0.15 | 0.06 | -0.96 | 0.29 | 3.75 | 95 | 0 | -5.5 | 5 | 0 | 8 |
| GA-DDPG + PID | Off | 2 | 219.32 | 3.11 | 0.26 | 0.32 | -0.95 | 2.60 | 3.62 | 125 | 3 | -15 | 0 | 0 | 4 |
| GA-DDPG + PID | On | 2 | 153.15 | 0.83 | 0.15 | 1.37 | -0.67 | 0.69 | 3.97 | 123 | 0 | -4.8 | 5 | 3 | 6 |
| GA-DDPG + PID | Off | 1 | 139.36 | 1.79 | 0.19 | 0.93 | -2.47 | 0.20 | 4.09 | 124 | 9 | -15 | 1 | 0 | 4 |
| GA-DDPG + PID | On | 1 | 198.75 | 2.10 | 0.24 | 0.88 | -1.11 | 0.35 | 3.68 | 156 | 1 | -5.5 | 0 | 0 | 5 |

PID Controllers

Acceptable Hybrid (PID+Ind#17)

Not Acceptable Hybrid (PID+Ind#17)

Highly Desired Aspect

Desired Aspect

Undesired Aspect

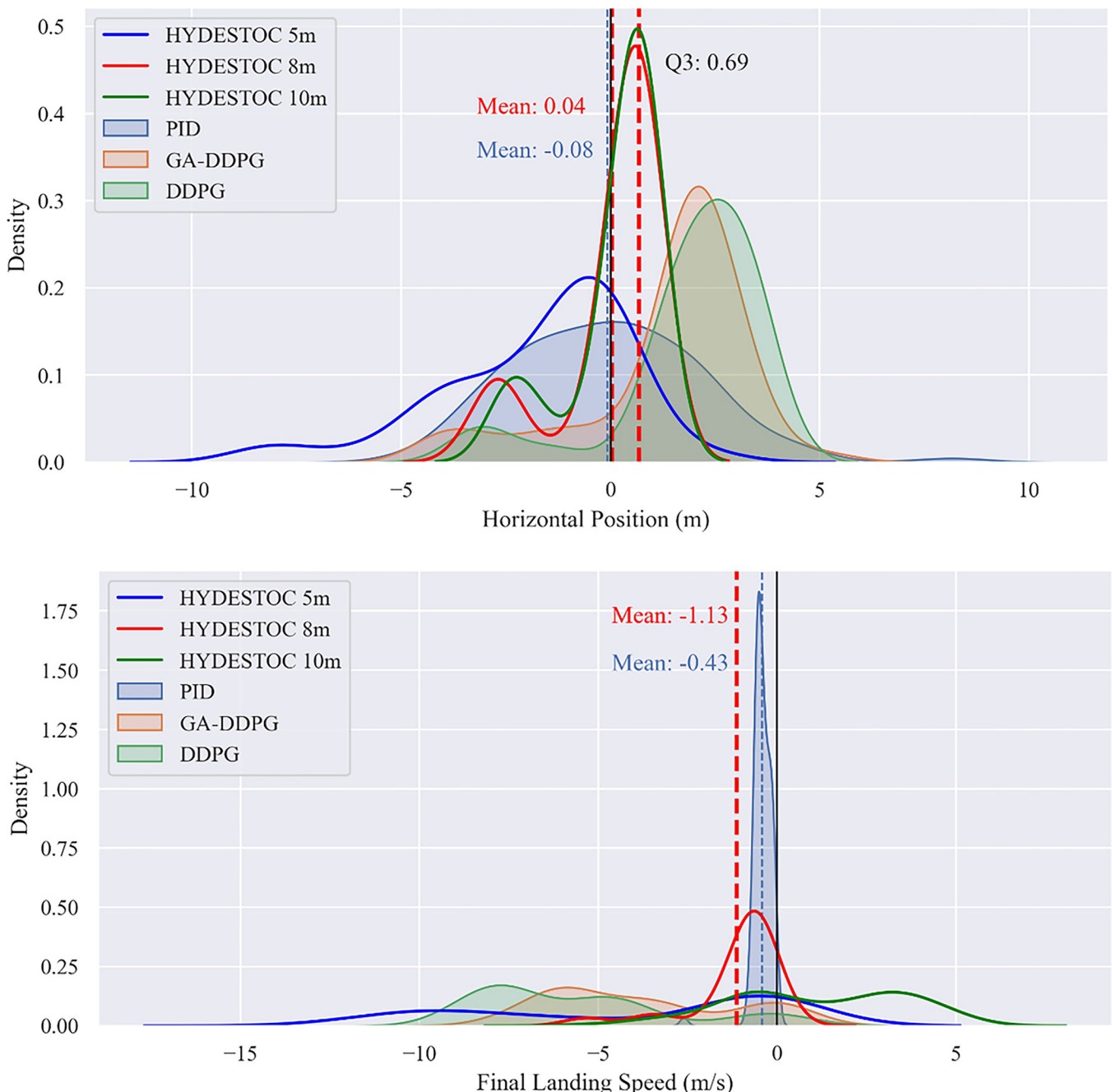

**Fig 27.** The improvement brought by HYDESTOC in terms of (a) horizontal position and (b) final vertical speed.

distance performance, but two points are to be remarked. First, it is not unbeatable. HYDES-TOC using $h_{switch}$ = 15m and 5m beat the PID controller without TVC in many aspects, and after deeper examination, HYDESTOC-15m-TVC Off can provide 6% of landing with miss distance under 10cm, and 2% for miss distance under 2cm, which leads to a new record in fitness value that can't be achieved by the PID controllers. Second point that is interesting to be investigated further is the high number of Failed Landings for the PID controllers (5–8%), while the HYDESTOC-15m-TVC Off is once again succeed to carry all the landings as

**Table 14. Averages of the benchmark results between GA-DDPG, DDPG, PID.**

| Controller | Miss distance (m) | Landing Speed (m/s) | Landing time(s) | % Faster | Remaining fuel (tons) | % Better |
|---|---|---|---|---|---|---|
| GA-DDPG | 1.22 | -2.50 | 128 | 60% | 4.09 | 30% |
| DDPG | 1.56 | -5.56 | 129 | 58% | 4.13 | 39% |
| PID | 0.08 | -0.43 | 313 | 0% | 2.97 | 0% |
| HYDESTOC-10 | 0.12 | -0.86 | 300 | 4% | 3.15 | 6% |
| HYDESTOC-8 | 0.04 | -1.13 | 197 | 37% | 3.70 | 28% |
| HYDESTOC-5 | 0.69 | -1.17 | 141 | 54% | 4.27 | 44% |

expected, which is no accidents. From the fitness value, we can conclude that the optimal switching height is 15m which reflects the time composition of the controller is 25% stochastic +75% deterministic. In Fig 27, all HYDESTOC prove to increase performances of the DDPG controller in terms of miss distance / horizontal position and landing speed. Table 14 and Fig 27A show the mean of HYDESTOC8m is 0.04m which is more better than the mean of PID that has 0.08m, even in Fig 27A HYDESTOC 8m and 10m's Q3 is 0.69m which lies under 1 meter, also the miss distance averages of HYDESTOC 8m and 10m are under 1 meter. In terms of landing speed average (Fig 27B), HYDESTOC 10m can achieve the average of -0.86 m/s that is under 1 m/s which resembles to the PID's average -0.43m/s, and all HYDESTOC's average are much better than the DDPG and GA-DDPG ones which are -5.56 m/s and -2.5 m/s respectively.

Density of probability is utilized here to describe the frequency distribution of the landing performance, in terms of final horizontal position and final landing speed. Higher density means higher frequency of occurrence that can be exerted by the corresponding algorithm. HYDESTOC-8 shows a higher one than the pure stochastic controllers. From mean value in Fig 27 and Table 14, it is emphasized that the algorithm HYDESTOC-8 shows significant improvement over the stochastic controller (DDPG, GA-DDPG) and slight improvement over the deterministic one.

To validate whether switching height plays big role in determining the performance of the HYDESTOC controller, statistic tests are conducted to check the existence of the relationship between input (states) and output (actions) for each data group generated by the selected switching heights. Generated data is considered continuous, therefore its hypothesis will be tested using ANOVA, then checked for its normality and homogeneity of its variances. Below are the results of the statistic tests conducted.

Table 15 shows that HYDESTOC 5m and 8m are nearly similar in variances (1.55 vs 1.59), while the 10m has a quite different one (1.21). Visually in Fig 27A we can see that the normal shaped green line constructs a similar shape with the red line, while the blue line has a quite different one. This leads to the only explanation that the switching height has a significant role to construct the data result's distribution. Table 16 reveals that P-value is lower than the designated significance level or $\alpha$-number that is 0.05 which indicates that the null-hypothesis is rejected, therefore the relation between switching altitude and miss distance is indeed exists.

**Table 15. Summary of the data population used in statistic tests are taken from the miss distances of the HYDESTOC 5m, 8m, 10m.**

| Groups | Count | Sum | Average | Variance |
|---|---|---|---|---|
| HYDESTOC-10m | 100 | 116.63 | 0.12 | 1.55 |
| HYDESTOC-8m | 38 | 1.46 | 0.04 | 1.59 |
| HYDESTOC-5m | 24 | 3.09 | 1.16 | 1.21 |
| | Global Average = | | 0.44 | |

**Table 16. Summary of the ANOVA test of the miss distances from HYDESTOC 5m, 8m, 10m.**

| Source of Variation | SS | df | MS | F | P-value | F crit |
|---|---|---|---|---|---|---|
| Between Groups | 45.82 | 2 | 22.91 | 15.09 | 9.95E-07 | 3.05 |
| Within Groups | 241.36 | 159 | 1.51 | | | |
| Total | 287.19 | 161 | | | | |

This remark is strengthened by the F-test, where the F-number is found to be higher than the F-criteria which also shows that the observed datas are not normally distributed. From this test, we can notice that between 8m to 10m the HYDESTOC have tendency to be similar in a good way where the miss distance averages are under 1 meter, while the 5m has more than 1 meter. The optimal range of switching altitude is discovered between 8m to 10m and needed to be scrutinized further.

Below are the trajectories and the flight states of the PIDs and HYDESTOC-15ms.

Fig 28 reveals few consistencies of the HYDESTOC controller that form certain pattern. First, the usage of TVC can reduce sharply the magnitude of the tilting angle to maximum of 4 degree, comparing to maximum of 10 degree when the TVC is turned off. As a consequence of this limited tilting angle, the mid-air deviation can also be reduced.

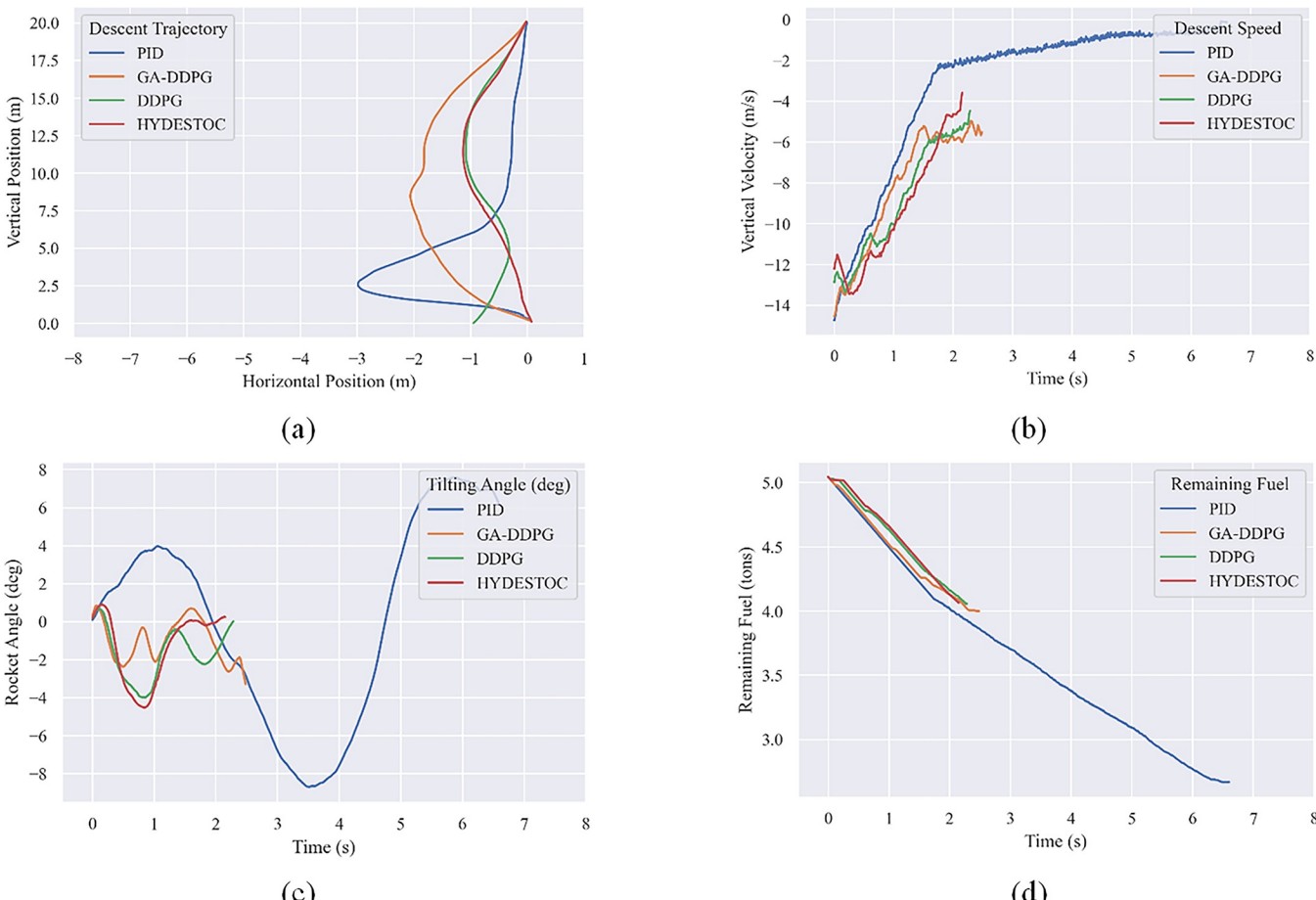

**Fig 28. Best Trajectories from HYDESTOC controller testing.** (a) Trajectory, (b) Descent Rate, (c) Tilt Angle and (d) Remaining Fuel Profile.

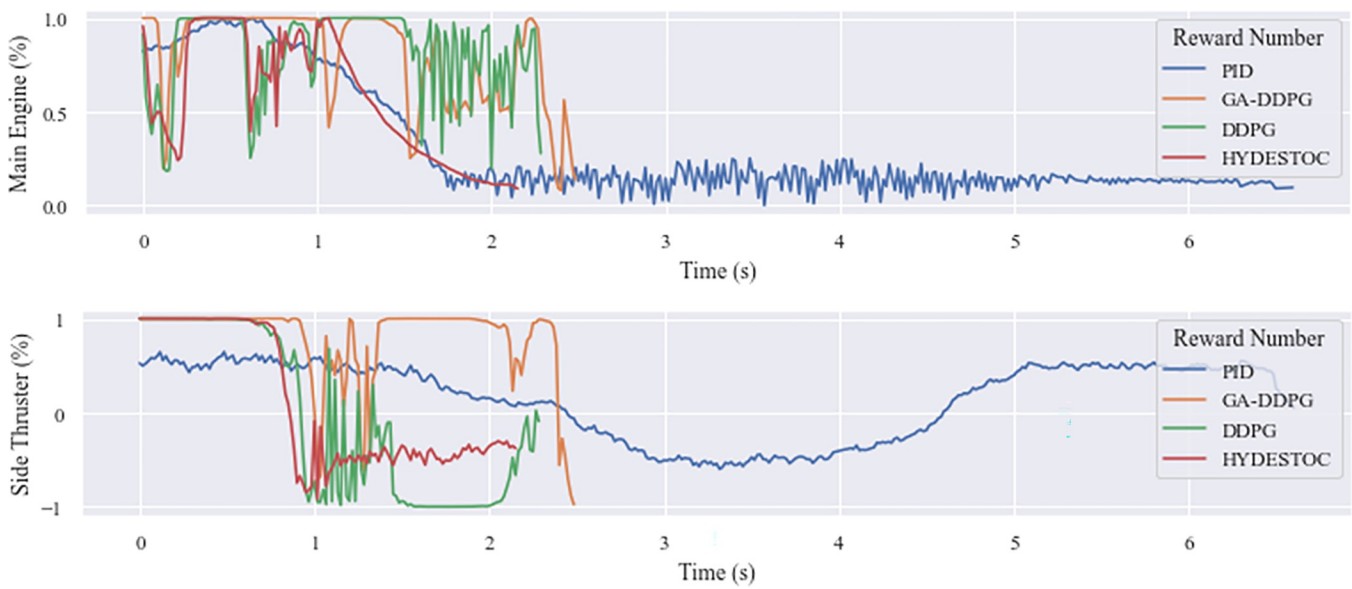

**Fig 29. Benchmark of thrust control HYDESTOC8m controller to previous ones.**

Second consistency is the tendency of the trajectory of HYDESTOC controller to make an 'onion-like' trajectory where at 2 to 5m the deviation is maximum, from there the controller tried to adjust for the last time the landing position and minimize the miss distance.

From Fig 29, we can see that the initial thrust of the HYDESTOC controller ask 100% of Fe command, while the PID and DDPG from Fig 29 didn't give thrust more than 50% of Fe, this phenomenon marks the third consistency where initial Fe number nearing 100% can reduce the flight time, thus the conserving fuel altogether.

HYDESTOC enhances both PID and DDPG controllers by implementing Full initial Fe thrust and extending the total shutdown time, providing the controller with more time to increase the precision of the rocket's landing position.

## 5.6 Robustness test against wind

Winds can disrupt the rocket landing process, therefore tests were conducted with wind interference to evaluate the robustness of the controllers in handling these conditions. The wind was simulated using the epsilon-greedy method, with random direction and magnitude more then the epsilon number. The wind tests were conducted five times on each controller, including the Individual 17, PID controller and reference DDPG. Table 17 shows the average fitness of each

**Table 17. Average wind test fitness.**

| Individual | $fit_{avg}$ |
|---|---|
| 17 | 5 |
| PID | 5 |
| DDPG | 4.6 |
| 52 | 3.9 |
| 36 | 3.2 |
| 57 | 2.4 |
| 29 | 1.6 |

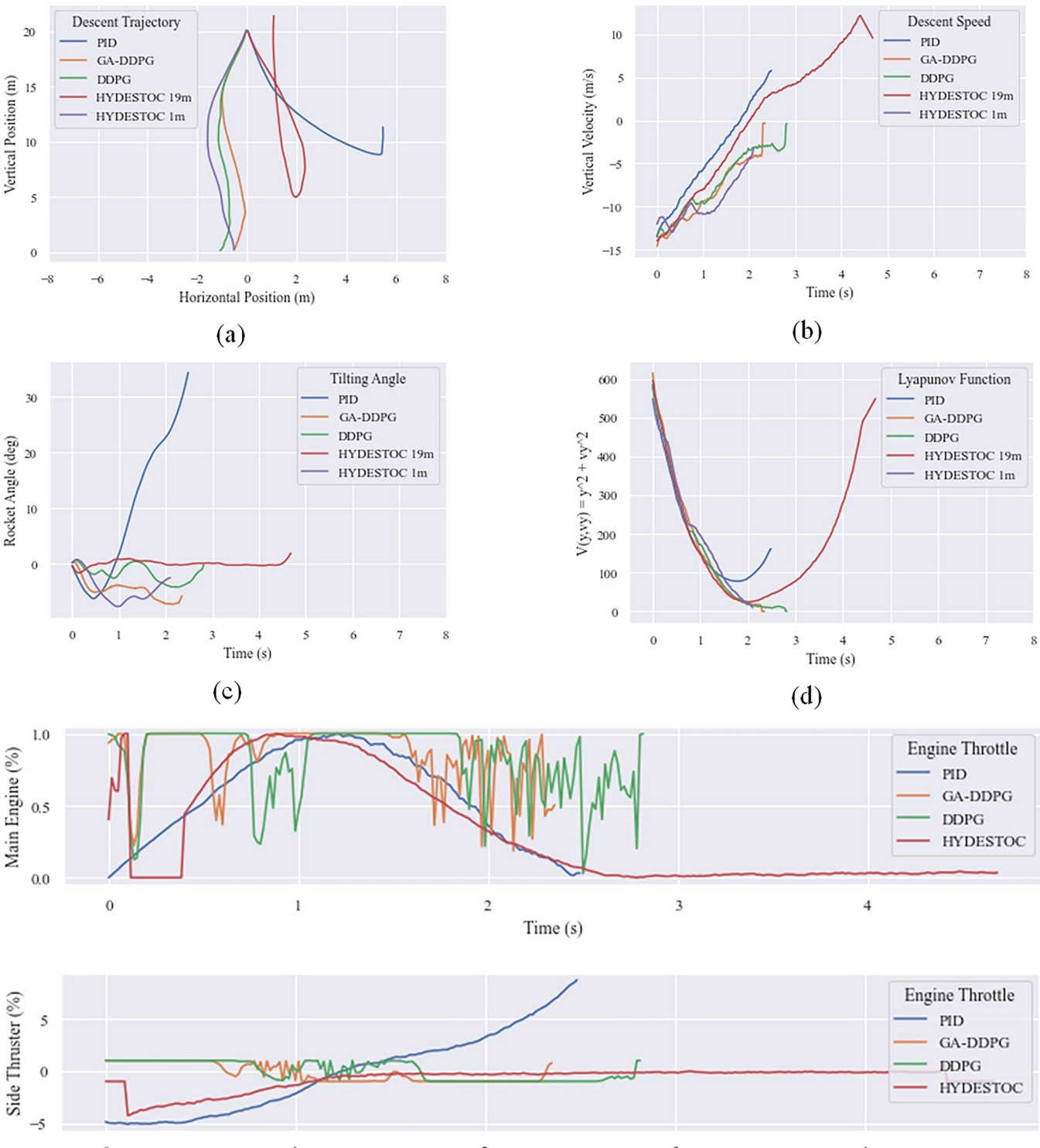

**Fig 30.** Wind disturbance test results for Fwind = 50 N, (a) Flight Trajectory, (b) Descent Speed, (c) Tilting Angle, (d) Lyapunov Function, (e) Engine Throttle.

controller from the five experiments. The best overall performance in handling wind interference was achieved by the Individual 17 and PID controllers, with an average fitness of 5.

The results of the landing trajectories in Fig 30 demonstrate that the GA-DDPG Individual 17 and reference DDPG controllers have a greater miss distance and higher landing speed compared to a windless landing, but are still within the safe range of under 3 meters and 5 m/s, while the tilt of the rocket is smaller and fuel consumption is not significantly increased. On the other hand, even when PID controller is able to maintain a low miss distance and descent rate in comparison to the windless landing, however, the rocket tilts more and results in a significantly increased fuel consumption. The GA-DDPG Individual 17 controller had a better performance than the reference DDPG and PID controllers, with improved miss distance, vertical speed and tilt of the rocket. However, with wind disturbance, the rocket was unable to return to the center point and had a miss distance of 0.5m.

The DDPG controllers, in the other hand, try to compensate the wind direction. The wind direction is from left to right, DDPGs move the rocket from center to the left first (h = 20m–10m), then start sliding down (h = 10m-5m) while drifting a little bit to the right. In the last phase (h = 5m-0m) the last correction maneuver is activated. It is in the last phase HYDESTOC differs to the DDPGs, because it used PID controller instead that has reputation to brake in the last phase to give time to correct position.

In terms of position, the rocket controller that landed closest to the center point is HYDESTOC-10m followed by GA-DDPG, HYDESTOC-1m and DDPG respectively. While the PID failed to land the rocket completely, which is similar to the HYDESTOC-19m, this is due to the inabilty of majority part of linearized controller to handle non-linearity caused by the wind.

In terms of landing speed, the rocket controller that landed with the lowest speed possible under wind disturbance is HYDESTOC-1m, HYDESTOC-10m, GA-DDPG then DDPG, respectively. While the PID and HYDESTOC-19m failed and fly away.

Fig 31 shows probability density comparison of the position and speed of the rocket's landing under wind disturbance. Table 18 shows that HYDESTOC 5m and 8m are nearly similar in variances (0.6 vs 0.69), while the 10m has a quite different one (0.87). Visually in Fig 31 we can see that the normal red green line constructs a similar shape with the blue line, while the green line has a quite different one, indicate that the switching height still has a significant role to construct the data result's distribution. In more detail, Table 19 displays P-value is lower than the designated significance level or $\alpha$-number that is 0.05 which indicates that the null-hypothesis is rejected, therefore the relation between switching altitude and miss distance is indeed exists. This remark is strengthened by the F-test, where the F-number is found to be higher than the F-criteria which also shows that the observed datas are not normally distributed. From this statisticstestof miss distance and the previous one, HYDESTOC 8m can be noticed to have consistent good results. Combined with similar consistent good result in final landing speed, we can take conclusion that the optimal switching altitude is 8m.

In more detail, the landing success rate under wind disturbance can be seen in last column of Table 20, where GA-DDPG is registered to never fail to land, followed by DDPG, HYDESTOC-10m, 8m and 5m. All of those data clearly show that hybridization of deterministic method to a stochastic controller or vice versa indeed improve the landing performance of the DDPGs and even increase the success rate PID. Nevertheless GA-DDPG is also proven to be a robust controller that can handle wind disturbance well. Deterministic control approach represented by PID can help accuracy of the DDPGs, while the failing tendency of PID to handle non-linearity of environment can be overcome by the stochastic approach of the DDPGs, especially the GA-DDPG which is proven to be highly robust. HYDESTOC 8m is selected to show the most optimal switching altitude.

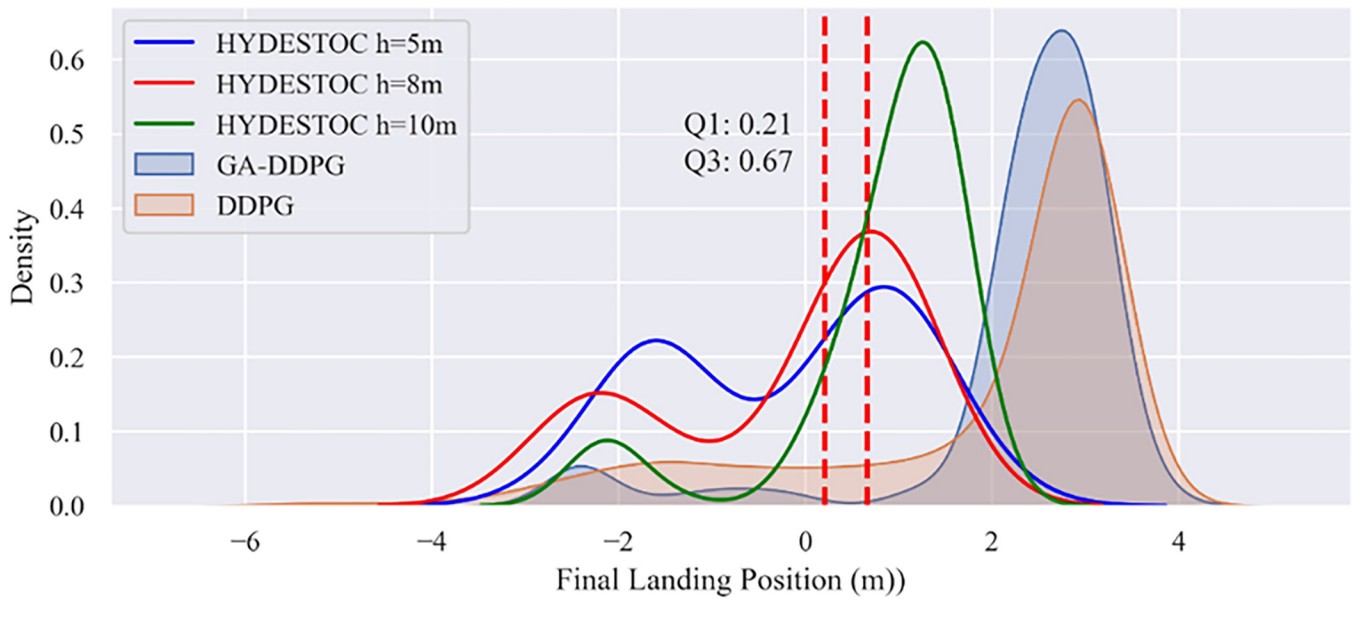

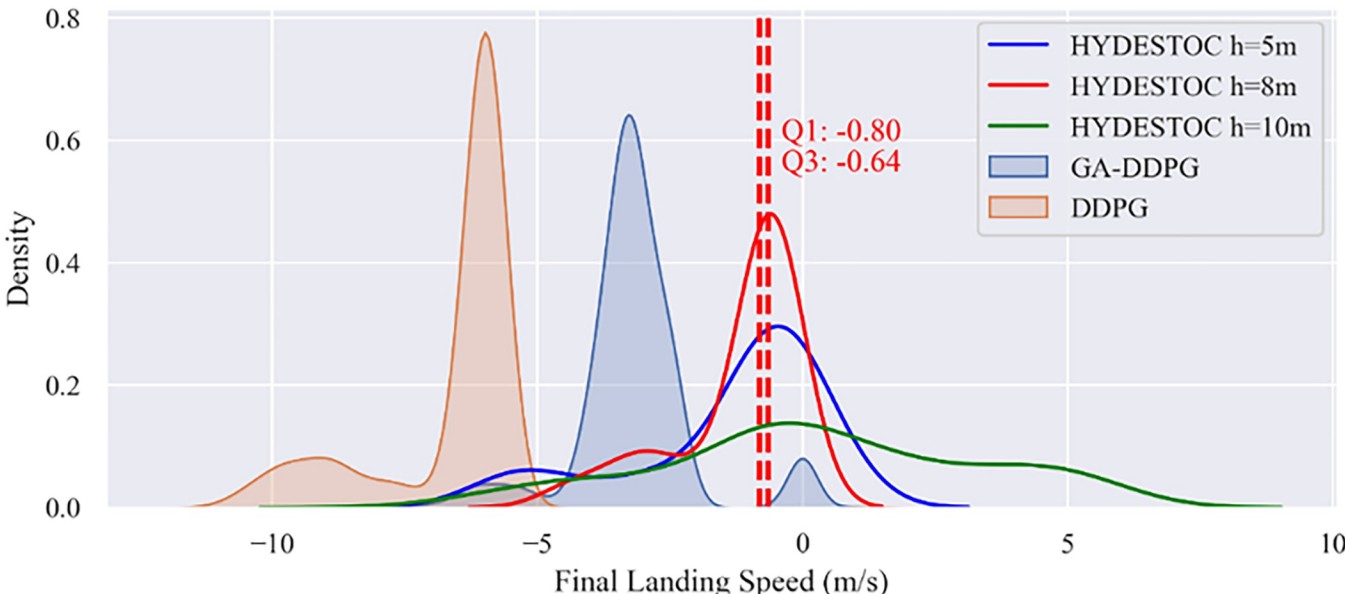

**Fig 31.** The improvement brought by HYDESTOC under Wind disturbance, compared to GA-DDPG and DDPG with$F_{wind}$ = 50.0 N in terms of (a) landing accuracy and (b) final vertical speed.

**Table 18. Summary of the data population used in statistic tests are taken from the miss distances of the HYDESTOC 5m, 8m, 10m.**

| Groups | Count | Sum | Average | Variance |
|---|---|---|---|---|
| HYDESTOC-5m | 100 | 32.06 | 0.32 | 0.60 |
| HYDESTOC-8m | 100 | -36.69 | -0.37 | 0.69 |
| HYDESTOC-10m | 100 | 70.48 | 0.70 | 0.87 |
| | Global Average = | | 0.44 | |

**Table 19. Summary of the ANOVA test of the miss distances from HYDESTOC 5m, 8m, 10m.**

| Source of Variation | SS | df | MS | F | P-value | F crit |
|---|---|---|---|---|---|---|
| Between Groups | 58.96 | 2 | 29.48 | 40.83 | 2.16E-16 | 3.03 |
| Within Groups | 214.42 | 297 | 0.72 | | | |
| Total | 273.38 | 299 | | | | |

# 6 Conclusions and suggestions

The experiments conducted on the search for a GA-optimized DDPG controller for the powered descent guidance of a VTVL rocket have yielded some notable findings. While the deterministic control approach represented by PID consistently performed better in terms of terminal accuracy, the stochastic approaches demonstrated their advantages in terms of optimization and disturbance rejection. Both the reference DDPG and the GA-assisted DDPG consumed less fuel than the PID approach, while maintaining acceptable position and speed performance. However, the most striking discovery was the chattering phenomenon of the main engine throttle command (Fe) in the control system, where the GA-based DDPG had the lowest chatter and PID had the highest. The presence of chattering is problematic, especially for liquid propulsion engines, and requires additional research to find a solution.

To address the limitations of both approaches, a novel smart guidance approach has been proposed that combines both deterministic and stochastic control methods, offering a more comprehensive and unified guidance solution. The optimization of GA-DDPG involves a GA-based search methodology that identifies the optimal reward shaping function required for training the RL agent to achieve its best neural network form.

Several levels of Monte Carlo testing were performed, including MC-10, MC-100, and MC-1000, representing 10, 100, and 1000 tests, respectively. The MC-10 was utilized for cross-validation of both the training and testing results of the GA-DDPG. Meanwhile, the MC-100 was used to investigate the capability of HYDESTOC and map the switching altitude based on environmental conditions. Finally, MC-1000 was extensively carried out to verify the robustness of the controllers against wind disturbances.

The results of the exhaustive Monte Carlo experiments showed that the GA-DDPG outperformed the DDPG in all aspects of performance, particularly when wind disturbances were introduced. The GA-DDPG achieved shorter flight times, shorter miss distances, and lower vertical velocities at touchdown than the DDPG. However, there were some observations to be made, such as the high standard deviation of the miss distance performance of the GA-DDPG in no-wind conditions, which was expected as the agent tended to land at (-)1.5m/s or (+) 2m/s, still within a 2-meter radius. Additionally, the DDPG performed well in no-wind conditions but poorly in wind conditions, suggesting that it is prone to disturbances, while the GA-DDPG is more robust.

**Table 20. Landing characteristics averages under wind disturbance $F_{wind}$ = 50.0 N.**

| Controllers | t (sec) | $x_t$ (m) | $y_t$ (m) | $\dot{x}_t$ (m/s) | $\dot{y}_t$ (m/s) | $\theta_t$ (°) | success (%) |
|---|---|---|---|---|---|---|---|
| Individual 17/GA-DDPG | 2.38 | 1.22 | 0.69 | 1.51 | -2.19 | -2.43 | 100 |
| Reference/DDPG | 2.68 | 2.02 | 0.26 | 0.08 | -2.76 | -5.98 | 91 |
| PID | failed | - | - | - | - | - | 0 |
| HYDESTOC Hswitch = 5m | 3.67 | 0.32 | 0.15 | 0.37 | -1.39 | -0.32 | 68 |
| HYDESTOC Hswitch = 8m | 3.52 | 0.37 | 0.15 | 0.28 | -1.22 | 0.15 | 71 |
| HYDESTOC Hswitch = 10m | 3.80 | 0.70 | 0.15 | 0.24 | -0.98 | 0.11 | 83 |

The cross-validation resulted that individual #17 is indeed the winner of the trained neural network found by the GA natural selection, but in terms of RSF for testing purpose individual #29 yields the best performance. The proposed HYDESTOC helps both the DDPGs and PID to overcome their deficiency, the accuracy of HYDESTOC 8m is under 0.1m in terms of landing miss distance, the landing speed also decrease to under 2 m/s in normal condition. When wind disturbance is present, HYDESTOC 8m can decrease the failure rate of the landing to 29%, compared to 100% of failure in previous classical method, with the landing speed also under 2 m/s. In Monte Carlo 1000 tests also put emphasis on robustness of GA-DDPG against multiple direction of wind disturbance, and can beat reference DDPG up to 50% in miss distance and up to 30% in landing speed.

Further research suggestions include embedding the trained algorithms into popular microcontrollers, using a software/hardware in-the-loop research scheme with a high-fidelity flight simulator, and conducting evolutionary algorithm-based fine-tuning of the RSF to beat the best controller individual.

## Acknowledgments

Authors would like to thank you to Dr. Prawito Prajitno, Dr. Djati Handoko, Dr. Arief Sudarmadji, Diva Kartika Larasati and Dr. Adhi Harmoko for their valuable inputs in analysis and interpretation of data presented.

## Author Contributions

**Conceptualization:** Larasmoyo Nugroho, Rika Andiarti, Rini Akmeliawati, Sastra Kusuma Wijaya.

**Data curation:** Larasmoyo Nugroho.

**Formal analysis:** Larasmoyo Nugroho.

**Funding acquisition:** Larasmoyo Nugroho, Rika Andiarti, Sastra Kusuma Wijaya.

**Investigation:** Rika Andiarti, Sastra Kusuma Wijaya.

**Methodology:** Larasmoyo Nugroho, Rini Akmeliawati.

**Project administration:** Larasmoyo Nugroho.

**Software:** Larasmoyo Nugroho.

**Supervision:** Rika Andiarti, Rini Akmeliawati, Sastra Kusuma Wijaya.

**Validation:** Rika Andiarti, Sastra Kusuma Wijaya.

**Visualization:** Larasmoyo Nugroho.

**Writing – original draft:** Larasmoyo Nugroho.

**Writing – review & editing:** Larasmoyo Nugroho, Rini Akmeliawati.

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
