## [Decision Letter · Decision Letter 0]

13 Sep 2022

PONE-D-22-18693Usage of potential-based genetic algorithm (PbGA) to optimize the reward shaping function (RSF) of a deep reinforcement learning (DRL) for landing guidance of a reusable launch vehicle (RLV)PLOS ONE

Dear Dr. Nugroho,

Thank you for submitting your manuscript to PLOS ONE. It was carefully reviewed by three Reviewers including me as an Academic Editor (Reviewer #3). After careful consideration, we feel that it has merit but does not fully meet PLOS ONE’s publication criteria as it currently stands. Therefore, we invite you to submit a revised version of the manuscript that addresses the points raised during the review process.

In particular:

language problems should be removed,presentation problems should be removed,considered models and obtained results should be analyzed statistically (e.g. n-fold cross validation should be used to make results less dependent on selection of examples to train/test sets),measurement errors should be analyzed,description of ideas and experiments including data processing should be clarified,claims not supported by presented data should be removed or modified.

We look forward to receiving your revised manuscript.

Kind regards,

Maciej Huk, Ph.D.

Academic Editor

PLOS ONE

Journal Requirements:

6. Please amend the manuscript submission data (via Edit Submission) to include author Arief 

Sudarmadji and Djati Handoko.

7. Please ensure that you refer to Figure 5 to 25 in your text as, if accepted, production will need this reference to link the reader to the figure.

8. We note you have included a table to which you do not refer in the text of your manuscript. Please ensure that you refer to Table 1,3,4,5 and 6 in your text; if accepted, production will need this reference to link the reader to the Table.

Reviewers' comments:

Reviewer's Responses to Questions

**Comments to the Author**

1. Is the manuscript technically sound, and do the data support the conclusions?

Reviewer #1: Yes

Reviewer #2: Partly

Reviewer #3: No

2. Has the statistical analysis been performed appropriately and rigorously? 

Reviewer #1: Yes

Reviewer #2: No

Reviewer #3: No

3. Have the authors made all data underlying the findings in their manuscript fully available?

Reviewer #1: Yes

Reviewer #2: Yes

Reviewer #3: Yes

4. Is the manuscript presented in an intelligible fashion and written in standard English?

Reviewer #1: Yes

Reviewer #2: No

Reviewer #3: No

5. Review Comments to the Author

Reviewer #1: The research is fascinating, and it was well conducted. There are minor inconveniences such as text typos (for example, in the last paragraph of page 2: "rateof" instead of "rate of"). However, nothing that can compromise the study. I have just a few comments:

a) Eqs [Disp-formula pone.0292539.e004] and [Disp-formula pone.0292539.e005], the " s', a' ” should be inside the parenthesis

b) In Table 1, the Disadvantages column was created/perceived by the authors, or was it stated in the original article? I think a paragraph after Table 1, should discuss its results.

c) Also, I think a paragraph after Table 2 should discuss its results. It is essential to understand the author's perspective on their findings

d) It’s a little obscure how the authors represent an individual in the GA. Is the individual represented using binary vectors (0, 1) or float numbers (from 0.0 to 1.0)? Please make it clear.

e) The section about the genetic operators' crossover and mutation is confusing, and I suggest a simple example using images instead of only the text explanation.

Finally, the experimentation is well done; however, the conclusion is weak and should be expanded as there is a lot to discuss and point out for future research.

Reviewer #2: This paper lends genetic algorithm to discover the optimal shape of the reward function used in the Deep Deterministic Policy Gradient. This contributes to reinforcement learning networks. My main concerns are:

1. The writing problems make this paper hard to read. For example, the sentence 'Figure 4.1 (a) shows the relationship of fuel consumption to FE, assuming FS = 0 and Figure 4.1 (b) shows fuel consumption to FS assuming FE = 0' in page 25, there is no (a) and (b) in figure 4, maybe figure 14?

2. There are a lot of errors in the equations in the paper, for example, Eqs [Disp-formula pone.0292539.e021] and [Disp-formula pone.0292539.e022].

We recommend the authors polish this paper and resubmit it again.

Reviewer #3: >>> 1. Language problems:

1.1 Tensorflow => TensorFlow

tensorflow => TensorFlow,

theano => Theano,

pytorch => PyTorch

1.2 rateof => rate of

1.3 toanother => to another

1.4 eachtime => each time

1.5 "with S denotes the state space, A denotes the action space"

"with S denoting" or "where S denotes" ?

1.6 "Qfunction becomes" => "Q-function becomes"

1.7 "there isa function" => "there is a function"

1.8 "learningalgorithm" => "learning algorithm"

1.9 "solvethe discreteness of datain" => "solve the discreteness of data in"

Spaces are missing in many, many places all over the text (the above cases are only examples).

1.10 [45] "Meric ¸li, C ¸., Meric ¸li, T" => "Meriçli, Ç., Meriçli, T"

1.11 Rubisztejn => Rubinsztejn

1.12 dynamicity => dynamics

1.13 [1] "IEEE SIGNAL PROCESSING MAGAZINE" => "IEEE Signal Processing Magazine"

[50,53,58] -> as above.

1.14 [22] "neural ´ networks" => "neural networks "

1.15 [53] "USING REINFORCEMENT LEARNING TO DESIGN MISSED THRUST RESILIENT TRAJECTORIES" => "Using Reinforcement Learning to Design Missed Thrust Resilient Trajectories"

Moreover, co-author "Laipert, Frank E." is missing on the list of authors of this reference

(please see: https://trs.jpl.nasa.gov/handle/2014/54432)

1.16 "Two types of neural networks used in the DDPG architecture and training: critic networks and actor networks."

grammar problem

1.17 "is arranged as follow" => "is arranged as follows"

grammar problem

1.18 "This idle status requires the pump engine to be runned at a low level fuel consumption"

=> "This idle state requires the pump engine to run at low fuel consumption"

grammar problem

>>> 2. Presentation problems:

2.1 Giving abbreviations of terms within the title is not a very good practice.

2.2 Table 1:

row 10, Disadvantages: cou-pled => coupled

rows 5, 14: poor formatting

2.3 "Wibben & Furfaro, 2016" - not needed new lines within the reference

2.4 Fig 4: Title is too general. There is no need to pup part of the description above the chart.

Axes need description, possibly with units

2.5 [Disp-formula pone.0292539.e004], [Disp-formula pone.0292539.e005], [Disp-formula pone.0292539.e014]: Why s,a parameters are presented as upper index?

2.6 Fig. 5: Navy blue background of the block above "actor.py" makes the text in the block unreadable

2.7 Fig. 6: all networks have the same internal structure. There is no point in repeating the same information.

Please consider separate presentation of NN architecture and then use it as a component to show its usage.

2.8: Fig 6: title seems to be not precise - it presents four neural networks of the same architecture used represent different input-output relations.

2.9 Table 3: Title is too general. Moreover: ALR, ACR, gamma are not explained within the title.

2.10 Fig 7: Title is too general. Is it an example? Is it schematic used by Authors in their work? If not then this should be clearly indicated. If yes then why example includes interactions with "plant" while the main work seems to be about spaceship control?

2.11 Fig. 8: Title is too general. The content is not clear. What are the parameters in "Reward shaping" block? How is it working?

It would be also good to add reference to Table 5 to help the Reader to find needed info.

2.12 Eq. 20, eq. 21: presentation of those equations is improper.

2.13 Fig. 9: Title is too general. Are parameters a-e related with parameters a-e on Fig. 8? If no then this can be misleading. If yes, then why parameters f-k are not presented on Fig. 9?

2.14 Fig. 10: conditional block is not clear. It should include the whole conditional being tested, not half of it.

2.15 Fig. 11: The title is too general. The figure presents schematic which seems to make no sense. Maybe more context within the title would make it more clear.

But I suggest to rethink this schematic. It has not much in common with Algorithm 1 (titles suggest both should present the same).

2.16 Table 5: Title is too general. Are the presented values initial for all the individuals?

2.17 Fig 13, Fig 14 - it is hard to understand what is the meaning of the horizontal axis. Is it the percentage of time of the landing maneuver or throttle setting of the given engine? Are those only examples or general characteristics of the engines assumed in the considered model? If the latter then how this is motivated?

2.18 Fig 15: The title is too general. The figure presents schematic which seems to make no sense. Maybe more context within the title would make it more clear. But I suggest to rethink this schematic.

2.19 Fig 16. The tile refers to "a" and "b". Where are "a" and "b"?

2.20 Table 7: Title is too general. How those values are used? How values of a and b are related to values of a and b for first individual in Table 5 (a is different in both tables)?

2.21 Table 10: "Individu" => "Individual"

>>> 3. Other problems:

3.1 Authors write:

"Genetic algorithms (GAs) in general and GAs in particular have been successful"

Such style of complicated writing is not making it easier to understand the most important information.

Maybe "Genetic algorithms (GAs) have been successful" will be enough?

3.2 In Table 6 Authors present specification of the hardware used during experiments.

Is there any influence of the details of the hardware specification on the results?

If yes then this should be discussed within the text in detail.

If not then please remove not needed information.

3.3 Authors write:

"Few ways to get around this issue are by introducing a target network to reduce the greediness of max-value of Q (?)"

What is the meaning of the "?" character? Missing reference?

3.4 It can be hard for the Reader to understand what are "Leg conditions". What are left and right leg? Graphical presentation of the model of the considered spaceship would be very helpful.

3.5 Many parameters of considered simulations seem to be initialized randomly (e.g. connection weights in neural networks, chromosomes of initial population in genetic algorithm). Thus experiments, results processing and presentation should be done to analyze statistical nature of analyzed method. Experiments should be repeated, n-fold cross-validation can be used during training/testing of NN based methods. Measured values should be presented with error bars and confidence intervals, etc. Comparison of outcomes of proposed method with results of other methods should be done with formal methods (statistic tests).

3.6 Conclusions given by the Authors are not backed up by the presented data (see remark 3.5).

>>> Summary:

The topic of the text is important and very interesting. But Authors seem to not care much for details of the presentation (both ideas and results) nor of the readability and clarity of described work. Many language and presentation errors exist within the text and should be removed before publication. Many elements should be reconsidered and reworked to be clear for the Reader. Now the text looks like full of complicated but not precisely described elements. In the effect the text is hard to follow. There is also very low chance to replicate the described experiments and results. Moreover, experiments and processing of results seem to be performed without care of the random nature of performed simulations and considered methods. In the effect, conclusions given by the Authors are not backed up by the presented data.

>>> Recommendation:

Reject.

===EOT===

6. PLOS authors have the option to publish the peer review history of their article (what does this mean?). If published, this will include your full peer review and any attached files.

Reviewer #1: **Yes: **Valdecy Pereira

Reviewer #2: No

Reviewer #3: No

---

## [Author Response · Author response to Decision Letter 0]

3 Nov 2022

Dear Dr. Maciej Huk and Reviewers, 

 We thank you very much for your positive response and invitation letter to our paper "PONE-D-22-18693. Usage of potential-based genetic algorithm (PbGA) to optimize the reward shaping function (RSF) of a deep reinforcement learning (DRL) for landing guidance of a reusable launch vehicle (RLV)". Your reviews and comments are very constructive and helpful for us to revise and improve the manuscript. 

 We have looked into the comments carefully and have made corrections and amendments accordingly.

 Please see the responses from the next page.

 The comments from the Editor’s and the Reviewers are highly appreciated.

Best regards.

Larasmoyo Nugroho, 

Rika Andiarti, 

Rini Akmeliawati, 

Ali Türker Kutay, 

Prawito Prajitno, 

Diva Kartika Larasati, 

Sastra Kusuma Wijaya.

Serpong, Indonesia. 28 October 2022

A. General Reviews:

Comment : language problems should be removed,

-We use internal MS Word's and Online Word's grammar checker to edit thoroughly any grammatical and syntax error.

-We depend also heavily to Word's internal thesaurus to ameliorate the vocabulary.

-Few paragraphs are resentenced via Quillbot, but still we refine them afterward. 

b. ✔ in the last paragraph of page 2: "rateof" instead of "rate of"). However, nothing that can compromise the study. 

c. ✔ Eq. 4 and 5, the " s', a' ” should be inside the parenthesis 

d. ✔ writing problems make this paper hard to read. For example, the sentence 'Figure 4.1 (a) shows the relationship of fuel consumption to FE, assuming FS = 0 and Figure 4.1 (b) shows fuel consumption to FS assuming FE = 0' in page 25, there is no (a) and (b) in figure 4, maybe figure 14? 

e. a lot of errors in the equations in the paper, for example, Eqs [Disp-formula pone.0292539.e021] and [Disp-formula pone.0292539.e022]. 

f. 1.1 ✔ Tensorflow => TensorFlow 

✔ tensorflow => TensorFlow, 

✔ theano => Theano, 

✔ pytorch => PyTorch 

1.2 ✔ rateof => rate of 

1.3 ✔ toanother => to another 

1.4 ✔ eachtime => each time 

1.5 ✔ "with S denotes the state space, A denotes the action space" 

"with S denoting" or "where S denotes" ? 

1.6 ✔ "Qfunction becomes" => "Q-function becomes" 

1.7 ✔ "there isa function" => "there is a function" 

1.8 ✔ "learningalgorithm" => "learning algorithm" 

1.9 ✔ "solvethe discreteness of datain" => "solve the discreteness of data in" 

✔ Spaces are missing in many, many places all over the text (the above cases are only examples). 

1.10 ✔ [45] "Meric ¸li, C ¸., Meric ¸li, T" => "Meriçli, Ç., Meriçli, T" 

1.11 ✔ Rubisztejn => Rubinsztejn 

1.12 ✔ dynamicity => dynamics 

1.13 ✔ [1] "IEEE SIGNAL PROCESSING MAGAZINE" => "IEEE Signal Processing Magazine" 

[50,53,58] -> as above. 

1.14 ✔ [22] "neural ´ networks" => "neural networks " 

1.15 ✔ [53] "USING REINFORCEMENT LEARNING TO DESIGN MISSED THRUST RESILIENT TRAJECTORIES" => "Using Reinforcement Learning to Design Missed Thrust Resilient Trajectories" 

Moreover, co-author "Laipert, Frank E." is missing on the list of authors of this reference 

(please see: https://trs.jpl.nasa.gov/handle/2014/54432) 

1.16 ✔ "Two types of neural networks used in the DDPG architecture and training: critic networks and actor networks." 

grammar problem 

1.17 ✔ "is arranged as follow" => "is arranged as follows" 

grammar problem 

1.18 ✔ "This idle status requires the pump engine to be runned at a low level fuel consumption" 

=> "This idle state requires the pump engine to run at low fuel consumption" 

grammar problem 

g. ✅ "Genetic algorithms (GAs) in general and GAs in particular have been successful" 

Such style of complicated writing is not making it easier to understand the most important information. 

Maybe "Genetic algorithms (GAs) have been successful" will be enough? 

h. 

Comment : presentation problems should be removed,

- We redraw few diagrams that need to be modified.

- The figure, table and equation captions are adjusted to the template

✅ a. Please ensure that you refer to Figure 5 to 25 in your text as, if accepted, production will need this reference to link the reader to the figure. 

✅ b. We note you have included a table to which you do not refer in the text of your manuscript. 

Please ensure that you refer to Table 1,3,4,5 and 6 in your text; if accepted, production will need this reference to link the reader to the Table. 

✅ c. In Table 1, the Disadvantages column was created/perceived by the authors, or was it stated in the original article? I think a paragraph after Table 1, should discuss its results. 

d. a paragraph after Table 2 should discuss its results. It is essential to understand the author's perspective on their findings.  removed 

e. 2.1 ✅ Giving abbreviations of terms within the title is not a very good practice. 

f. ✅ Tabel cross-validation diselipkan antara Tabel 10 & 11 

2.2 Table 1: 

✅ row 10, Disadvantages: cou-pled => coupled 

rows 5, 14: poor formatting 

2.3 ✅ "Wibben & Furfaro, 2016" - not needed new lines within the reference 

2.4 Fig 4 Three Typical OU-noise process, normal scale, by second (Fleming): Title is too general. There is no need to pup part of the description above the chart. 

Axes need description, possibly with units 

2.5 ✅ Eq. 4, Eq. 5, Eq. 13: Why s,a parameters are presented as upper index? 

2.6 ✅ Fig. 5 A generic DDPG and RSF framework: Navy blue background of the block above "actor.py" makes the text in the block unreadable 

2.7 ✅ Fig. 6 Input-output relations of the DDPG neural network architecture: all networks have the same internal structure. There is no point in repeating the same information. 

Please consider separate presentation of NN architecture and then use it as a component to show its usage. 

2.8: ✅ Fig 6 Input-output relations of the DDPG neural network architecture: title seems to be not precise - it presents four neural networks of the same architecture used represent different input-output relations. 

2.9 ✅ Table 3 Actor and Critic Network Hyperparameter Settings: Title is too general. Moreover: ✅ ALR, ACR, gamma are not explained within the title. 

2.10 ✅ Fig 7 A more detailed interaction between Agent - Reward Shaping Function – Environment: Title is too general. Is it an example? Is it schematic used by Authors in their work? If not then this should be clearly indicated. If yes then why example includes interactions with "plant" while the main work seems to be about spaceship control?  done

2.11 Fig. 8 ✅ The structure of the Reward Shaping function (RSF) and its interaction with the Potential-based Fitness (PbF): Title is too general. The content is not clear. What are the parameters in "Reward shaping" block? How is it working? 

It would be also good to add reference to Table 5 to help the Reader to find needed info. 

2.12 ✅ Eq. 20, eq. 21: presentation of those equations is improper. 

✅ 2.13 Fig. 9 Structure of an Individual and its GA terms: Title is too general. Are parameters a-e related with parameters a-e on Fig. 8? If no then this can be misleading. If yes, then why parameters f-k are not presented on Fig. 9? 

 2.14 V Fig. 10 Algorithm of the PbGA Search for the Optimal PbF: conditional block is not clear. It should include the whole conditional being tested, not half of it. - 

V 2.15 Fig. 11 GA Search of the Fittest Individual: The title is too general. The figure presents schematic which seems to make no sense. Maybe more context within the title would make it more clear. 

But I suggest to rethink this schematic. It has not much in common with Algorithm 1 (titles suggest both should present the same).  removed 

V 2.16 Table 5 Individual Chromosome of the Reward Shaping Function Components: Title is too general. Are the presented values initial for all the individuals?  yes, indeed. 

V 2.17 Fig 13 Fuel Consumption of Main engine FE,, Fig 14 Fuel Consumption of Side Thruster FS - it is hard to understand what is the meaning of the horizontal axis. Is it the percentage of time of the landing maneuver or throttle setting of the given engine? Are those only examples or general characteristics of the engines assumed in the considered model? If the latter then how this is motivated?  Rewritten in 5.2 5.2 Thrust Characteristics as Action’s Threshold

V 2.18 Fig 15 PbGA Search Phases in 15 Generations: The title is too general. The figure presents schematic which seems to make no sense. Maybe more context within the title would make it more clear. But I suggest to rethink this schematic. 

V 2.19 ✅ Fig 16. The tile refers to "a" and "b". Where are "a" and "b"? 

V 2.20 Table 6 Individual #1 - #11 Chromosome Mapping: Title is too general. How those values are used? How values of a and b are related to values of a and b for first individual in Table 5 (a is different in both tables)?  done 

2.21 ✅ Table 10: "Individu" => "Individual" 

f. ✅ "Few ways to get around this issue are by introducing a target network to reduce the greediness of max-value of Q (?)" 

V What is the meaning of the "?" character? Missing reference? 

V g. It can be hard for the Reader to understand what are "Leg conditions". What are left and right leg? Graphical presentation of the model of the considered spaceship would be very helpful.  shown in Figure 11

Comment : considered models and obtained results should be analyzed statistically (e.g. n-fold cross validation should be used to make results less dependent on selection of examples to train/test sets),

We conducted the cross validation methodology in Table 12 (Heatmap of Cross-validation of NN trained model against RSF-input test) to recheck our previous findings in Table 9 (Comparison of Elite Generation to PID and Reference Individuals) by combining them with Table 10 (RSF Components of the best five individuals) as test inputs, we found NN Individual #17 is proven to gain highest score independently to RSF input test. 

V a. (e.g. n-fold cross validation should be used to make results less dependent on selection of examples to train/test sets), 

V B. The manuscript must describe a technically sound piece of scientific research with data that supports the conclusions. 

Experiments must have been conducted rigorously, with 

-appropriate controls,  The delicate usage of control command Fe, Fs, Phi are already explained in Table 6, Fig 19,20, 21, 24, and 25 as action ratio. 

 -replication,  Ferrante is the reference model and replicated in individual #29

- and sample sizes.  1000x Monte Carlo are conducted in Fig 27 and 28

The conclusions must be drawn appropriately based on the data presented.  conclusion updated

V C. Many parameters of considered simulations seem to be initialized randomly (e.g. connection weights in neural networks, chromosomes of initial population in genetic algorithm). 

Thus experiments, results processing and presentation should be done to analyze statistical nature of analyzed method  in Fig 27, 28

Experiments should be repeated, n-fold cross-validation can be used during training/testing of NN based methods.  cross validation and MC 

Comment : measurement errors should be analyzed,

a. In Table 6 Authors present specification of the hardware used during experiments. 

Is there any influence of the details of the hardware specification on the results? 

If yes then this should be discussed within the text in detail. 

If not then please remove not needed information.  removed

B. Measured values should be presented with error bars and confidence intervals, etc. 

Comparison of outcomes of proposed method with results of other methods should be done with 

formal methods (statistic tests).  We introduce error bars in the performance's Monte Carlo test in Fig 27, 28

Comment : description of ideas and experiments including data processing should be clarified,

V a. Is the individual represented using binary vectors (0, 1) or float numbers (from 0.0 to 1.0)? Please make it clear. It’s a little obscure how the authors represent an individual in the GA.  float numbers are used in Table 5

V b. The section about the genetic operators' crossover and mutation is confusing, and I suggest a simple example using images instead of only the text explanation.  we built diagram crossover in 4.3.3/4.3.2  crossover = addition/substraction , mutation = multiplication/division 

- 

Comment : claims not supported by presented data should be removed or modified.

V a. claims not supported by presented data should be removed or modified.  MC 

V b. conclusion is weak and should be expanded as there is a lot to discuss and point out for future research. 

V c. Conclusions given by the Authors are not backed up by the presented data (see remark 3.5 - cross validation). 

The conclusion is updated becomes :

From the experiments done regarding the search of GA-optimized DDPG controller for powered descent guidance of a VTVL rocket, few notes can be taken: 

Deterministic control approach represented by PID is consistently gaining higher performance than the stochastic approaches can deliver, this is acceptable because PID usually excels for a specific and limited case of operation which fits for this rocket landing case. Therefore, it is not a fair method to compare deterministic approach with stochastic one in terms of accuracy. However, stochastic approaches demonstrate their strength when it comes to the problem of optimization and disturbance rejection. Reference DDPG and the GA-assisted DDPG constantly consumed lower amount of fuel compared to the PID approach, with acceptable position and speed performance. But the most striking discovery is the chattering phenomenon of the main engine throttle command or Fe, where GA-based DDPG has the lowest chatter and PID has the highest one. Chattering phenomenon in control system is not desired, especially for This new challenge is saved for the next research to be studied and solved. 

As results data revealed from the exhaustive Monte Carlo experiments, GA-DDPG convincingly surpass DDPG in terms of all aspect’s performance, especially when the wind disturbance is introduced. GA-DDPG beats DDPG in shorter flight time, shorter miss distance, and lower vertical velocity at touchdown. Few remarks can be observed here, where the GA-DDPG have high number of standard deviation for miss distance performance in no-wind condition, this is perfectly agreed as from the raw MC graphics the agent tends to land at -1.5m/s or 2m/s. The second remark is when the landing speed of reference DDPG performed well at no-wind condition but abysmally at wind condition, which suggests that DDPG is prone to disturbance, while GA-DDPG is quite stable against it. This confirms that wind disturbances affect all controllers’ performance, but GA-DDPG is proven to be more robust comparing reference DDPG. 

Few suggestions are considered for further research, such as: embedding the trained algorithms to popular microcontroller (Arduino, Raspberry Pi, MyRIO) via established code-authoring framework (LabView, MATLAB, Python) to control a vertical flying vehicle mockup, apply a software/hardware in the loop research scheme by using a high-fidelity flight simulator that can simulate in 3D with minimal scaling. From the cross-validation phase, an attractive operation needs to be studied further, that is the fine-tuning of the RSF to beat the already found best controller individual, traversing GA’s bigger family which is evolutionary algorithm will be conducted to enhance GA algorithm.

 Comment : share the datas 

The PLOS Data policy requires authors to make all data underlying the findings described in their manuscript fully available without restriction,

Response : Yes, we will comply completely to this rule of data sharing as it is required also by the University. But, we prioritized to finish the manuscript first. 

Not yet implemented actions.

- Legend information following all figure caption's titles

- The making of Confidence Interval graphs for the MonteCarlo testings

We thank you once again for your constructive and thorough reviews.

---

## [Decision Letter · Decision Letter 1]

21 Nov 2022

PONE-D-22-18693R1Usage of potential-based genetic algorithm to optimize the reward shaping function of a deep reinforcement learning for landing guidance of a reusable launch vehiclePLOS ONE

Dear Dr. Nugroho,

Thank you for submitting your manuscript to PLOS ONE. It was carefully reviewed by three Reviewers including me as an Academic Editor (Reviewer #3). After careful consideration, we feel that it has merit but does not fully meet PLOS ONE’s publication criteria as it currently stands. Therefore, we invite you to submit a revised version of the manuscript that addresses the points raised during the review process. In particular:    language problems should be removed,    presentation problems should be removed,    the process of statistical analysis of results should be described in detail (e.g. number of folds of cross validation should be given)    measurement errors / confidence intervals should be analyzed,    description of experiments including data processing should be clarified.Please submit your revised manuscript by Jan 05 2023 11:59PM. If you will need more time than this to complete your revisions, please reply to this message or contact the journal office at plosone@plos.org. Please include the following items when submitting your revised manuscript:A rebuttal letter that responds to each point raised by the academic editor and reviewer(s). You should upload this letter as a separate file labeled 'Response to Reviewers'.A marked-up copy of your manuscript that highlights changes made to the original version. You should upload this as a separate file labeled 'Revised Manuscript with Track Changes'.An unmarked version of your revised paper without tracked changes. You should upload this as a separate file labeled 'Manuscript'.If applicable, we recommend that you deposit your laboratory protocols in protocols.io to enhance the reproducibility of your results. Protocols.io assigns your protocol its own identifier (DOI) so that it can be cited independently in the future. For instructions see: https://journals.plos.org/plosone/s/submission-guidelines#loc-laboratory-protocols. Additionally, PLOS ONE offers an option for publishing peer-reviewed Lab Protocol articles, which describe protocols hosted on protocols.io. Read more information on sharing protocols at https://plos.org/protocols?utm_medium=editorial-email&utm_source=authorletters&utm_campaign=protocols.

We look forward to receiving your revised manuscript.

Kind regards,

Maciej Huk, Ph.D.

Academic Editor

PLOS ONE

Journal Requirements:

Reviewers' comments:

Reviewer's Responses to Questions

**Comments to the Author**

1. If the authors have adequately addressed your comments raised in a previous round of review and you feel that this manuscript is now acceptable for publication, you may indicate that here to bypass the “Comments to the Author” section, enter your conflict of interest statement in the “Confidential to Editor” section, and submit your "Accept" recommendation.

Reviewer #1: All comments have been addressed

Reviewer #2: All comments have been addressed

Reviewer #3: (No Response)

2. Is the manuscript technically sound, and do the data support the conclusions?

Reviewer #1: Yes

Reviewer #2: Partly

Reviewer #3: Partly

3. Has the statistical analysis been performed appropriately and rigorously? 

Reviewer #1: Yes

Reviewer #2: Yes

Reviewer #3: I Don't Know

4. Have the authors made all data underlying the findings in their manuscript fully available?

Reviewer #1: Yes

Reviewer #2: Yes

Reviewer #3: No

5. Is the manuscript presented in an intelligible fashion and written in standard English?

Reviewer #1: Yes

Reviewer #2: Yes

Reviewer #3: No

6. Review Comments to the Author

Reviewer #1: The authors have addressed all issues, and I think it should be accepted. Also, it is a solid contribution to current the literature.

Reviewer #2: (No Response)

Reviewer #3: >>> 1. Language problems:

1.1 "Chattering phenomenon in control system is not desired, especially for This new challenge is saved for the next research to be studied and solved."

grammar

>>> 2. Presentation problems:

2.1 It is not clear why ab and cd components are not presented in Table 5, Table 10

2.2 Fig 8. Title is too general. What was the control algorithm?

2.3 Fig 12. It is not clear how the presented crossover works.

- Where is mentioned 1 point crossover or 2 points crossover?

- four parents are used to create one individual?

2.4 Fig 12. Presenting not needed individuals make no sense if the goal is to explain how genetic operator works.

2.5 Fig. 11 - bounding box of the figure is not an a rectangle - it looks as a lack of care about the details.

>>> 3. Other problems:

3.1 Table 12. Heatmap of Cross-validation of NN trained model against RSF-input test

It is hard to understand what this has to do with cross-validation. How different was behavior of NN models created in different steps of cross-validation (for test data)?

3.2 Fig 17: The figure presents schematic which seems to make no sense in relation to the title. Any GA is a loop of generation of new populations made with genetic operations. Here - no main loop. (not fixed in relation to the previous review despite Authors claim it was fixed)

3.3 It is not clear what is the relation between Fig 9 "PbGA Search" and Fig 17. "PbGA search phases"

3.4 Authors write: "We conducted the cross validation methodology in Table 12"

It is not clear what was done. What was the number of folds, was stratified CV used ore some other?

3.5 Authors in their response write:

"Table 5. Individual Chromosome of the Reward Shaping Function Components: Title is too general. Are the presented values initial for all the individuals?  yes, indeed."

Thus the title of actual Table 3 is not clear. It should be written in the tile that those are initial values for all the individuals.

===EOT===

7. PLOS authors have the option to publish the peer review history of their article (what does this mean?). If published, this will include your full peer review and any attached files.

Reviewer #1: **Yes: **Valdecy Pereira

Reviewer #2: No

Reviewer #3: No

---

## [Author Response · Author response to Decision Letter 1]

15 Feb 2023

Dear PLOS ONE Reviewers, 

Thank you very much for your valuable remarks and correction. 

 According that, we make many adjustments and updates, but the most significant one is the introduction of a new subtopic, that is hybridization of the stochastic approach with the deterministic one. 

The main background is due to the inability of the GA-based DDPG to beat PID as the classical control approach in terms of landing accuracy. This results ridiculed our hard-earned proposed novelty at its best. 

Therefore our teams, agreed to tackle this before submitting the second revision. 

 We proposed two ways to ameliorate our works : 

1. Revise the internal NN hyperparameters, RSF, etc. 

2. Combine or lend the power of PID to fill the gap of the DDPG, and vice versa.

The first approach seems demanding another bulk of time, therefore we proceed to second approach.

Apparently the latter also didn't give much hope at first, but then we realized we missed one big major basic thing... the PID must also conduct the drilling of Monte Carlo.

Bam ! 

The real face of PID is opened....but still is handsome...

At least, we found one big surprise, the rate of failure is quite high in PID, nearly 10%.

 The new hybrid approach also seems has many failures initially, but after we sort the results, the so much looked for treasure is found, that is the HYDESTOC at switching height of 15meters regardless of the TVC on/off condition. It has good behavior in many major aspects that even PID can't achieve.

 Thank you very much Editorial Team (Adrian Cyrus Luczon, Oriel Jerome Delas Alas Vida, Marinel Ersando et al.), Reviewer Team, and Mr. Maciej Huk, Ph.D. as Academic Team for your patience and trust on us. 

Here are our response to your second phase remarks. We are very sorry for the delay.

We looked eagerly for your responses. Thank you.

Best Regards,

Larasmoyo Nugroho

Reviewer #3: >>> 1. Language problems:

V 1.1 "Chattering phenomenon in control system is not desired, especially for This new challenge is saved for the next research to be studied and solved."

grammar

 The presence of chattering is problematic, especially for liquid propulsion engines, and requires additional research to find a solution

>>> 2. Presentation problems:

V 2.1 It is not clear why ab and cd components are not presented in Table 5, Table 10

V 2.2 Fig 8. Title is too general. What was the control algorithm?

V 2.3 Fig 12. It is not clear how the presented crossover works.

- Where is mentioned 1 point crossover or 2 points crossover?

- four parents are used to create one individual?

V 2.4 Fig 12. Presenting not needed individuals make no sense if the goal is to explain how genetic operator works.

2.5 Fig. 11 - bounding box of the figure is not an a rectangle - it looks as a lack of care about the details.

>>> 3. Other problems:

3.1 Table 12. Heatmap of Cross-validation of NN trained model against RSF-input test

It is hard to understand what this has to do with cross-validation. 

How different was behavior of NN models created in different steps of cross-validation (for test data)?

 To illustrate, the already trained NN reference DDPG (red ellipse) when tested using RSF from Ind #29 (blue ellipse), will retain the high number of fitness 5.5, but when tested with RSF Ind #57 (green ellipse) the fitness number drops to 5.0. This means that the NN Ind#17 is highly compatible with RSF #29, but has lower compatibility with RSF #57, this is accepted since reference DDPG is trained with RSF symmetrical to RSF #29. Therefore, performance consistency is validated via the number of fitness here. 

3.2 Fig 17: The figure presents schematic which seems to make no sense in relation to the title. Any GA is a loop of generation of new populations made with genetic operations. Here - no main loop. (not fixed in relation to the previous review despite Authors claim it was fixed)

 GA search is a loop of generation producing new offsprings using the genetic operators, in our case the loop is conducted from the last phase before final selection that is elitism phase, to the phase where the replication generation is utilized, i.e. Order Reduction Proportional Mutation Mapping. In other word, the replication generation plays a significant role in our search. 

3.3 It is not clear what is the relation between Fig 9 "PbGA Search" and Fig 17. "PbGA search phases"

 This loop from Fig 17 share the same activity with the loop in Fig 9. The loop from Fig 9 shows the reviewing process of each offspring, which is extensively used in the loop of Fig 17.

3.4 Authors write: "We conducted the cross validation methodology in Table 12"

It is not clear what was done. What was the number of folds, was stratified CV used ore some other?

Proposed cross validation in RL is used to determine the best combination of a neural network trained model and a reward shaping function (RSF) through resampling based on the fitness numbers. In our proposed approach, the fitness resampling is conducted over 150 testing iterations, resulting in a matrix that compares trained NN models on the x-axis and tested RSFs on the y-axis, as shown in Table 12.

 To illustrate, the already trained NN reference DDPG (red ellipse) when tested using RSF from Ind #29 (blue ellipse), will retain the high number of fitness 5.5, but when tested with RSF Ind #57 (green ellipse) the fitness number drops to 5.0. This means that the NN Ind#17 is highly compatible with RSF #29, but has lower compatibility with RSF #57, this is accepted since reference DDPG is trained with RSF symmetrical to RSF #29. Therefore, performance consistency is validated via the number of fitness here. 

3.5 Authors in their response write:

"Table 5. Individual Chromosome of the Reward Shaping Function Components: Title is too general. Are the presented values initial for all the individuals? 

 yes, indeed."

Thus the title of actual Table 3 is not clear. It should be written in the tile that those are initial values for all the individuals.

---

## [Decision Letter · Decision Letter 2]

27 Mar 2023

PONE-D-22-18693R2Enhancing the landing guidance of a reusable launch vehicle by improving genetic algorithm-based deep reinforcement learning using Hybrid Deterministic-Stochastic algorithmPLOS ONE

Dear Dr. Nugroho,

Thank you for submitting your manuscript to PLOS ONE. It was analyzed by five Reviewers including me as an Academic Editor (Reviewer #3). Please also notice, that too keep high standard of the reviewing process reviews received from Reviewer #5 and Reviewer #6 are considered but with low confidence (those reviews are not detailed enough). In the effect and after careful consideration, we feel that your manuscript has merit but does not fully meet PLOS ONE’s publication criteria as it currently stands. Therefore, we invite you to submit a revised version of the manuscript that addresses the points raised during the review process.

In particular:the language needs improvements,statistical analysis of results should be presented,analysis of the accuracy of measurements should be presented,the text should be made more easy to follow,presentation problems need to be removed (e.g. numbering of figures and tables, text formatting).

We look forward to receiving your revised manuscript.

Kind regards,

Maciej Huk, Ph.D.

Academic Editor

PLOS ONE

Reviewers' comments:

Reviewer's Responses to Questions

**Comments to the Author**

1. If the authors have adequately addressed your comments raised in a previous round of review and you feel that this manuscript is now acceptable for publication, you may indicate that here to bypass the “Comments to the Author” section, enter your conflict of interest statement in the “Confidential to Editor” section, and submit your "Accept" recommendation.

Reviewer #1: All comments have been addressed

Reviewer #3: (No Response)

Reviewer #4: (No Response)

Reviewer #5: All comments have been addressed

Reviewer #6: All comments have been addressed

2. Is the manuscript technically sound, and do the data support the conclusions?

Reviewer #1: Yes

Reviewer #3: Partly

Reviewer #4: Partly

Reviewer #5: Yes

Reviewer #6: Yes

3. Has the statistical analysis been performed appropriately and rigorously? 

Reviewer #1: Yes

Reviewer #3: No

Reviewer #4: No

Reviewer #5: Yes

Reviewer #6: Yes

4. Have the authors made all data underlying the findings in their manuscript fully available?

Reviewer #1: Yes

Reviewer #3: Yes

Reviewer #4: Yes

Reviewer #5: Yes

Reviewer #6: Yes

5. Is the manuscript presented in an intelligible fashion and written in standard English?

Reviewer #1: Yes

Reviewer #3: No

Reviewer #4: No

Reviewer #5: Yes

Reviewer #6: Yes

6. Review Comments to the Author

Reviewer #1: (No Response)

Reviewer #3: 

>>> 1. Language problems: not detected.

>>> 2. Presentation problems:

2.1 Fig 12. row #21: third "1" from the right needs to be justified or changed to "0".

2.2. Fig 142. => Fig 22 (page 31)

2.3 Numbering of figures is improper. After Fig 25 one can see Figures 15-17.

2.4 Table 1413. => Table 14. (page 35)

2.5 Text formatting should be done according to the PLOS ONE standard (e.g. titles of figures/tables do not need bold font, references should be in the format expected by PLOS ONE)

2.6 Titles should be centered below figures (see e.g. fig 20).

2.7 Title of table 12 is too general (information about "1000 times Monte Carlo test" would be helpful.

>>> 3. Other problems:

3.1 All abbreviations need to be explained before their first use (e.g. RLV, PID).

Moreover, over-usage of abbreviations should be limited.

3.2 Statistical analysis of results is not presented.

3.3 Analysis of the accuracy of measurements is not presented.

3.4 In many places Authors write "fitness number" (also within Fig. 8). More popular term in "fitness value".

3.5 The text is hard to follow. The meaning of the diagrams is often hard to understand without reading the whole text. An initial, clear top down graphical presentation showing what is needed, performed and achieved within presented work would be bvery helpful.

>>> Recommendation: major rework.

Reviewer #4: The authors aimed to develop a method for controlling a rocket using reinforcement learning. I believe several aspects of the manuscript need improvements.

Here are my key points:

- The language needs improvements.

- What do RLV and PID stand for? I could not find where the authors explain these two in the manuscript. The authors should consider that not all readers are familiar with some technical jargon and acronyms the authors used. Introduce every acronym before using it in the text, not the other way around.

- It is minor, but the way authors use citations is cumbersome; for example, in the third paragraph in section 3.1.

- I suggest the authors create a visualization that explains their methodologies and shows how the authors used different algorithms, compared these algorithms, and made a conclusion. This way, the manuscript will be easier to follow.

- What is the main performance criterion for comparing the algorithms? In the manuscript, the authors mentioned three performance criteria: accuracy, time, and fuel efficiency. However, it was not clear how they choose the best (balanced) solution based on these criteria (taking into account inconsistencies in each algorithm's performance the authors explained).

- Most of the evaluation was based on visual inspection. I suggest the authors add some statistical tests to validate the results.

To summarize, the manuscript has the potential for publication. However, the manuscript was hard to follow due to unclear methodologies and conclusions.

Reviewer #5: The Revised Paper stands Accepted with no further revisions. All the comments are properly addressed and no further revisions are required.

Reviewer #6: Enhancing the landing guidance of a reusable launch vehicle by improving genetic algorithm-based deep reinforcement learning using Hybrid Deterministic-Stochastic algorithm is presented in this paper. Paper is revised well. I can accept now.

7. PLOS authors have the option to publish the peer review history of their article (what does this mean?). If published, this will include your full peer review and any attached files.

Reviewer #1: **Yes: **Valdecy Pereira

Reviewer #3: No

Reviewer #4: No

Reviewer #5: No

Reviewer #6: No

---

## [Author Response · Author response to Decision Letter 2]

16 May 2023

Dear Team Reviewer of PLOS ONE

Here are my responses to the third reviews.

Answer to Reviewer 3

>>> 1. Language problems: not detected.

>>> 2. Presentation problems:

V2.1 Fig 12. row #21: third "1" from the right needs to be justified or changed to "0".  ok

V2.2. Fig 142. => Fig 22 (page 31)  ok

V2.3 Numbering of figures is improper. After Fig 25 one can see Figures 15-17.  ok

V2.4 Table 1413. => Table 14. (page 35)

V2.5 Text formatting should be done according to the PLOS ONE standard (e.g. titles of figures/tables do not need bold font, references should be in the format expected by PLOS ONE)

V2.6 Titles should be centered below figures (see e.g. fig 20).

V2.7 Title of table 12 is too general (information about "1000 times Monte Carlo test" would be helpful.

- table 12 is updated. 

>>> 3. Other problems:

V3.1 All abbreviations need to be explained before their first use (e.g. RLV, PID).

Moreover, OVERUSAGE of abbreviations should be limited.  considered and updated

 checked

V3.2 Statistical analysis of results is not presented.  incorporated

-mean,  nearly all tables employed mean formula

-freq.dist, freq. density, q1-q3,  applied it in figure 21, 27, 31

-anova, null hypothesis  applied in table 15, 16, 18, 19

3.3 Analysis of the accuracy of measurements is not presented.

-drag between fin and finless.. not enough time

V3.4 In many places Authors write "fitness number" (also within Fig. 8). More popular term in "fitness value".  updated

V 3.5 The text is hard to follow. The meaning of the diagrams is often hard to understand without reading the whole text. 

.. in figure 1, ddpg streamlined 

.. Figure 6, ga-ddpg + pid framework reconstructed

V3.6 An initial, clear top down graphical presentation showing what is needed, performed and achieved within presented work would be very helpful

.. figure 5, research methodology reflowed

>>> Recommendation: major rework.

*+*+*+*+*+*+*+*+*+*+*+*+

Answer to review 4

Reviewer #4: The authors aimed to develop a method for controlling a rocket using reinforcement learning. I believe several aspects of the manuscript need improvements.

Here are my key points:

- The language needs improvements.  yes, i was too focused on the technical sides

- What do RLV and PID stand for? I could not find where the authors explain these two in the manuscript. 

The authors should consider that not all readers are familiar with some technical jargon and acronyms the authors used. 

Introduce every acronym before using it in the text, not the other way around.

 explained the first acronyms

V- It is minor, but the way authors use citations is cumbersome; for example, in the third paragraph in section 3.1.  okey

V- I suggest the authors create a visualization that explains their methodologies  Figure 1

and shows how the authors used different algorithms (PID and reference DDPG),  Figure 5,6 

compared these algorithms,  whole chapter 5

and made a conclusion.  check in conclusion

This way, the manuscript will be easier to follow.

V- What is the main performance criterion for comparing the algorithms? 

In the manuscript, the authors mentioned three performance criteria: accuracy, time, and fuel efficiency. 

However, it was not clear how they choose the best (balanced) solution based on these criteria  in GA : via mapping the RSF, crossover and mutate most impactful DNA, eliminate excessive DNA, in HYDESTOC : change height altitude, in PID adjust sensors pid

(taking into account inconsistencies in each algorithm's performance the authors explained).

V- Most of the evaluation was based on visual inspection. 

I suggest the authors add some statistical tests to validate the results.

table 15, 16, 18, 19,20

To summarize, the manuscript has the potential for publication. 

However, the manuscript was hard to follow due to unclear methodologies and conclusions.

*+*+*+*+*+

Reviewer #5: The Revised Paper stands Accepted with no further revisions. All the comments are properly addressed and no further revisions are required.

*+*+*+*+*+

Reviewer #6: Enhancing the landing guidance of a reusable launch vehicle by improving genetic algorithm-based deep reinforcement learning using Hybrid Deterministic-Stochastic algorithm is presented in this paper. Paper is revised well. I can accept now.

---

## [Decision Letter · Decision Letter 3]

4 Aug 2023

PONE-D-22-18693R3Enhancing the landing guidance of a reusable launch vehicle by improving genetic algorithm-based deep reinforcement learning using Hybrid Deterministic-Stochastic algorithmPLOS ONE

Dear Dr. Nugroho,

Thank you for submitting your manuscript to PLOS ONE. It was reviewed by two Reviewers including me as and Academic Editor (Reviewer #3). After careful consideration, we feel that it has merit but does not fully meet PLOS ONE’s publication criteria as it currently stands. Therefore, we invite you to submit a revised version of the manuscript that addresses the points raised during the review process. In particular:quality of presentation should be improved,descriptions which are hard to understand need to be made clear,references which link to not existing resources or are of inadequate quality should be removed.Please submit your revised manuscript by Sep 18 2023 11:59PM. If you will need more time than this to complete your revisions, please reply to this message or contact the journal office at plosone@plos.org. Please include the following items when submitting your revised manuscript:A rebuttal letter that responds to each point raised by the academic editor and reviewer(s). You should upload this letter as a separate file labeled 'Response to Reviewers'.A marked-up copy of your manuscript that highlights changes made to the original version. You should upload this as a separate file labeled 'Revised Manuscript with Track Changes'.An unmarked version of your revised paper without tracked changes. You should upload this as a separate file labeled 'Manuscript'.

We look forward to receiving your revised manuscript.

Kind regards,

Maciej Huk, Ph.D.

Academic Editor

PLOS ONE

Reviewers' comments:

Reviewer's Responses to Questions

**Comments to the Author**

1. If the authors have adequately addressed your comments raised in a previous round of review and you feel that this manuscript is now acceptable for publication, you may indicate that here to bypass the “Comments to the Author” section, enter your conflict of interest statement in the “Confidential to Editor” section, and submit your "Accept" recommendation.

Reviewer #3: (No Response)

Reviewer #4: All comments have been addressed

2. Is the manuscript technically sound, and do the data support the conclusions?

Reviewer #3: Partly

Reviewer #4: Yes

3. Has the statistical analysis been performed appropriately and rigorously? 

Reviewer #3: I Don't Know

Reviewer #4: Yes

4. Have the authors made all data underlying the findings in their manuscript fully available?

Reviewer #3: Yes

Reviewer #4: Yes

5. Is the manuscript presented in an intelligible fashion and written in standard English?

Reviewer #3: No

Reviewer #4: Yes

6. Review Comments to the Author

Reviewer #3: >>> 1. Language problems: not detected.

>>> 2. Presentation problems:

2.1 Title of table 12 is too general (information about "1000 times Monte Carlo test" would be helpful.

2.2 Fig 18, Fig 19, Fig 26 are of low quality

>>> 3. Other problems:

3.1 In many places Authors write "fitness value". Above Fig. 21 Authors use "fitness number" term instead:

"which don’t focus in reward or fitness number afterward". Please use "fitness value" also in this place.

3.2 Under Fig. 21 Authors write: "Probability density of the benchmarked algorithm under MC-100 testings"

It is unclear what is the "Probability density of an algorithm". Authors should make the title clear,

3.3 Fig. 27: It is unclear what is the "Density" and how it relates to claimed improvement.

3.4 References [4], [11], [33], [39] and [51] link to not existing resources. This should not happen.

3.5 Are Authors sure that consumer-level comparison of products published on youtube (reference [61]) is the right quality reference to be used within scientific text? E.g. can Authors guarantee that this will not become unavailable soon. Was this test performed with scientific scrutinity?

3.6 The text is hard to follow. The meaning of many diagrams is hard to understand without reading the whole text.

In the effect it is hard to understand what is presented and/or to verify it. I think this is the main drawback of the manuscript in its actual form. This highly limits the value of this manuscript.

>>> Recommendation: major rework.

Reviewer #4: The manuscript is worth publishing. The authors have addressed all my questions and concerns.

7. PLOS authors have the option to publish the peer review history of their article (what does this mean?). If published, this will include your full peer review and any attached files.

Reviewer #3: No

Reviewer #4: No

---

## [Author Response · Author response to Decision Letter 3]

12 Sep 2023

Dear PLOS ONE Reviewer Team, 

Kindly check our corrections to your reviews:

Reviewer#3:

>>> 2. Presentation problems:

V 2.1 Title of table 12 is too general (information about "1000 times Monte Carlo test" would be helpful.

 done

2.2 

VFig 18, 

VFig 19, 

VFig 26 are of low quality

>>> 3. Other problems:

V3.1 In many places Authors write "fitness value". Above Fig. 21 Authors use "fitness number" term instead:

"which don’t focus in reward or fitness number afterward". Please use "fitness value" also in this place.

 done

V3.2 Under Fig. 21 Authors write: "Probability density of the benchmarked algorithm under MC-100 testings"

It is unclear what is the "Probability density of an algorithm". Authors should make the title clear,

 Probability density of an algorithm = results freq. of landing performance conducted by benchmarked control algorithm  page 31/51

V3.3 Fig. 27: It is unclear what is the "Density" and how it relates to claimed improvement. 

 density = frequency 

" Density of probability is utilized here to describe the frequency distribution of the landing performance, in terms of final horizontal position and final landing speed. Higher density means higher frequency of occurrence that can be exerted by the corresponding algorithm. HYDESTOC-8 shows a higher one than the pure stochastic controllers. From mean value in Figure 27 and Table 14, it is emphasized that the algorithm HYDESTOC-8 shows significant improvement over the stochastic controller (DDPG, GA-DDPG) and slight improvement over the deterministic one. 

. " page 38 of 51

3.4 References [4], [11], [33], [39] and [51] link to not existing resources. This should not happen.

 corrected

V3.5 Are Authors sure that consumer-level comparison of products published on youtube (reference [61]) is the right quality reference to be used within scientific text? E.g. can Authors guarantee that this will not become unavailable soon. Was this test performed with scientific scrutinity?

 updated

3.6 The text is hard to follow. 

The meaning of many diagrams is hard to understand without reading the whole text. 

 We reviewed carefully all diagrams and make sure that all diagrams are covered and explained in the paragraphs

---

## [Decision Letter · Decision Letter 4]

25 Sep 2023

Enhancing the landing guidance of a reusable launch vehicle by improving genetic algorithm-based deep reinforcement learning using Hybrid Deterministic-Stochastic algorithm

PONE-D-22-18693R4

Dear Dr. Nugroho,

We’re pleased to inform you that your manuscript has been judged scientifically suitable for publication and will be formally accepted for publication once it meets all outstanding technical requirements.

Kind regards,

Maciej Huk, Ph.D.

Academic Editor

PLOS ONE

Additional Editor Comments (optional):

Reviewers' comments:

Reviewer's Responses to Questions

**Comments to the Author**

1. If the authors have adequately addressed your comments raised in a previous round of review and you feel that this manuscript is now acceptable for publication, you may indicate that here to bypass the “Comments to the Author” section, enter your conflict of interest statement in the “Confidential to Editor” section, and submit your "Accept" recommendation.

Reviewer #3: (No Response)

2. Is the manuscript technically sound, and do the data support the conclusions?

Reviewer #3: Yes

3. Has the statistical analysis been performed appropriately and rigorously? 

Reviewer #3: I Don't Know

4. Have the authors made all data underlying the findings in their manuscript fully available?

Reviewer #3: Yes

5. Is the manuscript presented in an intelligible fashion and written in standard English?

Reviewer #3: Yes

6. Review Comments to the Author

Reviewer #3: Authors have improved most of indicated problems. The manuscript is still packed with a lot of hard to understand diagrams and data but maybe this is the nature of described problem. Language problems were not detected.

Manuscript can be considered for publication.

Recommendation: accept

7. PLOS authors have the option to publish the peer review history of their article (what does this mean?). If published, this will include your full peer review and any attached files.

Reviewer #3: No

---

## [Editor Report · Acceptance letter]

28 Nov 2023

PONE-D-22-18693R4 

Enhancing the landing guidance of a reusable launch vehicle by improving genetic algorithm-based deep reinforcement learning using Hybrid Deterministic-Stochastic algorithm 

Dear Dr. Wijaya:

I'm pleased to inform you that your manuscript has been deemed suitable for publication in PLOS ONE. Congratulations! Your manuscript is now with our production department. 

Kind regards, 

on behalf of

Dr. Maciej Huk 

Academic Editor

PLOS ONE